# Mapping brain-behavior space relationships along the psychosis spectrum

Jie Lisa Ji[1,2]*, Markus Helmer[1], Clara Fonteneau[1], Joshua B Burt[3], Zailyn Tamayo[1], Jure Demšar[4,5], Brendan D Adkinson[1,2], Aleksandar Savić[6], Katrin H Preller[7], Flora Moujaes[7], Franz X Vollenweider[7], William J Martin[8], Grega Repovš[6], Youngsun T Cho[1,9], Christopher Pittenger[1,9], John D Murray[1,2,10], Alan Anticevic[1,2,11]*

[1]Department of Psychiatry, Yale University School of Medicine, New Haven, United States; [2]Interdepartmental Neuroscience Program, Yale University School of Medicine, New Haven, United States; [3]RBNC Therapeutics, San Francisco, United States; [4]Department of Psychology, University of Ljubljana, Ljubljana, Slovenia; [5]Faculty of Computer and Information Science, University of Ljubljana, Ljubljana, Slovenia; [6]Department of Psychiatry, University of Zagreb, Zagreb, Croatia; [7]Department of Psychiatry, Psychotherapy and Psychosomatics, University Hospital for Psychiatry Zurich, Zurich, Switzerland; [8]The Janssen Pharmaceutical Companies of Johnson and Johnson, San Francisco, United States; [9]Child Study Center, Yale University School of Medicine, New Haven, United States; [10]Department of Physics, Yale University, New Haven, United States; [11]Department of Psychology, Yale University School of Medicine, New Haven, United States

**Abstract** Difficulties in advancing effective patient-specific therapies for psychiatric disorders highlight a need to develop a stable neurobiologically grounded mapping between neural and symptom variation. This gap is particularly acute for psychosis-spectrum disorders (PSD). Here, in a sample of 436 PSD patients spanning several diagnoses, we derived and replicated a dimensionality-reduced symptom space across hallmark psychopathology symptoms and cognitive deficits. In turn, these symptom axes mapped onto distinct, reproducible brain maps. Critically, we found that multivariate brain-behavior mapping techniques (e.g. canonical correlation analysis) do not produce stable results with current sample sizes. However, we show that a univariate brain-behavioral space (BBS) can resolve stable individualized prediction. Finally, we show a proof-of-principle framework for relating personalized BBS metrics with molecular targets via serotonin and glutamate receptor manipulations and neural gene expression maps derived from the Allen Human Brain Atlas. Collectively, these results highlight a stable and data-driven BBS mapping across PSD, which offers an actionable path that can be iteratively optimized for personalized clinical biomarker endpoints.

*For correspondence:
jielisa.ji@yale.edu (JLJ);
alan.anticevic@yale.edu (AA)

## Introduction

Mental health conditions cause profound disability, yet most treatments offer limited efficacy across psychiatric symptoms (*Tohen et al., 2003*; *McEvoy et al., 2007*; *aan het Rot et al., 2010*; *Cipriani et al., 2018*). A key step toward developing more effective therapies for specific psychiatric symptoms is reliably mapping them onto underlying neural targets. This goal is particularly

challenging because neuropsychiatric diagnoses and consequently drug development still operate under 'legacy' categorical constraints, which were not quantitatively informed by neural data.

Critically, diagnostic systems in psychiatry (e.g. the Diagnostic and Statistical Manual of Mental Disorders (DSM) [*American Psychiatric Association, 1994*]) were built to aid clinical consensus, but were not designed to guide quantitative mapping of symptoms onto neural alterations (*Phillips et al., 2008*; *Gillihan and Parens, 2011*). Consequently, the current diagnosis system cannot optimally map onto patient-specific brain-behavioral alterations. This challenge is particularly evident along the psychosis spectrum disorders (PSD) where there is notable shared symptom variation across distinct DSM diagnostic categories, including schizophrenia (SZP), schizo-affective (SADP), bipolar disorder with psychosis (BPP). For instance, despite BPP being a distinct diagnosis, BPP patients exhibit similar but attenuated psychosis symptoms and neural alterations similar to SZP (e.g. thalamic functional connectivity (FC) [*Anticevic et al., 2014*]). It is essential to quantitatively map such shared clinical variation onto common neural alterations, to circumvent categorical constraints for biomarker development (*Casey et al., 2013*; *Phillips et al., 2008*; *Gillihan and Parens, 2011*) – a key goal for development of neurobiologically informed personalized therapies (*Anticevic et al., 2014*; *Casey et al., 2013*).

Recognizing the limitations of categorical frameworks, the NIMH's Research Domain Criteria (RDoC) initiative introduced dimensional mapping of functional domains on to neural circuits (*Insel, 2014*). This motivated cross-diagnostic multisite studies for mapping PSD symptom and neural variation (*Tamminga et al., 2014*; *Koutsouleris et al., 2018*; *Casey et al., 2018*; *Di Martino et al., 2014*). Multivariate neurobehavioral analyses across PSD and mood spectra reported brain-behavioral relationships across diagnoses, with the goal of informing individualized treatment (*Drysdale et al., 2017*). These studies attempted to address the challenge of moving beyond traditional a priori clinical scales, which provide composite scores (*Cronbach and Meehl, 1955*) that may not optimally capture neural variation (*Gillihan and Parens, 2011*; *Barch et al., 2013*). For instance, despite many data-driven dimensionality-reduction symptom studies (*van der Gaag et al., 2006a*; *Lindenmayer et al., 1994*; *Emsley et al., 2003*; *Dollfus et al., 1996*; *Blanchard and Cohen, 2006*; *Chen et al., 2020*; *Lefort-Besnard et al., 2018*), a common approach in PSD neuroimaging research is still to sum 'positive' or 'negative' psychosis symptoms into a single score for relating to neural measures (*Ji et al., 2019a*; *Anticevic et al., 2014*). Importantly, neural alterations in PSD may reflect a more complex weighted combination of symptoms than a priori composite scores.

While multivariate neurobehavioral studies have a way to address this, such studies face the risk of failing to replicate due to overfitting (*Dinga et al., 2019*), arising from high dimensionality of behavioral and neural features and a comparatively limited number of subjects (*Helmer et al., 2020*). Notably, current state-of-the-art large-scale clinical neuroimaging studies have target enrollment totals of ~200–600 subjects (https://www.humanconnectome.org/disease-studies). We therefore wanted to test whether a reproducible neurobehavioral geometry can be derived with a sample size that is on par with current consortia studies in psychiatry.

We hypothesized that a linearly weighted low-dimensional symptom solution (capturing key disease-relevant information) may produce a robust and reporoducible univariate brain-behavioral mapping. Indeed, recent work used dimensionality reduction methods successfully to compute a neural mapping across canonical SZP symptoms (*Chen et al., 2020*). However, it remains unknown if this approach generalizes across PSD. Moreover, it is unknown if incorporating cognitive assessment, a hallmark and untreated PSD symptom (*Barch et al., 2013*), explains neural feature variation that is distinct from canonical PSD symptoms. Finally, prior work has not tested if a low-dimensional symptom-neural mapping can be predictive at the single patient level – a prerequisite for individualized clinical endpoints.

To inform these gaps, we first tested two key questions: (i) Can data-reduction methods reliably reveal symptom axes across PSD that include both canonical symptoms and cognitive deficits? (ii) Do these lower dimensional symptom axes map onto a reproducible brain-behavioral solution across PSD? Specifically, we combined fMRI-derived resting-state measures with psychosis and cognitive symptoms (*Canuso et al., 2008*; *Kay et al., 1987*) obtained from a public multi-site cohort of 436 PSD patients and 202 healthy individuals collected by the Bipolar-Schizophrenia Network for Intermediate Phenotypes (BSNIP-1) consortium across six sites in North America (*Tamminga et al., 2014*). The dataset included included 150 patients formally diagnosed with BPP, 119 patients diagnosed SADP, and 167 patients diagnosed with SZP (*Appendix 1—table 2*). This cohort enabled

cross-site symptom-neural comparisons across multiple psychiatric diagnostic categories, which we then mapped onto specific neural circuits. First, we tested if dimensionality reduction of PSD symptoms revealed a stable solution for individual patient prediction. Next, we tested if this low-dimensional symptom solution yields novel, stable, and statistically robust neural mapping compared to canonical composite symptom scores or DSM diagnoses. We then tested if the computed symptom-neural mapping is reproducible across symptom axes and actionable for individual patient prediction. Finally, we anchor the derived symptom-relevant neural maps by computing their similarity against mechanistically-informed neural maps. Here we used independently collected pharmacological fMRI maps from healthy adults in response to putative PSD receptor treatment targets (glutamate via ketamine and serotonin via LSD) (*Preller et al., 2018*; *Anticevic et al., 2015*). We also computed transcriptomic maps from the Allen Human Brain Atlas (AHBA) (*Hawrylycz et al., 2012*; *Burt et al., 2018*) for genes implicated in PSD. The primary purpose of this study was to derive a robust and reproducible symptom-neural geometry across symptom domains in chronic PSD that can be resolved at the single-subject level and mechanistically linked to molecular benchmarks. This necessitated the development of novel methods and rigorous power analytics (as we found that existing multivariate methods are drastically underpowered [*Helmer et al., 2020*]), which we have also presented in the paper for full transparency and reproducibility. This approach can be iteratively improved upon to inform neural mechanisms underlying PSD symptom variation, and in turn applied to other psychiatric neuro-behavioral spectra.

To our knowledge, no study to date has mapped a data-reduced symptom geometry encompassing a transdiagnostic PSD cohort across combined cognitive and psychopathology symptom domains, with demonstrable statistical stability and reproducibility at the single subject level. Additionally, this symptom geometry can be mapped robustly onto neural data to achieve reproducible group-level effects as well as individual-level precision following neural feature selection. Furthermore, while other other studies have evaluated relationships between neural maps and complementary molecular datasets (e.g. transcriptomics), no study to our knowledge has benchmarked single-subject selected neural maps using reproducible neurobehavioral features against both pharmacological neuroimaging and gene expression maps that may inform single subject selection for targeted treatment. Collectively, this study used data-driven dimensionality-reduction methods to derive reproducible symptom dimensions across 436 PSD patients and mapped them onto novel and robust neuroimaging features. These effects were then benchmarked against molecular imaging targets, outlining an actionable path toward personalized clinical neuro-behavioral endpoints.

## Results

The key questions and results of this study are summarized in *Appendix 1—table 5* and an overview of the workflow is presented in *Appendix 1—figure 1*. Additionally, for convenient reference, a glossary of the key terms and abbreviations used throughout the study are provided in *Appendix 1—table 1*.

### Dimensionality-reduced PSD symptom variation is stable and reproducible

First, to evaluate PSD symptom variation, we examined core PSD psychopathology metrics captured by two instruments: the Brief Assessment of Cognition in Schizophrenia (BACS) and the Positive and Negative Syndrome Scale (PANSS). We refer to these 36 items as 'symptom measures' throughout the rest of the paper. We observed group mean differences across DSM diagnoses (*Figure 1A*, p<0.05 Bonferroni corrected); however, symptom measure distributions revealed notable overlap that crossed diagnostic boundaries (*Tamminga et al., 2014*; *Keshavan et al., 2011*). Furthermore, we observed marked collinearity between symptom measures across the PSD sample (*Figure 1B*), indicating that a dimensionality-reduced solution may sufficiently capture meaningful varation in this symptom space. We hypothesized that such a dimensionality-reduced symptom solution may improve PSD symptom-neural mapping as compared to pre-existing symptom scales.

Here, we report results from a principal component analysis (PCA) as it produces a deterministic solution with orthogonal axes (i.e. no a priori number of factors needs to be specified) and explains all variance in symptom measures. Results were highly consistent with prior symptom-reduction

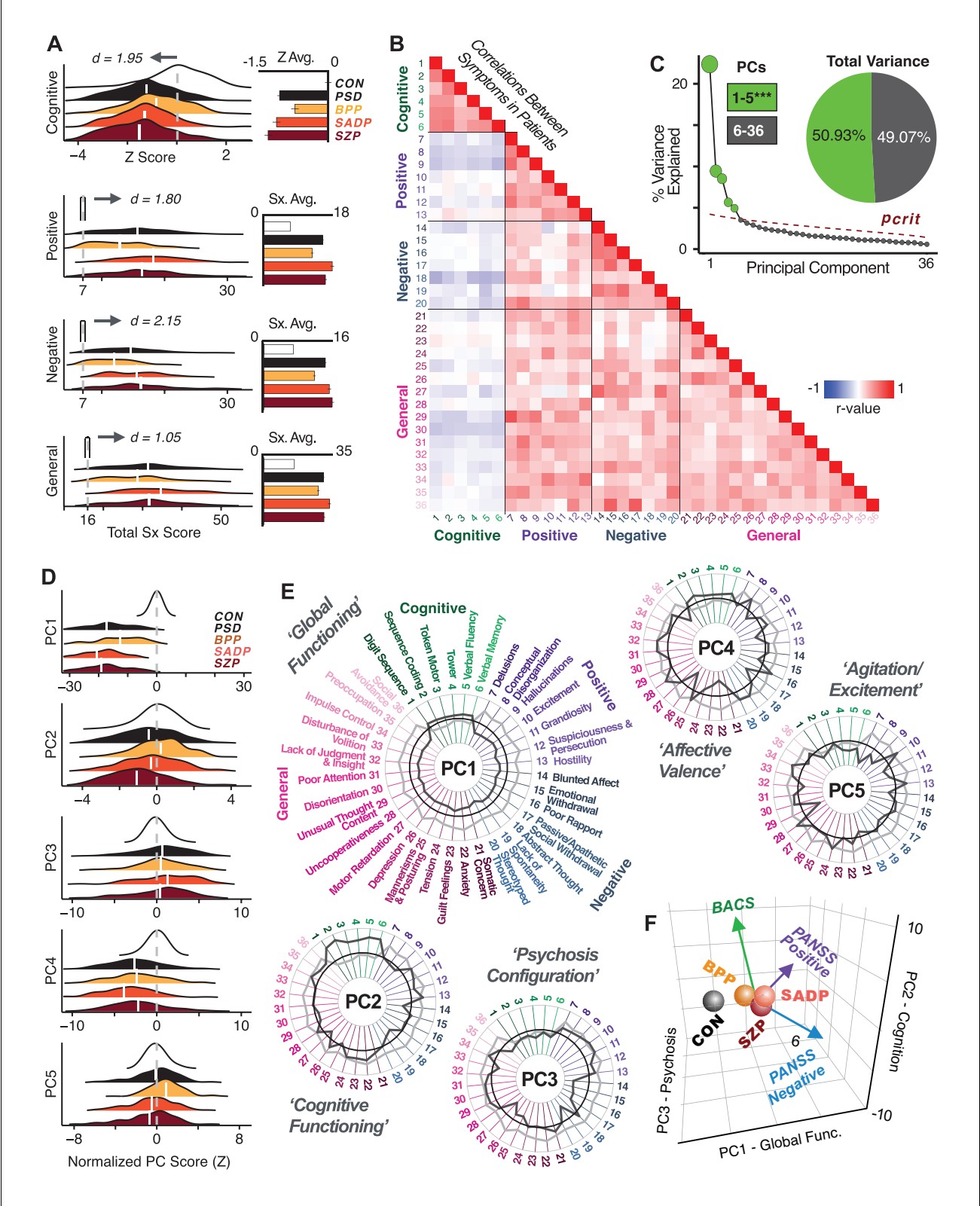

**Figure 1.** Quantifying data-driven low-dimensional variation of cross-diagnostic psychosis spectrum disorder (PSD) symptoms and cognitive deficits. (A) Distributions of symptom scores for each of the DSM diagnostic groups across core psychosis symptom measures (PANSS positive, negative, and general symptoms tracking illness severity) and cognitive deficits (BACS composite cognitive performance). BPP: bipolar disorder with psychosis (yellow, N = 150); SADP: schizo-affective disorder (orange, N = 119); SZP: schizophrenia (red, N = 167); All PSD patients (black, N = 436); Controls (white,

*Figure 1 continued on next page*

*Figure 1 continued*

N = 202). Bar plots show group means; error bars show standard deviations. (**B**) Correlations between 36 symptom measures across all PSD patients (N = 436). (**C**) Screeplot shows the % variance explained by each of the principal components (PCs) from a PCA performed using all 36 symptom measures across 436 PSD patients. The size of each point is proportional to the variance explained. The first five PCs (green) survived permutation testing (p<0.05, 5000 permutations). Together they capture 50.93% of all symptom variance (inset). (**D**) Distribution plots showing subject scores for the five significant PCs for each of the clinical groups, normalized relative to the control group. Note that control subjects (CON) were not used to derive the PCA solution; however, all subjects, including CON, can be projected into the data-reduced symptom geometry. (**E**) Loading profiles shown in dark gray for the 36 PANSS/BACS symptom measures on the five significant PCs. Each PC ('Global Functioning', 'Cognitive Functioning', 'Psychosis Configuration', 'Affective Valence', 'Agitation/Excitement') was named based on the pattern of loadings on symptom measures. See *Appendix 1— figure 2G* for numerical loading values. The PSD group mean score on each symptom measure is also shown, in light gray (scaled to fit on the same radarplots). Note that the group mean configuration resembles the *PC1* loading profile most closely (as *PC1* explains the most variance in the symptom measures). (**F**) PCA solution shown in the coordinate space defined by the first three PCs. Colored arrows show a priori composite PANSS/BACS vectors projected into the *PC1-3* coordinate space. The a priori composite symptom vectors do not directly align with data-driven PC axes, highlighting that PSD symptom variation is not captured fully by any one aggregate a priori symptom score. Spheres denote centroids (i.e. center of mass) for each of the patient diagnostic groups and control subjects. Alternative views showing individual patients and controls projected into the PCA solution are shown in *Appendix 1—figure 2A-F*.

studies in PSD: we identified five PCs (*Figure 1C*), which captured ~50.93% of all variance (see *Materials and methods* and *Appendix 1—figure 2*; *Chen et al., 2020*).

The key innovation here is the combined analysis across PSD diagnoses of core PSD symptoms and cognitive deficits, a fundamental PSD feature (*Barch et al., 2013*). The five PCs revealed few distinct boundaries between DSM categories (*Figure 1D*). Notably, for *PCs 2-5* we observed substantial overlap in PC scores between controls and patients for all DSM diagnostic groups. This overlap is not unexpected, given that behaviors measured by the BACS and PANSS (e.g. mood, cognition) occur to some extent in 'healthy' individuals not meeting DSM diagnostic criteria (*Verdoux and van Os, 2002*; *Nuevo et al., 2012*; *Kelleher and Cannon, 2011*; *Stefanis et al., 2002*). In contrast, *PC1* showed marked differentiation between PSD and CON, reflecting global functioning which was reduced across DSM diagnostic groups.

*Figure 1E* highlights loading configurations of symptom measures forming each PC. To aid interpretation, we assigned a name for each PC based on its most strongly weighted symptom measures. This naming is qualitative but informed by the pattern of loadings of the original 36 symptom measures. For example, *PC1* was highly consistent with a general impairment dimension (i.e. 'Global Functioning'); *PC2* reflected variation in cognition (i.e. 'Cognitive Functioning'); *PC3* indexed a complex configuration of psychosis-spectrum relevant items (i.e. 'Psychosis Configuration'); *PC4* captured variation in mood and anxiety related items (i.e. 'Affective Valence'); finally, *PC5* reflected variation in arousal and level of excitement (i.e. 'Agitation/Excitation'). For instance, a generally impaired patient would have a highly negative *PC1* score, which would reflect low performance on cognition and elevated scores on most other symptoms. Conversely, an individual with a high positive *PC3* score would exhibit delusional, grandiose, and/or hallucinatory behavior, whereas a person with a negative *PC3* score would exhibit motor retardation, social avoidance, and possibly a withdrawn emotional state with blunted affect (*Gelenberg, 1976*). Comprehensive loadings for all five PCs are shown in *Appendix 1—figure 1*. *Figure 1F* highlights the mean of each of the three diagnostic groups (colored spheres) and healthy controls (black sphere) projected into a three-dimensional orthogonal coordinate system for *PCs 1,2 and 3* (x,y,z axes respectively; alternative views of the three-dimensional coordinate system with all patients projected are shown in *Appendix 1—figure 1*). Critically, PC axes were not parallel with traditional aggregate symptom scales. For instance, *PC3* is angled at ~45° to the dominant direction of PANSS Positive and Negative symptom variation (purple and blue arrows respectively in *Figure 1F*).

*PC3* loads most strongly onto hallmark symptoms of PSD (including strong positive loadings onto PANSS Positive symptom and strong negative loadings onto most PANSS Negative symptoms). Therefore, we focus on *PC3* as an opportunity to quantify a fully data-driven dimension of symptom variation that is highly characteristic of the PSD patient population. Additionally, this bidirectional symptom axis captured shared variance from additional symptoms, such as PANSS General items and cognition. *PC3* provides an empirical demonstration that using a data-driven dimensionality-reduced solution (via PCA) can reveal novel symptom patterns underlying PSD psychopathology.

Notably, independent component analysis (ICA), an alternative dimensionality reduction procedure which does not enforce component orthogonality, produced similar effects for this PSD sample (see *Appendix 1 - Note 1* and *Appendix 1—figure 3A*). Certain pairs of components between the PCA and ICA solutions appear to be highly similar and directly comparable (*IC5* and *PC4*; *IC4* and *PC5*) (*Appendix 1—figure 3B*). On the other hand, *PCs 1–3* and *ICs 1–3* do not exhibit a one-to-one mapping. For example, *PC3* appears to correlate positively with *IC2* and equally strongly negatively with *IC3*, suggesting that these two ICs are oblique to the PC and perhaps reflect symptom variation that is explained by a single PC. The orthogonality of the PCA solution forces the resulting components to capture maximally separated, unique symptom variance, which in turn map robustly on to unique neural circuits. We observed that the data may be distributed in such a way that highly correlated independent components emerge in the ICA, which do not maximally separate the symptom variance associated with neural variance. We demonstrate this by plotting the relationship between

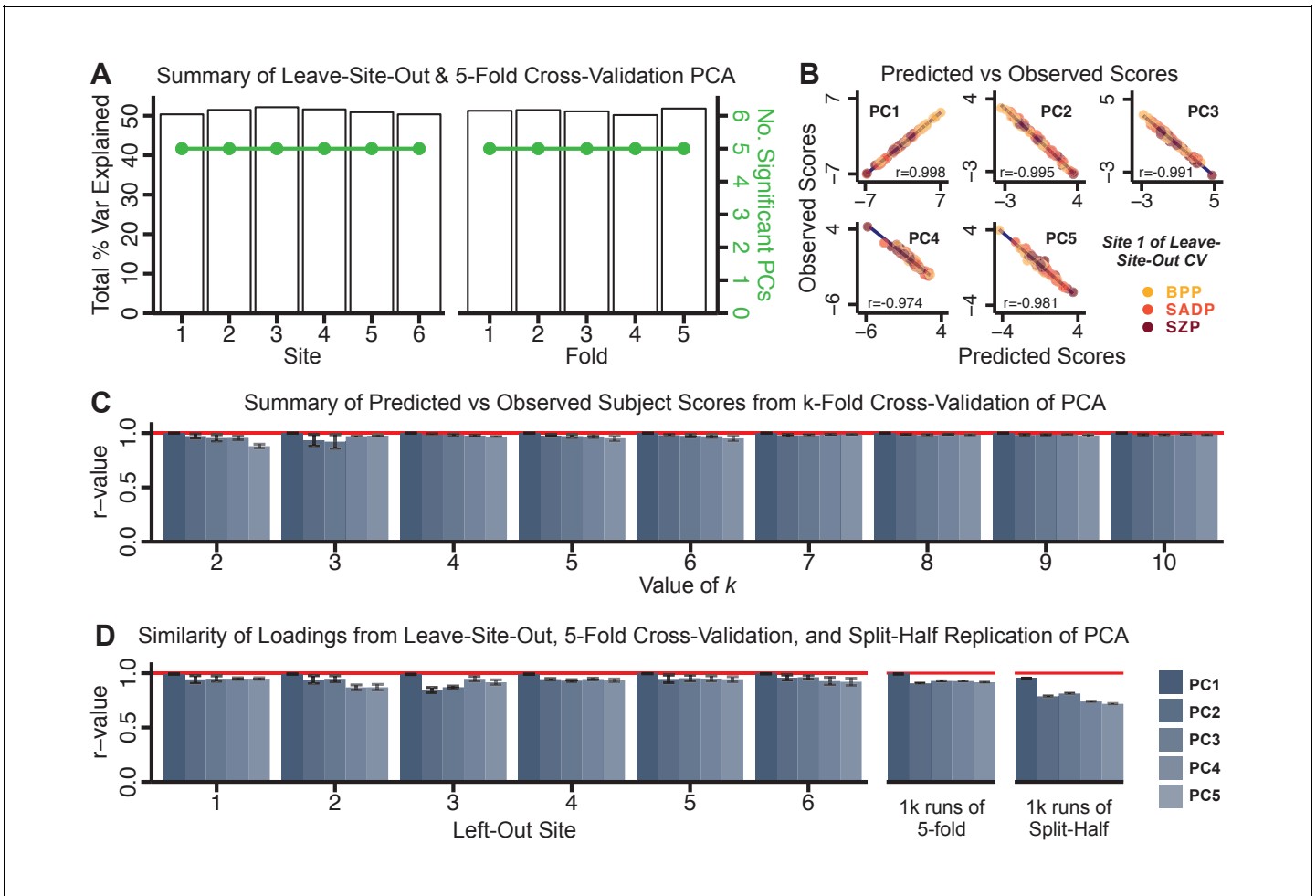

**Figure 2.** Dimensionality reduction of PSD symptom measures is highly stable and reproducible. (**A**) PCA solutions for leave-one-site out cross-validation (left) and 5-fold bootstrapping (right) explain a consistent total proportion of variance with a consistent number of significant PCs after permutation testing. Full results available in *Appendix 1—figure 4* and *Appendix 1—figure 5*. (**B**) Predicted versus observed single subject PC scores for all five PCs are shown for an example site (shown here for Site 1). (**C**) Mean correlations between predicted and observed PC scores across all patients calculated via $k$-fold bootstrapping for $k = 2–10$. For each $k$ iteration, patients were randomly split into $k$ folds. For each fold, a subset of patients was held out and PCA was performed on the remaining patients. Predicted PC scores for the held-out sample were computed from the PCA obtained from the retained samples. Original observed PC scores for the held-out sample were then correlated with the predicted PC scores derived from the retained sample. (**D**) Mean correlations between predicted and observed symptom measure loadings are shown for leave-one-site-out cross-validation (left), across 1000 runs of 5-fold cross-validation (middle), and 1000 split-half replications (right). For split-half replication, loadings were compared between the PCA performed in each independent half sample. Note: for panels C-D correlation values were averaged all $k$ runs, all six leave-site-out-runs, or all 1000 runs of the fivefold cross-validation and split-half replications. Error bars indicate the standard error of the mean.

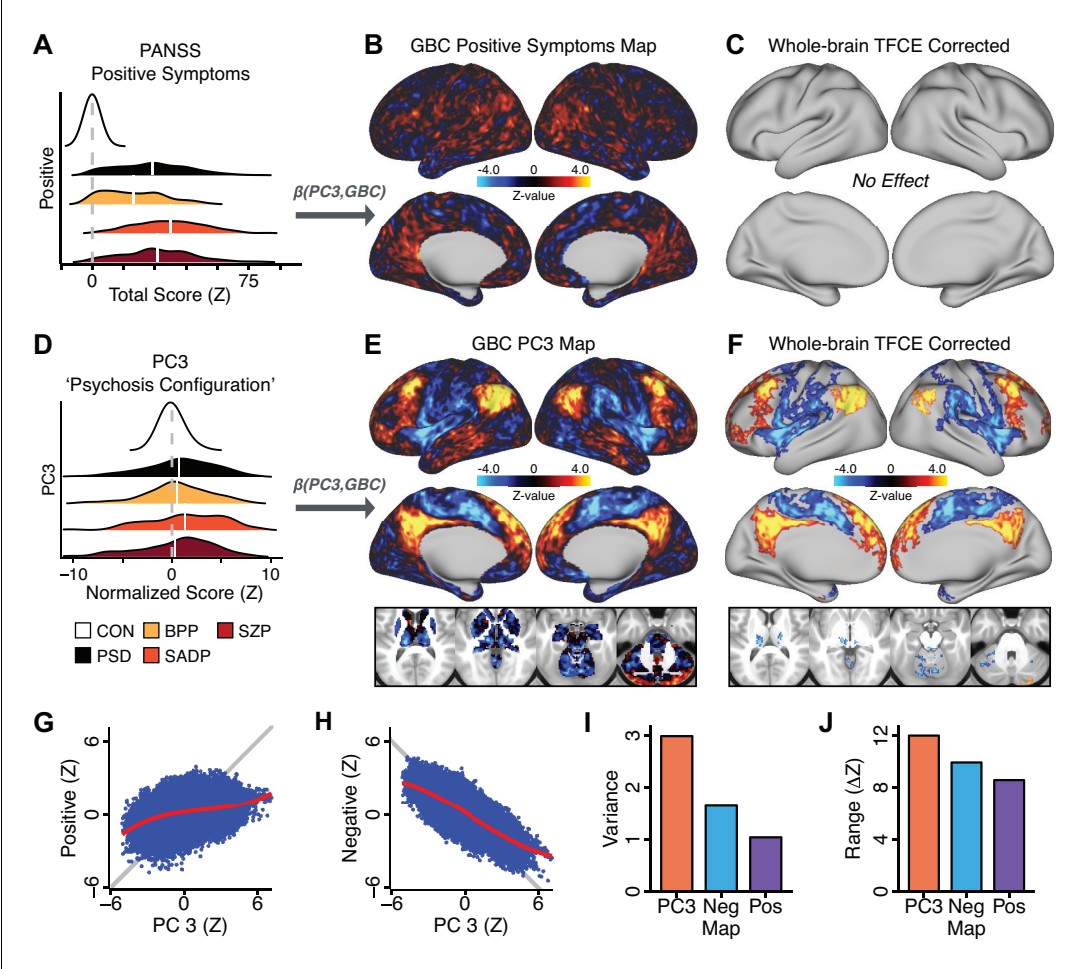

**Figure 3.** Dimensionality-reduced symptom variation reveals robust neurobehavioral mapping. (**A**) Distributions of total PANSS Positive symptoms for each of the clinical diagnostic groups normalized relative to the control group (white = CON; black = all PSD patients; yellow = BPP; orange = SADP; red = SZP). (**B**) $\beta_{Positive}GBC$ map showing the relationship between the aggregate PANSS Positive symptom score for each patient regressed onto global brain connectivity (GBC) across all patients (N = 436). (**C**) No regions survived non-parametric family-wise error (FWE) correction at p<0.05 using permutation testing with threshold-free cluster enhancement (TFCE). (**D**) Distributions of scores for *PC3 'Psychosis Configuration'* across clinical groups, again normalized to the control group. (**E**) $\beta_{PC3}GBC$ map showing the relationship between the *PC3 'Psychosis Configuration'* score for each patient regressed onto GBC across all patients (N = 436). (**F**) Regions surviving p<0.05 FWE whole-brain correction via TFCE showed clear and robust effects. (**G**) Comparison between the *Psychosis Configuration* symptom score versus the aggregate PANSS Positive symptom score GBC map for every datapoint in the neural map (i.e. greyordinate in the CIFTI map). The sigmoidal pattern indicates an improvement in the Z-statistics for the *Psychosis Configuration* symptom score map (panel E) relative to the aggregate PANSS Positive symptom map (panel B). (**H**) A similar effect was observed when comparing the *Psychosis Configuration* GBC map relative to the PANSS Negative symptoms GBC map (*Appendix 1—figure 8*). (**I**) Comparison of the variances for the *Psychosis Configuration*, PANSS Negative and PANSS Positive symptom map Z-scores. (**J**) Comparison of the ranges between the *Psychosis Configuration*, Negative and Positive symptom map Z-scores. Symptom-neural maps for all five PCs and all four traditional symptom scales (BACS and PANSS subscales) are shown in *Appendix 1—figure 8*.

parcel beta coefficients for the $\beta_{PC3}GBC$ map versus the $\beta_{IC2}GBC$ and $\beta_{IC3}GBC$ maps (*Appendix 1—figure 11G*).

Next, we show that the PCA solution was highly *stable* when tested across sites, and *k*-fold cross-validations, and was *reproducible* in independent split-half samples. First, we show that the symptom-derived PCA solution remained highly robust across all sites (*Figure 2A*) and 5-fold cross-validation iterations (see *Materials and methods*). The total proportion of variance explained as well as the total number of significant PCs remained stable (*Figure 2A*). Second, PCA loadings accurately and reliably computed the scores of subjects in a hold-out sample (*Figure 2B*). Specifically, for each left-out site, we tested if single-subject predicted scores were similar to the originally observed scores of these same subjects in the full PCA (i.e. performed with all 436 subjects). This similarity was

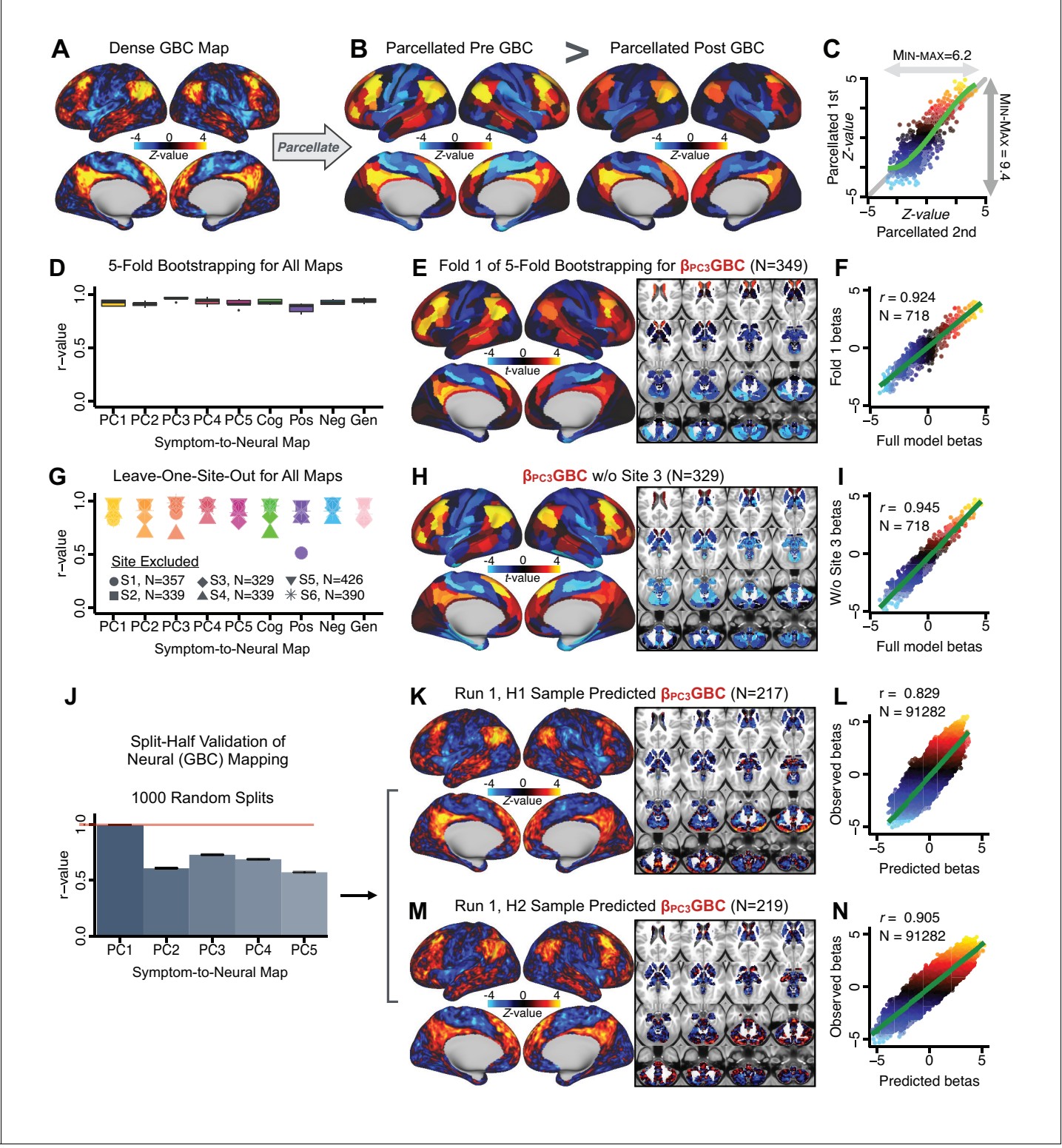

**Figure 4.** Parcellated symptom-neural GBC maps reflecting psychosis configuration are statistically robust and reproducible. (**A**) Z-scored *PC3 Psychosis Configuration* GBC neural map at the 'dense' (full CIFTI resolution) level. (**B, C**) Neural data parcellated using a whole-brain functional partition (*Ji et al., 2019b*) *before* computing subject-level GBC yielded stronger statistical values in the Z-scored *Psychosis Configuration* GBC neural map as compared to when parcellation was performed *after* computing GBC for each subject. (**D**) Summary of similarities between all symptom-neural *βGBC* maps (PCs and traditional symptom scales) across fivefold cross-validation. Boxplots show the range of *r* values between *βGBC* maps for each fold and

*Figure 4 continued on next page*

*Figure 4 continued*
the full model. (E) Normalized $\beta GBC$ map from regression of individual patients' *PC3* scores onto parcellated GBC data, shown here for a subset of patients from Fold 1 out of 5 (N = 349). The greater the magnitude of the coefficient for a parcel, the stronger the statistical relationship between GBC of that parcel and *PC3* score. (F) Correlation between the $\beta GBC$ value of each parcel in the regression model computed using patients in Fold one and the full PSD sample (N = 436) model indicates that the leave-one-fold-out $\beta GBC$ map was highly similar to the $\beta GBC$ map obtained from the full PSD sample model ($r = 0.924$). (G) Summary of leave-one-site-out regression for all symptom-neural maps. Regression of PC symptom scores onto parcellated GBC data, each time leaving out subjects from one site, resulted in highly similar maps. This highlights that the relationship between *PC3* scores and GBC is robust and not driven by a specific site. (H) $\beta GBC$ map for all PSD except one site. As an example, Site 3 is excluded here given that it recruited the most patients (and therefore may have the greatest statistical impact on the full model). (I) Correlation between the value of each parcel in the regression model computed using all patients minus Site 3, and the full PSD sample model. (J) Split-half replication of $\beta_{PC3}GBC$ map. Bar plots show the mean correlation across 1000 runs; error bars show standard error. Note that the split-half effect for *PC1* was exceptionally robust. The split-half consistency for *PC3*, while lower, was still highly robust and well above chance. (K) $\beta_{PC3}GBC$ map from *PC3*-to-GBC regression for the first half (H1) patients, shown here for one exemplar run out of 1000 split-half validations. (L) Correlation across 718 parcels between the H1 predicted coefficient map (i.e. panel K) and the observed coefficient map for H1. (M–N) The same analysis as K-L is shown for patients in H2, indicating a striking consistency in the *Psychosis Configuration* $\beta_{PC3}GBC$ map correspondence.

high for all sites (*Appendix 1—figure 4*). Also, we verified that the predicted-versus-observed PCA scores also remained similar via a *k*-fold cross-validation (for *k* = 2 to 10, *Figure 2C* and *Appendix 1—figure 3*). Finally, across all five PCs, predicted-to-observed similarity of PCA loadings was very high. Moreover, PCA loadings were stable when testing via leave-site-out analysis and 5-fold cross-validation. We furthermore demonstrated the reproducibility of the solution using 1000 independent split-half replications (*Figure 2D*). For each run of split-half replication, PSD subjects were split into two samples and a PCA was performed independently for each sample. The loadings from both PCAs were then compared using a Pearson's correlation. Importantly, results were not driven by medication status or dosage (*Appendix 1—figure 6*). Collectively, these data reduction analyses strongly support a stable and reproducible low-rank PSD symptom geometry.

## Dimensionality-reduced PSD symptom geometry reveals novel and robust neurobehavioral relationships

Next, we tested if the dimensionality-reduced symptom geometry can identify robust and novel patterns of neural variation across PSD patients. We opted to use global brain connectivity (GBC), a summary FC metric, to measure neural covariance because it yields a parsimonious measure reflecting how globally coupled an area is to the rest of the brain (*Cole et al., 2010*; see *Materials and methods*). Furthermore, we selected GBC because: (i) the metric is agnostic regarding the location of dysconnectivity as it weights each area equally; (ii) it yields an interpretable dimensionality-reduction of the full FC matrix; (iii) unlike the full FC matrix or other abstracted measures, GBC produces a neural map, which can be related to other independent neural maps (e.g. gene expression or pharmacology maps, discussed below). Furthermore, GBC has been shown to be sensitive to altered patterns of global neural connectivity in PSD cohorts (*Anticevic et al., 2013*; *Fornito et al., 2011*; *Hahamy et al., 2014*; *Cole et al., 2011*) as well as in models of psychosis (*Preller et al., 2018*; *Driesen et al., 2013*).

All five PCs captured unique patterns of GBC variation across the PSD (*Appendix 1—figure 9*), which were not observed in CON (*Appendix 1—figure 10*). Again we highlight the hallmark 'Psychosis Configuration' dimension (i.e. *PC3*), here to illustrate the benefit of the low-rank PSD symptom geometry for symptom-neural mapping relative to traditional aggregate PANSS symptom scales. The relationship between total PANSS Positive scores and GBC across N = 436 PSD patients ($\beta_{Pos}GBC$, *Figure 3A*) was statistically modest (*Figure 3B*) and no areas survived whole-brain type-I error protection (*Figure 3C*, $p < 0.05$). In contrast, regressing *PC3* scores onto GBC across N = 436 patients revealed a robust symptom-neural $\beta_{PC3}GBC$ map (*Figure 3E–F*), which survived whole-brain type-I error protection. Of note, the PC3 'Psychosis Configuration' axis is bi-directional whereby individuals who score either highly positively or negatively are symptomatic. Therefore, a high positive *PC3* score was associated with both reduced GBC across insular and superior dorsal cingulate cortices, thalamus, and anterior cerebellum and elevated GBC across precuneus, medial prefrontal, inferior parietal, superior temporal cortices, and posterior lateral cerebellum – consistent with the default-mode network (*Fox et al., 2005*). A high negative *PC3* score would exhibit the opposite

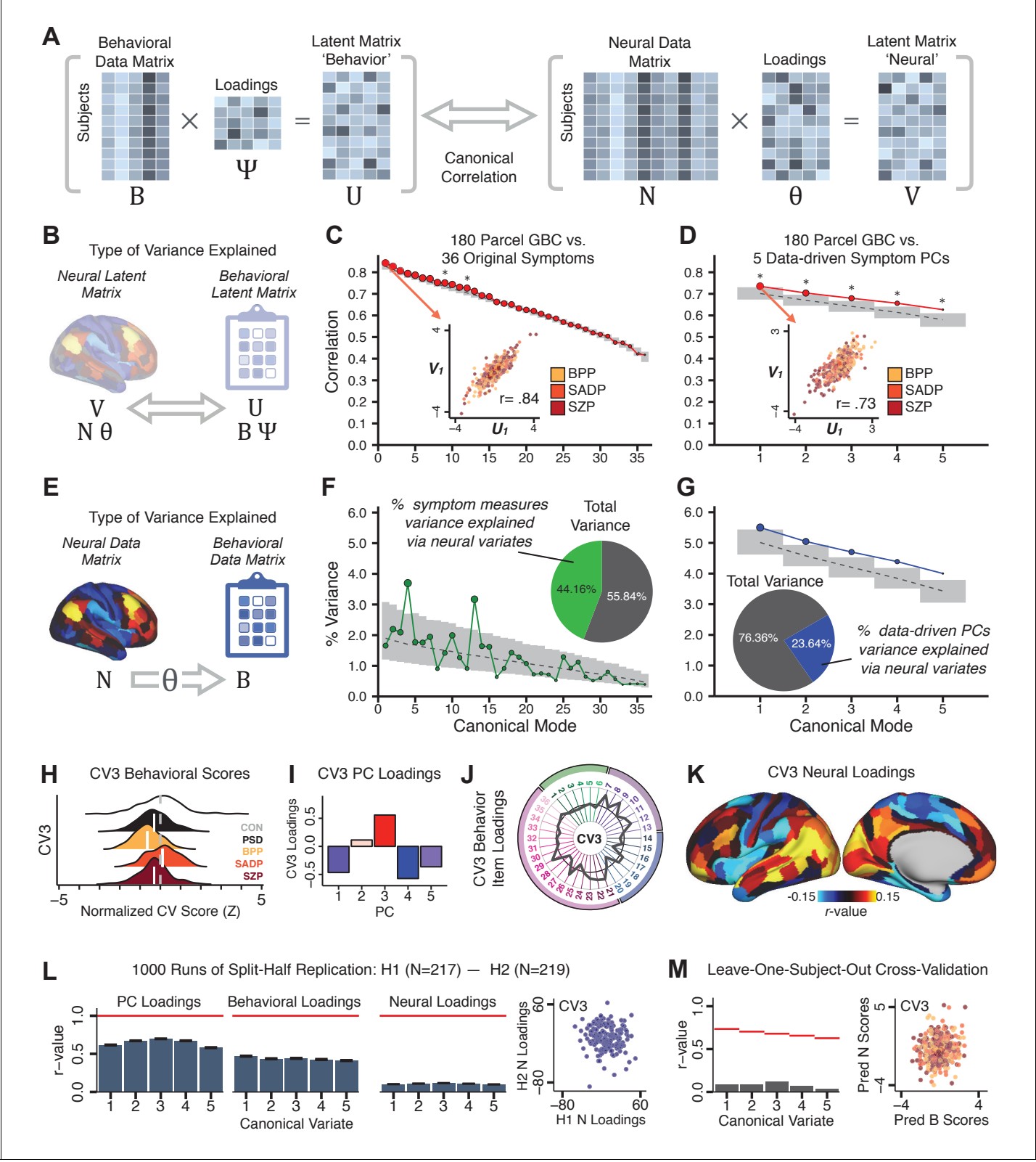

**Figure 5.** Multivariate symptom-neural feature mapping using canonical correlation analysis (CCA). (**A**) Schematic of CCA data (**B, N**), transformation (, ), and transformed 'latent' (U, V) matrices. Here, each column in *U* and *V* is referred to as a canonical variate (CV); each corresponding pair of CVs (e.g. U₁ and V₁) is referred to as a canonical mode. (**B**) CCA maximized correlations between the CVs (U and V) (**C**) Screeplot showing canonical modes obtained from 180 neural features (cortical GBC symmetrized across hemispheres) and 36 symptom measures ('180 vs. 36 CCA'). Inset illustrates the

*Figure 5 continued on next page*

*Figure 5 continued*

correlation ($r = 0.85$) between the CV of the first mode, $U_1$ and $V_1$ (note that the correlation was not driven by a separation between diagnoses). Modes 9 and 12 remained significant after FDR correction. (D) CCA computed with 180 neural features and five PC symptom features ('180 vs. 5 CCA'). Here, all modes remained significant after FDR correction. Dashed black line shows the null calculated via a permutation test with 5000 shuffles; gray bars show 95% confidence interval. (E) Correlation between $B$ and N Θ reflects how much of the symptom variation can be explained by the latent neural features. (F) Proportion of symptom variance explained by each of the neural CVs in the 180 vs. 36 CCA. Inset shows the total proportion of behavioral variance explained by the neural CVs. (G) Proportion of total symptom variance explained by each of the neural CVs in the 180 vs. 5 CCA. While CCA using symptom PCs has fewer dimensions and thus accounts for lower total variance (see inset), each neural variate explains a higher amount of symptom variance than in F, suggesting that CCA could be optimized by first obtaining a low-rank symptom solution. Dashed black line indicates the null calculated via a permutation test with 5000 shuffles; gray bars show 95% confidence interval. Neural variance explained by symptom CVs are plotted in *Appendix 1—figure 15*. (H) Distributions of *CV3* scores from the 180 vs. 5 CCA are shown here as an example of characterizing CV configurations. Scores for all diagnostic groups are normalized to CON. Additionally, (I) symptom canonical factor loadings, (J) loadings of the original 36 symptom measures, and (K) neural canonical factor loadings for *CV3* are shown. (L) Within-sample CCA cross-validation appeared robust (see *Appendix 1—figure 17*). However, a split-half replication of the 180 vs. 5 CCA (using two independent non-overlapping samples) was not reliable. Bar plots show the mean correlation for each CV between the first half (H1) and the second half (H2) CCA, each computed across 1000 runs. *Left:* split-half replication of the symptom PC loadings matrix ; *Middle:* individual symptom measure loadings; *Right:* the neural loadings matrix , which in particular was not stable. Error bars show the standard error of the mean. Scatterplot shows the correlation between *CV3* neural loadings for H1 vs. H2 for one example CCA run, illustrating lack of reliability. (M) Leave-one-subject-out cross-validation further highlights CCA instability.

pattern. Critically, this robust symptom-neural mapping emerged despite no differences between mean diagnostic group *PC3* scores (*Figure 3D*). These two diverging directions may be captured separately in the ICA solution, when orthogonality is not enforced (*Appendix 1—figure 12*). Moreover, the *PC3* symptom-neural map exhibited improved statistical properties relative to other GBC maps computed from traditional aggregate PANSS symptom scales (*Figure 3G–J*). Notably, the *PC2 – Cognitive Functioning* dimension, which captured a substantial proportion of cognitive performance-related symptom variance independent of other symptom axes, revealed a circuit that was moderately (anti-)correlated with other PC maps but strongly anti-correlated with the BACS composite cognitive deficit map ($r = -0.81$, *Appendix 1—figure 8O*). This implies that the *PC2* map reflects unique neural circuit variance that is relevant for cognition, independent of the other *PC* symptom dimensions.

## Univariate neurobehavioral map of psychosis configuration is reproducible

After observing improved symptom-neural *PC3* statistics, we tested if these neural maps replicate. Recent attempts to derive stable symptom-neural mapping using multivariate techniques, while inferentially valid, have not replicated (*Dinga et al., 2019*). This fundamentally relates to the tradeoff between the sample size needed to resolve multivariate neurobehavioral solutions and the size of the feature space. To mitigate the feature size issue we re-computed the $\beta_{PC}GBC$ maps using a functionally-derived whole-brain parcellation via the recently-validated CAB-NP atlas (*Glasser et al., 2016*; *Ji et al., 2019b*; *Materials and methods*). Here, a functional parcellation is a principled way of neural feature reduction (to 718 parcels) that can also appreciably boost signal-to-noise (*Glasser et al., 2016*; *Ji et al., 2019b*). Indeed, parcellating the full-resolution 'dense' resting-state signal for each subject prior to computing GBC statistically improved the group-level symptom-neural maps compared to parcellating after computing GBC (*Figure 4A–C*, all maps in *Appendix 1—figure 9*). Results demonstrate that the univariate symptom-neural mapping was indeed stable across fivefold bootstrapping, leave-site-out, and split-half cross-validations (*Figure 4D–N*, see *Appendix 1 - Note 2*), yielding consistent symptom-neural *PC3* maps. Importantly, the symptom-neural maps computed using ICA showed comparable results (*Appendix 1—figure 12*).

## A multivariate PSD neurobehavioral solution can be computed but is not reproducible with the present sample size

Several recent studies have reported 'latent' neurobehavioral relationships using multivariate statistics (*Xia et al., 2018*; *Drysdale et al., 2017*; *Yu et al., 2019*), which would be preferable because they simultaneously solve for maximal covariation across neural and behavioral features. Given the possibility of deriving a stable multivariate effect, here we tested if results improve with canonical correlation analysis (CCA) (*Hardoon et al., 2004*; *Figure 5A*).

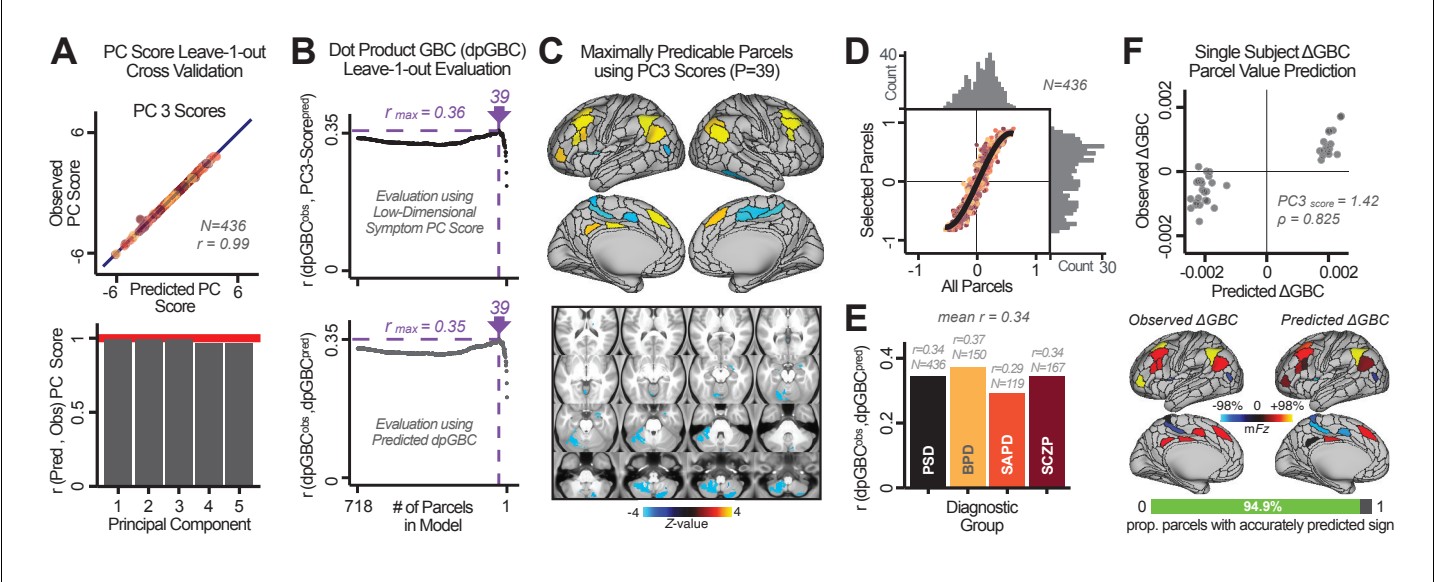

**Figure 6.** Optimizing neural feature selection to inform single-subject prediction via a low-dimensional symptom solution. (A) Leave-one-out cross-validation for the symptom PCA analyses indicates robust individual score prediction. Top panel: Scatterplot shows the correlation between each subject's predicted *PC3* score from a leave-one-out PCA model and their observed *PC3* score from the full-sample PCA model, *r* = 0.99. Bottom panel: Correlation between predicted and observed individual PC scores was above 0.99 for each of the significant PCs (see *Figure 1*). The red line indicates *r* = 1. (B) We developed a univariate step-down feature selection framework to obtain the most predictive parcels using a subject-specific approach via the *dpGBC* index. Specifically, the 'observed' patient-specific $dpGBC^{obs}$ was calculated using each patient's $\Delta GBC^{obs}$ (i.e. the patient-specific GBC map vs. the group mean GBC for each each parcel) and the 'reference' symptom-to-GBC *PC3* map (described in *Figure 4B*) [$dpGBC^{obs} = \Delta GBC^{obs} \cdot \beta_{PC3}GBC^{obs}$]. See *Materials and methods* and *Appendix 1—figure 24* for complete feature selection procedure details. In turn, we computed the predicted *dpGBC* index for each patient by holding their data out of the model and predicting their score ($dpGBC^{pred}$). We used two metrics to evaluate the maximally predictive feature subset: (i) The correlation between *PC3* symptom score and $dpGBC^{obs}$ across all N = 436, which was maximal for *P* = 39 parcels [*r* = 0.36, purple arrow]; (ii) The correlation between $dpGBC^{obs}$ and $dpGBC^{pred}$, which also peaked at *P* = 39 parcels [*r* = 0.35, purple arrow]. (C) The *P* = 39 maximally predictive parcels from the $\beta_{PC3}GBC^{obs}$ map are highlighted (referred to as the 'selected' map). (D) Across all n = 436 patients we evaluated if the selected parcels improve the statistical range of similarities between the $\Delta GBC^{obs}$ and the $\beta_{PC3}GBC^{obs}$ reference for each patient. For each subject the value on the X-axis reflects a correlation between their $\Delta GBC^{obs}$ map and the $\beta_{PC3}GBC^{obs}$ map across all 718 parcels; the Y-axis reflects a correlation between their $\Delta GBC^{obs}$ map and the $\beta_{PC3}GBC^{obs}$ map only within the 'selected' 39 parcels. The marginal histograms show the distribution of these values across subjects. (E) Each DSM diagnostic group showed comparable correlations between predicted and observed *dpGBC* values. The *r*-value shown for each group is a correlation between the $dpGBC^{obs}$ and $dpGBC^{pred}$ vectors, each of length N. (F) Scatterplot for a single patient with a positive behavioral loading (*PC3* score = 1.42) and also with a high correlation between predicted $\Delta GBC^{pred}$ versus observed $\Delta GBC^{obs}$ values for the 'selected' 39 parcels ($\rho = 0.825$). Right panel highlights the observed vs. predicted $\Delta GBC$ map for this patient, indicating that 94.9% of the parcels were predicted in the correct direction (i.e. in the correct quadrant).

To evaluate if the number of neural features affects the solution, we computed CCA using GBC from: (i) 180 symmetrized cortical parcels; (ii) 359 bilateral subcortex-only parcels; (iii) 192 symmetrized subcortical parcels; (iv) 12 functional networks, all from the brain-wide cortico-subcortical CAB-NP parcellation (*Glasser et al., 2016*; *Ji et al., 2019b*). We did not compute a solution using 718 bilateral whole-brain parcels, as this exceeded the number of samples and rendered the CCA insolvable. Notably, the 359 subcortex-only solution did not produce stable results according to any criterion, whereas the 192 symmetrized subcortical features (*Appendix 1—figure 13*) and 12 network-level features (*Appendix 1—figure 14*) solutions captured statistically modest effects relative to the 180 symmetrized cortical features (*Figure 5B–D*). Therefore, we characterized the 180-parcel solution further. We examined two CCA solutions using these 180 neural features in relation to symptom scores: (i) all 36 item-level symptom measure scores from the PANSS/BACS ('180 vs. 36 CCA', *Figure 5C,F*); and (ii) five PC symptom scores ('180 vs. 5 CCA', *Figure 5D,G*, see *Materials and methods*). Only 2 out of 36 modes for the 180 vs. 36 CCA solution survived permutation testing and false discovery rate (FDR) correction (*Figure 5C*). In contrast, all 5 modes of the 180 vs. 5 CCA survived (*Figure 5D*). Critically, we found that no single CCA mode computed on item-level symptom measures captured more variance than the CCA modes derived from PC symptom

scores, suggesting that the PCA-derived dimensions capture more neurally relevant variation than any one single clinical item (*Figure 5E–G*). Additional CCA details are presented in *Appendix 1 - Note 3* and *Appendix 1—figure 15*.

We highlight here an example canonical variate, *CV3*, across both symptom and neural effects, with data for all *CVs* shown in *Appendix 1—figure 16*. *CV3* scores across diagnostic groups normalized to controls are shown *Figure 5H* in addition to how *CV3* loads onto each *PC* (*Figure 5I*). The negative loadings on to *PCs 1, 4, and 5* and the high positive loadings on to *PC3* in *Figure 5I* indicate that *CV3* captures some shared variation across symptom *PCs*. This can also be visualized by computing how *CV3* projects onto the original 36 symptom measures (*Figure 5J*). Finally, the neural loadings for *CV3* are shown in *Figure 5K*.

Lastly, we tested if the 180 vs. 5 CCA solution is stable and reproducible, as done with PC-to-GBC univariate results. The CCA solution was robust when tested with *k*-fold and leave-site-out cross-validation (*Appendix 1—figure 17*) likely because these methods use CCA loadings derived from the full sample. However, the CCA loadings did not replicate in non-overlapping split-half samples (*Figure 5L*, see *Appendix 1 - Note 4*). Moreover, a leave-one-subject-out cross-validation revealed that removing a single subject from the sample affected the CCA solution such that it did not generalize to the left-out subject (*Figure 5M*). This is in contrast to the PCA-to-GBC univariate mapping, which was substantially more reproducible for all attempted cross-validations relative to the CCA approach. This is likely because substantially more power is needed to resolve a stable multivariate neurobehavioral effect with this many features (*Dinga et al., 2019*). Indeed, a multivariate power analysis using 180 neural features and five symptom features and assuming a true canonical correlation of $r = 0.3$ suggests that a minimal sample size of $N = 8,145$ is needed to sufficiently detect the effect (*Helmer et al., 2020*), which is an order of magnitude greater than the available sample size (*Appendix 1—figure 18*). Therefore, we leverage the univariate symptom-neural result for subsequent subject-specific model optimization and comparisons to molecular neuroimaging maps.

## A major proportion of overall neural variance may not be relevant for psychosis symptoms

Most studies look for differences between clinical and control groups, but to our knowledge no study has tested whether both PSD and healthy controls actually share a major portion of neural variance that may be present across all people. If the bulk of the neural variance is similar across both PSD and CON groups then including this clinically irrelevant neural signal might obscure clinical-relevant neurobehavioral relationships. To test this, we examined the shared variance structure of the neural signal for all PSD patients (N = 436) and all controls (N = 202) independently by conducting a PCA on the GBC maps (see *Materials and methods*). Patients' and controls' neural signals were highly similar for each of the first three neural PCs (>30% of all neural variance in each group) (*Appendix 1—figure 19A–J*). These neural *PCs* may reflect a 'core' shared symptom-irrelevant neural variance that generalizes across all people. These data suggest that the bulk of neural variance in PSD patients may actually not be symptom-relevant, which highlights the importance of optimizing symptom-neural mapping. Under this assumption, removing the shared neural variance between PSD and CON should not drastically affect the reported symptom-neural univariate mapping solution, because this common variance *does not map to clinical features*. Thus, we conducted a PCA using the parcellated GBC data from all 436 PSD and 202 CON (see *Materials and methods*), which we will refer to as 'GBC-PCA' to avoid confusion with the symptom/behavioral PCA. This GBC-PCA resulted in 637 independent GBC-PCs. Since PCs are orthogonal to each other, we then partialled out the variance attributable to GBC-PC1 from the PSD data by reconstructing the PSD GBC matrix using only scores and coefficients from the remaining 636 GBC-PCs ($\hat{GBC}_{woPC1}$). We then reran the univariate regression as described in *Figure 3*, using the same five symptom PC scores across 436 PSD (*Appendix 1—figure 20*). Removing the first PC of shared neural variance (which accounted for about 15.8% of the total GBC variance across CON and PSD) from PSD data attenuated the statistics slightly (not unexpected as the variance was by definition reduced) but otherwise did not strongly affect the univariate mapping solution. We repeated the symptom-neural regression next with the first two GBC-PCs partialled out of the PSD data *Appendix 1—figure 21*, with the first three PCs parsed out *Appendix 1—figure 22*, and with the first four neural PCs parsed out *Appendix 1—*

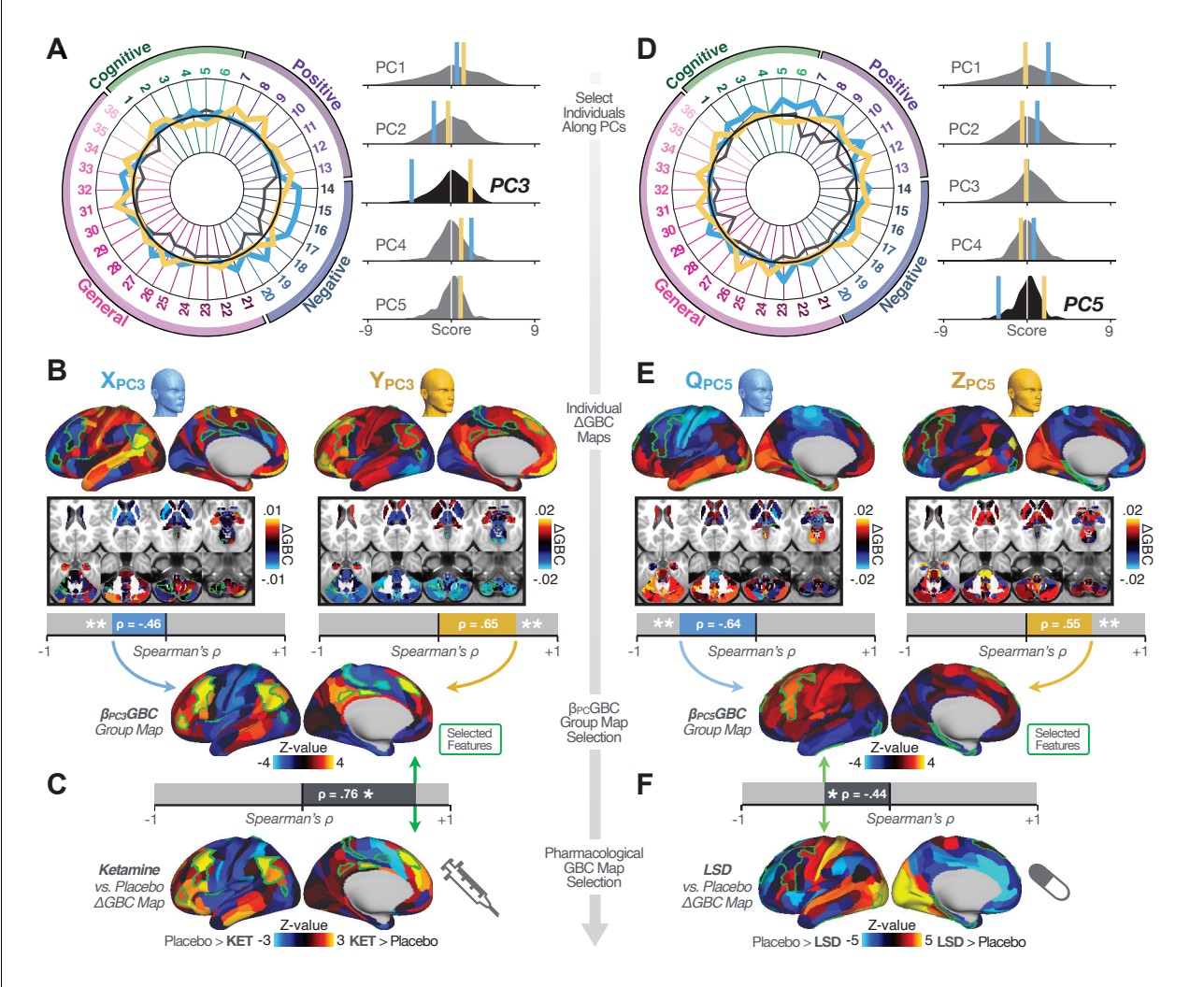

**Figure 7.** Leveraging subject-specific brain-behavioral maps for molecular neuroimaging target selection. (**A**) Data for two individual patients from the replication dataset are highlighted for PC3: $X_{PC3}$ (blue) and $Y_{PC3}$ (yellow). Both of these patients scored above the neural and behavioral thresholds for PC3 defined in the 'discovery' PSD dataset. Patient $X_{PC3}$ loads highly negatively on the PC5 axis and Patient $Y_{PC3}$ loads highly positively. Density plots show the projected PC scores for Patients $X_{PC3}$ and $Y_{PC3}$ overlaid on distributions of PC scores from the discovery PSD sample. (**B**) Neural maps show cortical and subcortical $\Delta GBC^{obs}$ for the two patients $X_{PC3}$ and $Y_{PC3}$ specifically reflecting a difference from the mean PC3. The similarity of $\Delta GBC^{obs}$ and the $\beta_{PC3}GBC^{obs}$ map within the most predictive neural parcels for PC3 (outlined in green). Note that the sign of neural similarity to the reference PC3 map and the sign of the PC3 score is consistent for these two patients. (**C**) The selected PC3 map (parcels outlined in green) is spatially correlated to the neural map reflecting the change in GBC after ketamine administration ($\rho = 0.76$, **Materials and methods**). Note that Patient $X_{PC3}$ who exhibits $\Delta GBC^{obs}$ that is anti-correlated to the ketamine map also expresses depressive moods symptoms (panel A). This is consistent with the possibility that this person may clinically benefit from ketamine administration, which may elevate connectivity in areas where they show reductions (**Berman et al., 2000**). In contrast, Patient $Y_{PC3}$ may exhibit an exacerbation of their psychosis symptoms given that their $\Delta GBC^{obs}$ is positively correlation with the ketamine map. (**D**) Data for two individual patients from the discovery dataset are highlighted for PC5: $Q_{PC5}$ (blue) and $Z_{PC5}$ (yellow). Note that no patients in the replication dataset were selected for PC5 so both of these patients were selected from 'discovery' PSD dataset for illustrative purposes. Patient $Q_{PC5}$ loads highly negatively on the PC5 axis and Patient $Z_{PC5}$ loads highly positively. Density plots show the projected PC scores for Patients $Q_{PC5}$ and $Z_{PC5}$ overlaid on distributions of PC scores from the discovery PSD sample. (**E**) Neural maps show cortical and subcortical $\Delta GBC^{obs}$ for Patients $Q_{PC5}$ and $Z_{PC5}$, which are highly negatively and positively correlated with the selected PC5 map respectively. (**F**) The selected PC5 map (parcels outlined in green) is spatially anti-correlated with the LSD response map ($\rho = -0.44$, see **Materials and methods**), suggesting that circuits modulated by LSD (i.e. serotonin, in particular 5-HT2A) may be relevant for the PC5 symptom expression. Here, a serotonin receptor agonist may modulate the symptom-neural profile of Patient $Q_{PC5}$, whereas an antagonist may be effective for Patient $Z_{PC5}$.

*figure 23*. The symptom-neural maps remain fairly robust, although the similarity with the original $\beta_{PC}GBC$ maps does drop as more common neural variance is parsed out.

## Optimizing symptom-neural mapping features for personalized prediction via dimensionality-reduced symptoms

Above we demonstrated that symptom *PC* scores can be reliably computed across sites and cross-validation approaches (*Figure 2*). Next, we show that leave-one-subject-out cross-validation yields reliable effects for the low-rank symptom PCA solution (*Figure 6A*). This stable single-subject PC score prediction provides the basis for testing if the derived neural maps can yield an individually reliable subset of features. To this end, we developed a framework for univariate neural feature selection (*Appendix 1—figure 23*) based on PC scores (e.g. *PC3* score). Specifically, we computed a dot product GBC metric (*dpGBC*) that provides an index of similarity between an individual $\Delta GBC$ topography relative to a 'reference' group-level $\beta_{PC}GBC$ map (see **Materials and methods** and *Appendix 1—figure 23*). Using this *dpGBC* index via a feature selection step-down regression, we found a subset of parcels for which the symptom-neural statistical association was maximal (*Figure 6A*). For *PC3*, we found $P = 39$ maximally predictive parcels out of the group neural $\beta_{PC}GBC$ map. Specifically, the relationship between *PC3* symptom scores and *dpGBC* values across subjects was maximal (*Figure 6B*, top panel, $r = 0.36$) as was the relationship between predicted $dpGBC^{pred}$ vs. observed $dpGBC^{obs}$ (*Figure 6B*, bottom panel, $r = 0.35$) (see *Appendix 1—figure 25* for all PCs). Importantly, the 'subset' feature map (i.e. $[\beta_{PC3}GBC^{obs}]^{P=39}$, *Figure 6C*) exhibited improved statistical properties relative to the full map (i.e. $[\beta_{PC3}GBC^{obs}]^{P=718}$, *Figure 6D*). Furthermore, the relationship between observed vs. predicted subset feature maps (i.e. $r[dpGBC^{obs}, dpGBC^{pred}]$) was highly consistent across DSM diagnoses (*Figure 6E*). Finally, a single patient is highlighted for whom the correlation between predicted and observed subset feature maps was high (i.e. $r[\Delta GBC^{pred}, \Delta GBC^{obs}]$, *Figure 6F*), demonstrating that the dimensionality-reduced symptom scores can be used to quantitatively optimize symptom-neural map features for individuals.

## Single patient evaluation via neurobehavioral target map similarity

We showed a quantitative framework for computing a neurobehavioral model at the single-patient level. This brain-behavior space (BBS) model was optimized along a single dimensionality-reduced symptom PC axis. Next, we tested a hybrid patient selection strategy by first imposing a PC-based symptom threshold, followed by a target neural similarity threshold driven by the most highly predictive symptom-neural map features. This is described in **Materials and methods**, **Appendix 1 - Note 5**, and *Appendix 1—figure 26–27*. We found that for patients with a high (either positive or negative) *PC* symptom score the symptom-neural relationship was robust (*Appendix 1—figure 26A-D*), and these patients could be reliably selected for that particular dimension (*Appendix 1—figure 26E,G*). Conversely, patients with a low absolute *PC* score showed a weak relationship with symptom-relevant neural features. This is intuitive because these individuals do not vary along the selected PC symptom axis. We also tested this patient selection strategy on an independent cross-diagnostic 'replication' sample, which yielded consistent results (*Appendix 1—figure 26F,H*). Collectively, these results show that data-driven symptom scores can pinpoint individual patients for whom neural variation strongly maps onto a target neural reference map. These data also highlight that both symptom and neural information for an independent patient can be quantified in the reference 'discovery' BBS using their symptom data alone.

## Subject-specific PSD neurobehavioral features track neuropharmacological map patterns

Next, we use the personalized BBS selection in a proof-of-concept framework for informing molecular mechanism of possible treatment response by relating subject-specific BBS to independently acquired pharmacological neuroimaging maps. Here, we examine two mechanisms implicated in PSD neuropathology via ketamine, a N-methyl-D-aspartate (NMDA) receptor antagonist (*Krystal et al., 1994*), and lysergic acid diethylamide (LSD), primarily a serotonin receptor agonist (*Preller et al., 2018*; *González-Maeso et al., 2007*; *Egan et al., 1998*). We first quantified individual subjects' BBS 'locations' in the established reference neurobehavioral geometry. The radarplot in *Figure 7A* shows original symptoms whereas *Figure 7B* shows $\Delta GBC^{obs}$ maps for two

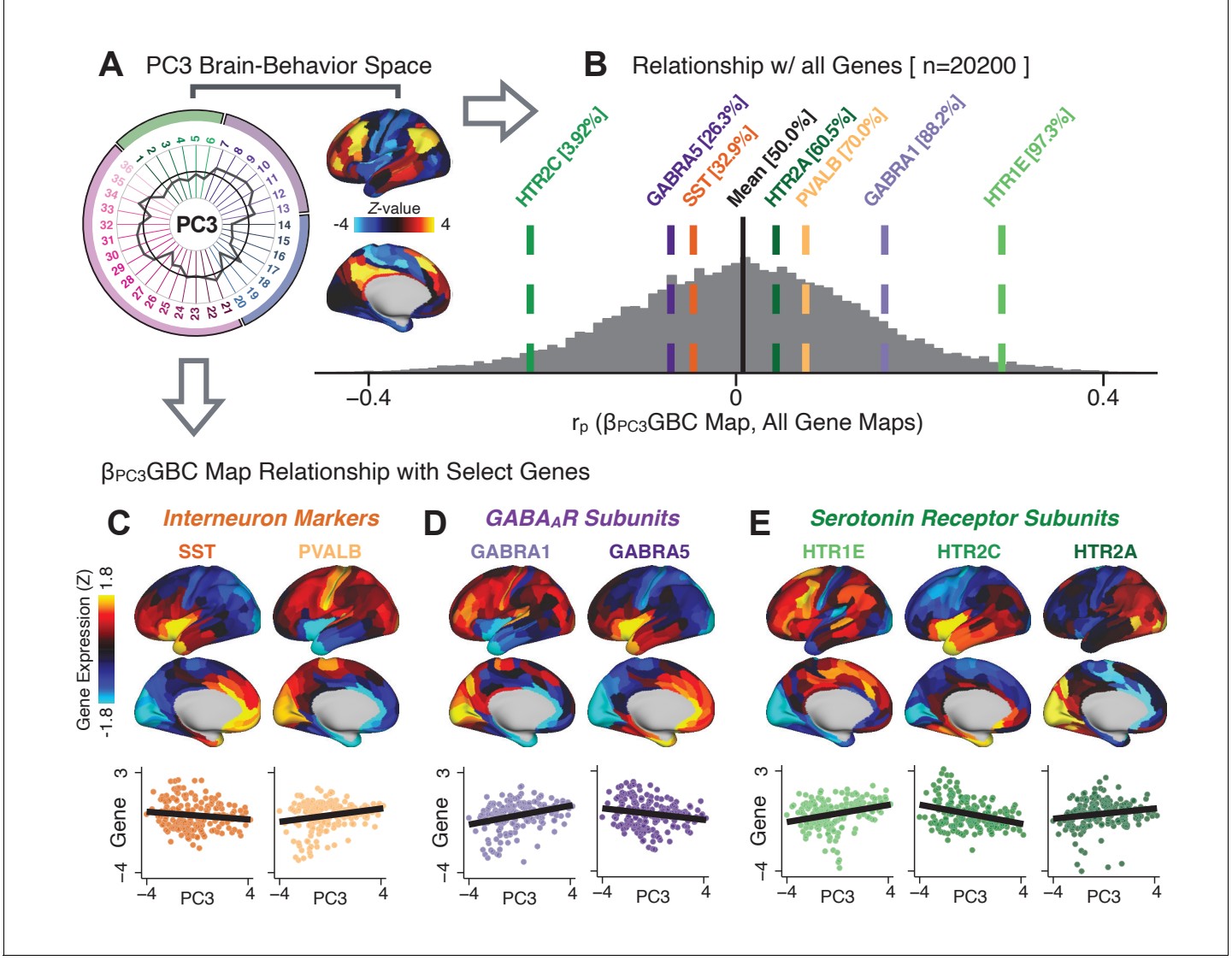

**Figure 8.** Psychosis spectrum symptom-neural maps track neural gene expression patterns computed from the Allen Human Brain Atlas (AHBA). (A) The symptom loadings and the associated neural map jointly reflect the *PC3* brain-behavioral space (BBS) profile, which can be quantitatively related to human cortical gene expression patterns obtained from the AHBA (*Burt et al., 2018*). (B) Distribution of correlation values between the *PC3* BBS map and ~20,000 gene expression maps derived from the AHBA dataset. Specifically, AHBA gene expression maps were obtained using DNA microarrays from six postmortem brains, capturing gene expression topography across cortical areas. These expression patterns were then mapped onto the cortical surface models derived from the AHBA subjects' anatomical scans and aligned with the Human Connectome Project (HCP) atlas, described in prior work and methods (*Burt et al., 2018*). Note that because no significant inter-hemispheric differences were found in cortical gene expression all results were symmetrized to the left hemisphere, resulting in 180 parcels. We focused on a select number of psychosis-relevant genes – namely genes coding for the serotonin and GABA receptor subunits and interneuron markers. Seven genes of interest are highlighted with dashed lines. Note that the expression pattern of HTR2C (green dashed line) is at the low negative tail of the entire distribution, that is highly anti-correlated with *PC3* BBS map. Conversely, GABRA1 and HTR1E are on the far positive end, reflecting a highly similar gene-to-BBS spatial pattern. (C) Upper panels show gene expression patterns for two interneuron marker genes, somatostatin (SST) and parvalbumin (PVALB). Positive (yellow) regions show areas where the gene of interest is highly expressed, whereas negative (blue) regions indicate low expression values. Lower panels highlight all gene-to-BBS map spatial correlations where each value is a symmetrized cortical parcel (180 in total) from the HCP atlas parcellation. (D) Gene expression maps and spatial correlations with the *PC3* BBS map for two GABA$_A$ receptor subunit genes: GABRA1 and GABRA5. (E) Gene expression maps and spatial correlations with the *PC3* BBS map for three serotonin receptor subunit genes: HTR1E, HTR2C, and HTR2A.

patients from the replication dataset (denoted here with $X_{PC3}$ and $Y_{PC3}$, see *Appendix 1—figure 11* for other example patients). Both of these patients exceeded the neural and behavioral BBS selection indices for *PC3* (defined independently in the 'discovery' dataset, *Appendix 1—figure 26C*).

Furthermore, both patients exhibited neurobehavioral variation in line with their expected locations in the BBS geometry. Specifically, patient $X_{PC3}$ from the replication dataset scored highly negatively on the *PC3* axis defined in the 'discovery' PSD sample (*Figure 7A*). In contrast, patient $Y_{PC3}$ scored positively on the *PC3* axis. Importantly, the correlation between the $\Delta GBC^{obs}$ map for each patient and the group-reference $\beta_{PC3}GBC$ was directionally consistent with their symptom PC score (*Figure 7B–C*).

We then tested if the single-subject BBS selection could be quantified with respect to a neural map reflecting glutamate receptor manipulation, a hypothesized mechanism underlying PSD symptoms (*Moghaddam and Javitt, 2012*). Specifically, we used an independently collected ketamine infusion dataset, collected in healthy adult volunteers during resting-state fMRI (*Anticevic et al., 2012a*). As with the clinical data, here we computed a $\Delta GBC$ map reflecting the effect of ketamine on GBC relative to placebo (*Materials and methods*). The maximally predictive *PC3* parcels exhibited high spatial similarity with the ketamine map (ρ = 0.76, see *Materials and methods*), indicating that the $\Delta GBC$ pattern induced by ketamine tracks with the GBC pattern reflecting *PC3* symptom variation.

Critically, because $X_{PC3}$ is negatively loaded on the *PC3* symptom axis, an NMDA receptor antagonist like ketamine may modulate symptom-relevant circuits in a way that reduces similarity with the *PC3* map. This may in turn have an impact on the *PC3*-like symptoms. Consistent with this hypothesis, $X_{PC3}$ expresses predominantly depressive symptoms (*Figure 7A*), and ketamine has been shown to act as an anti-depressant (*Berman et al., 2000*). This approach can be applied for patients that load along another axis, such as *PC5*. *Figure 7D–E* shows the symptom and neural data for two patients whom met thresholds for *PC5* selection (*Appendix 1—figure 26C*). Notably, the selected *PC5* map is anti-correlated with a $\Delta GBC$ map reflecting LSD vs. placebo effects (*Preller et al., 2018*) (ρ = −0.44, *Figure 7F*). Hence areas modulated by LSD may map onto behavioral variation along *PC5*. Consequently, serotonergic modulation may be effective for treating $Q_{PC5}$ and $Z_{PC5}$, via an antagonist or an agonist respectively. These differential similarities between pharmacological response maps and BBS maps (*Appendix 1—figure 29*) can be refined for quantitative patient segmentation.

## Group-level PSD neurobehavioral features track neural gene expression patterns

To further inform molecular mechanism for the derived BBS results, we compared results with neural gene expression profiles derived from the Allen Human Brain Atlas (AHBA) (*Hawrylycz et al., 2012*; *Burt et al., 2018*; *Figure 8A*, *Materials and methods*). Specifically, we tested if BBS cortical topographies, which reflect stable symptom-neural mapping along PSD, covary with the expression of genes implicated in PSD neuropathology. We focus here on serotonin receptor subunits (HTR1E, HTR2C, HTR2A), GABA receptor subunits (GABRA1, GABRA5), and the interneuron markers somatostatin (SST) and parvalbumin (PVALB). Serotonin agonists such as LSD have been shown to induce PSD-like symptoms in healthy adults (*Preller et al., 2018*) and the serotonin antagonism of 'second-generation' antipsychotics are thought to contribute to their efficacy in targeting broad PSD symptoms (*Geyer and Vollenweider, 2008*; *Meltzer et al., 2012*; *Abi-Dargham et al., 1997*). Abnormalities in GABAergic interneurons, which provide inhibitory control in neural circuits, may contribute to cognitive deficits in PSD (*Benes and Berretta, 2001*; *Inan et al., 2013*; *Dienel and Lewis, 2019*) and additionally lead to downstream excitatory dysfunction that underlies other PSD symptoms (*Lisman et al., 2008*; *Grace, 2016*). In particular, a loss of prefrontal parvalbumin-expression fast-spiking interneurons has been implicated in PSD (*Enwright Iii et al., 2018*; *Lodge et al., 2009*; *Beasley and Reynolds, 1997*; *Lewis et al., 2012*). *Figure 8B* shows the distribution of correlations between the *PC3* map and the cortical expression patterns of 20,200 available AHBA genes (results for other PCs are shown in *Appendix 1—figure 30*). Seven genes of interest are highlighted, along with their cortical expression topographies and their similarity with the *PC3* BBS map (*Figure 8C–E*). This BBS-to-gene mapping can potentially reveal novel therapeutic molecular targets for neurobehavioral variation. For example, the HTR1E gene, which encodes the serotonin 5-HT$_{1E}$ receptor, is highly correlated with the *PC3* BBS map. This could drive further development of novel selective ligands for this receptor, which are not currently available (*Kitson, 2007*).

## Discussion

We found a robust and reproducible symptom-neural mapping across the psychosis spectrum that emerged from a low-dimensional symptom solution. Critically, this low-rank symptom solution was predictive of a neural circuit pattern, which reproduced at the single-subject level. In turn, we show that the derived PSD symptom-neural feature maps exhibit spatial correspondence with independent pharmacological and gene expression neural maps that are directly relevant for PSD neurobiology. We demonstrate, for the first time in the literature, a transdiagnostic data-reduced PSD symptom geometry across hallmark psychopathology symptoms and cognitive measures, that shows stable and robust relationships with neural connectivity with singles subject-level precision. Critically, we anchor these symptom-neural maps to mechanistically informed molecular imaging benchmarks that can be linked to relevant pathophysiology, and furthermore demonstrate how these methods can be combined to inform personalized patient selection decisions.

### Deriving an individually predictive low-dimensional symptom representation across the psychosis spectrum

Psychosis spectrum is associated with notable clinical heterogeneity such deficits in cognition as well as altered beliefs (i.e. delusions), perception (i.e. hallucinations), and affect (i.e. negative symptoms) (*Lefort-Besnard et al., 2018*). This heterogeneity is captured by clinical instruments that quantify PSD symptoms across dozens of specific questions and ratings. This yields a high-dimensional symptom space that is intractable for reliable mapping of neural circuits (*Helmer et al., 2020*). Here we show that a low-rank solution captures principal axes of PSD symptom variation, a finding in line with prior work in SZP (*van der Gaag et al., 2006a*; *van der Gaag et al., 2006b*; *Lindenmayer et al., 1994*; *Emsley et al., 2003*; *White et al., 1997*; *Dollfus et al., 1996*; *Blanchard and Cohen, 2006*; *Chen et al., 2020*; *Lefort-Besnard et al., 2018*).

These results highlights two key observations: (i) Existing symptom reduction studies (even those in SZP specifically) have not evaluated solutions that include cognitive impairment – a hallmark deficit across the psychosis spectrum (*Barch et al., 2013*). Here we show that cognitive performance captures a notable portion of the symptom variance independent of other axes. We observed that cognitive variation captured 10% of PSD sample variance even after accounting for 'Global Functioning' psychopathology. (ii) No study has quantified cognitive deficit variation along with core psychosis symptoms via dimensionality reduction across multiple PSD diagnoses. While existing studies have evaluated stability of data-reduced solutions within a single DSM category (*van der Gaag et al., 2006a*; *Lindenmayer et al., 1995*; *Lefort-Besnard et al., 2018*), current results show that dimensionality-reduced PSD symptom solutions can be reproducibly obtained across DSM diagnoses.

Importantly, on each data-reduced symptom axis, some subset of PSD patients received a score near zero. This does not imply that these patients were unimpaired; rather, the symptom configurations for these patients were orthogonal to variation along this specific axis. While in the current paper we demonstrate individual selection based on symptom scores along a single axis, it may also be possible to use a combination of several symptom dimensions to further inform individual treatment selection. For example, for patients with above-threshold symptom scores along several dimensions, a combination of these *PCs* may be more predictive of their neural features and in turn be a better indication of treatment selection for these individuals. Alternatively, the symptom dimensions can be used as exclusionary criteria for selectively picking out individuals along only one particular axis (e.g. filtering out patients above-threshold on *PCs 1, 3, 4,* and five to selectively study patients with only severe *PC2* cognitive deficits). Additionally, the ICA solution revealed oblique axes which are linear combinations of the *PCs*, for example *IC2* which is oblique to *PCs 1, 2,* and *3* (*Appendix 1—figure 3*). However, these *ICs* do not result in as robust or unique a neural circuit mapping as do the *PCs*, as shown in *Appendix 1—figure 12*. The observation that PSD are associated with multiple complex symptom dimensions highlights an important intuition that may extend to other mental health spectra. Additionally, the PSD symptom axes reported here are neither definitive nor exhaustive. In fact, close to 50% of all clinical variance was not captured by the symptom PCA – an observation often overlooked in symptom data-reduction studies, which focus on attaining 'predictive accuracy'. Such studies rarely consider how much variance remains unexplained in the final data-reduced model and, relatedly, if the proportion of explained variance is reproducible

across samples. This is a key property for reliable symptom-to-neural mapping. Thus, we tested if this reproducible low-dimensional PSD symptom space robustly mapped onto neural circuit patterns.

## Leveraging a robust low-dimensional symptom representation for mapping brain-behavior relationships

We show that the dimensionality-reduced symptom space improved the mapping onto neural circuit features (i.e. GBC), as compared to a priori item-level clinical symptom measures (*Figure 3*). This symptom-neural mapping was highly reproducible across various cross-validation procedures and split-half replication (*Figure 4*). The observed statistical symptom-neural improvement after dimensionality reduction suggests that data-driven clinical variation more robustly covaried with neural features. As noted, the low-rank symptom axes generalized across DSM diagnoses. Consequently, the mapping onto neural features (i.e. GBC) may have been restricted if only a single DSM category or clinical item was used. Importantly, as noted, traditional clinical scales are uni-directional (i.e. zero is asymptomatic, hence there is an explicit floor). Here, we show that data-driven symptom axes (e.g. *PC3*) were associated with bi-directional variation (i.e. no explicit floor effect). Put differently, patients who score highly on either end of these data-driven axes are severely symptomatic but in very different ways. If these axes reflect clinically meaningful phenomena at both tails then they should more robustly map to neural feature variation, which is in line with reported effects. Therefore, by definition, the derived map for each of the *PCs* will reflect the neural circuitry that may be modulated by the behaviors that vary along that PC (but not others). For example, we named the *PC3* axis *'Psychosis Configuration'* because of its strong loadings onto conventional 'positive' and 'negative' PSD symptoms. This *PC3 'Psychosis Configuration'* showed strong positive variation along neural regions that map onto the canonical default mode network (DMN), which has frequently been implicated in PSD (*Fryer et al., 2013*; *Anticevic et al., 2012b*; *Ongür et al., 2010*; *Woodward et al., 2011*; *Baker et al., 2014*; *Meda et al., 2014*) suggesting that individuals with severe 'positive' symptoms exhibit broadly elevated connectivity with regions of the DMN. On the contrary, this bi-directional *'Psychosis Configuration'* axis also showed strong negative variation along neural regions that map onto the sensory-motor and associative control regions, also strongly implicated in PSD (*Ji et al., 2019a*; *Anticevic et al., 2014*). The 'bi-directionality' property of the *PC* symptom-neural maps may thus be desirable for identifying neural features that support individual patient selection. For instance, it may be possible that *PC3* reflects residual untreated psychosis symptoms in this chronic PSD sample, which may reveal key treatment neural targets. In support of this circuit being symptom-relevant, it is notable that we observed a mild association between GBC and *PC* scores in the CON sample (*Appendix 1—figure 10*). The symptom-neural mapping of dimensionality-reduced symptom scores also produced notable neural maps along other *PC* axes (*Appendix 1—figure 9*). Higher *PC1* scores, indicating higher general functioning, may be associated with lower global connectivity in sensory/cingulate cortices and thalamus, but higher global connectivity with the temporal lobe. Higher *PC2* scores, indicating better cognitive functioning were positively associated with higher medial prefrontal and cerebellar GBC, and negatively with visual cortices and striatum. Importantly, the unique neural circuit variance shown in the $\beta_{PC2}GBC$ map suggest that there are PSD patients with more (or less) severe cognitive deficits independent of any other symptom axis, which would be in line with the observation that these symptoms are not treatable with antipsychotic medication (and therefore should not correlate with symptoms that are treatable by such medications; i.e. *PC3*). Of note, the statistics in the $\beta_{PC2}GBC$ map were relatively moderate, opening up the possibility that a more targeted measure of neural connectivity (e.g. seed FC, see below) would reveal a robust circuit map for this particular symptom dimension. Given the key role of cognitive deficits in PSD (*Barch et al., 2013*), this will be a particularly important direction to pursue, especially in prodromal or early-stage individuals where cognitive dysfunction has been shown to predict illness trajectories (*Fusar-Poli et al., 2012*; *Bora et al., 2014*; *Bora, 2015*; *Antshel et al., 2017*). The *PC4 – Affective Valence* axis was positively associated with higher GBC in cingulate and sensori-motor cortices and subcortically with the thalamus, basal ganglia, and anterior cerebellum, which may be implicated in the deficits in social functioning and affective aspects of this symptom dimension. Lastly, positive scores on *PC5*, which we named *'Agitation/Excitement'*, were associated with broad elevated global connectivity in the frontal associative and sensori-motor cortices, thalamus, and basal ganglia. Interestingly, however, the nucleus accumbens (as well as

hippocampus, amygdala, cerebellum, and temporal lobe) appear to be negatively associated with *PC5* score (i.e. more severe grandiosity, unusual thought, delusional and disorganized thinking), illustrating another example of a bi-directional axis.

## Deriving individually actionable brain-behavior mapping across the psychosis spectrum

Deriving a neurobehavioral mapping that is resolvable and stable at the individual patient level is a necessary benchmark for deploying symptom-neural 'biomarkers' in a clinically useful way. Therefore, there is increasing attention placed on the importance of achieving reproducibility in the psychiatric neuroimaging literature (*Noble et al., 2019*; *Balsters et al., 2016*; *Cao et al., 2019*; *Woo et al., 2017*), which becomes especially important for individualized symptom-neural mapping. Recently, several efforts have deployed multivariate methods to quantify symptom-neural relationships (*Drysdale et al., 2017*; *Xia et al., 2018*; *Smith et al., 2015*; *Moser et al., 2018*; *Rodrigue et al., 2018*; *Yu et al., 2019*), highlighting how multivariate techniques may perhaps provide clinically innovative insights. However, such methods face the risk of overfitting for high-dimensional but underpowered datasets (*Dinga et al., 2019*), as recently shown via formal generative modeling (*Helmer et al., 2020*).

Here, we attempted to use multivariate solutions (i.e. CCA) to quantify symptom and neural feature co-variation. In principle, CCA is well-suited to address the brain-behavioral mapping problem. However, symptom-neural mapping using CCA in our sample was not reproducible even when using a low-dimensional symptom solution and parcellated neural data as a starting point. Therefore, while CCA (and related multivariate methods such as partial least squares) are theoretically appropriate and may be helped by regularization methods such as sparse CCA, in practice many available psychiatric neuroimaging datasets may not provide sufficient power to resolve stable multivariate symptom-neural solutions (*Helmer et al., 2020*). A key pressing need for forthcoming studies will be to use multivariate power calculators to inform sample sizes needed for resolving stable symptom-neural geometries at the single subject level. Of note, athough we were unable to derive a stable CCA in the present sample, this does not imply that the multivariate neurobehavioral effect may not be reproducible with larger effect sizes and/or sample sizes. Critically, this does highlight the importance of power calculations prior to computing multivariate brain-behavioral solutions (*Helmer et al., 2020*).

Consequently, we tested if a low-dimensional symptom solution can be used in a univariate symptom-neural model to optimize individually predictive features. Indeed, we found that a univariate brain-behavioral space (BBS) relationship can result in neural features that are stable for individualized prediction. Critically, we found that if a patient exhibited a high PC symptom score, they were more likely to exhibit a topography of neural $\Delta GBC$ that was topographically similar to the $\beta_{PC}GBC$ map. This suggests that optimizing such symptom-neural mapping solutions (and ultimately extending them to multivariate frameworks) can inform cross-diagnostic patient segmentation with respect to symptom-relevant neural features. Importantly, this could directly inform patient identification based on neural targets that are of direct symptom relevance for clinical trial design.

## Utilizing independent molecular neuroimaging maps to 'benchmark' symptom-relevant neural features

Selecting single patients via stable symptom-neural mapping of BBS solutions is necessary for individual patient segmentation, which may ultimately inform treatment indication. However, it is also critical to be able to relate the derived symptom-neural maps to a given mechanism. Here we highlight two ways to 'benchmark' the derived symptom-neural maps by calculating their similarity against independent pharmacological neuroimaging and gene expression maps. We show a proof-of-principle framework for quantifying derived symptom-neural reference maps with two PSD-relevant neuropharmacological manipulations derived in healthy adults via LSD and ketamine. These analyses revealed that selecting single patients, via the derived symptom-neural mapping solution, can yield above-chance quantitative correspondence to a given molecular target map. These data highlight an important effect: it is possible to construct a 'strong inference' (*Platt, 1964*) evaluation of single patients' differential similarity to one molecular target map versus another. For instance, this approach could be applied to maps associated with already approved PSD treatments (such as

clozapine, olanzapine, or chlorpromazine [*Lally and MacCabe, 2015*; *Miyamoto et al., 2005*]) to identify patients with symptom-neural configurations that best capture available treatment-covarying neural targets.

Relatedly, AHBA gene expression maps (*Burt et al., 2018*) may provide an a priori benchmark for treatment targets that may be associated with a given receptor profile. Here we show that identified BBS maps exhibit spatial correspondence with neural gene expression maps implicated in PSD – namely serotonin, GABA and interneuron gene expression. This gene-to-BBS mapping could be then used to select those patients that exhibit high correspondence to a given gene expression target.

Collectively, this framework could inform empirically testable treatment selection methods (e.g. a patient may benefit from ketamine, but not serotonin agonists such as LSD/psilocybin). In turn, this independent molecular benchmarking framework could be extended to other approaches (e.g. positron emission tomography (PET) maps reflecting specific neural receptor density patterns [*Farde et al., 1986*; *Arakawa et al., 2008*]) and iteratively optimized for quantitative patient-specific selection against actionable molecular targets.

## Considerations for generalizing solutions across time, severity, and mental health spectra

There are several constraints of the current result that require future optimization – namely the ability to generalize across time (early course vs. chronic patients), across a range of symptom severity (e.g. severe psychotic episode or persistent low-severity psychosis) and across distinct symptom spectra (e.g. mood). This applies to both the low-rank symptom solution and the resulting symptom-neural mapping. It is possible that the derived lower-dimensional symptom solution, and consequently the symptom-neural mapping solution, exhibits either time-dependent (i.e. state) or severity-dependent (i.e. trait) re-configuration. Relatedly, medication dose, type, and timing may also impact the solution. Another important aspect that will require further characterization is the possibility of oblique axes in the symptom-neural geometry. While orthogonal axes derived via PCA were appropriate here and similar to the ICA-derived axes in this solution, it is possible that oblique dimensions more clearly reflect the geometry of other psychiatric spectra and/or other stages in disease progression. For example, oblique components may better capture dimensions of neurobehavioral variation in a sample of prodromal individuals, as these patients are exhibiting early-stage psychosis-like symptoms and may show signs of diverging along different trajectories.

Critically, these factors should constitute key extensions of an iteratively more robust model for individualized symptom-neural mapping across the PSD and other psychiatric spectra. Relatedly, it will be important to identify the 'limits' of a given BBS solution – namely a PSD-derived effect may not generalize into the mood spectrum (i.e. both the symptom space and the resulting symptom-neural mapping is orthogonal). It will be important to evaluate if this framework can be used to initialize symptom-neural mapping across other mental health symptom spectra, such as mood/anxiety disorders.

These types of questions will require longitudinal and clinically diverse studies that start prior to the onset of full-blown symptoms (e.g. the North American Prodrome Longitudinal Study (NAPLS) [*Addington et al., 2012*; *Seidman et al., 2010*]). A corollary of this point is that ~50% of unexplained symptom variance in the current PCA solution necessitates larger samples with adequate power to map this subtle, but perhaps clinically essential, PSD variation.

Notably, the current cohort was adequately powered for symptom data reduction that drove univariate neural mapping. However, this sample was insufficiently powered for resolving stable multivariate symptom-neural relationships even with dimensionality-reduced symptom features. Consequently, the limited sample size necessitated choices for dimensionality reduction of the neural feature space in this study even for univariate analyses. While both parcellation and GBC constitute principled choices, symptom-relevant neural information may have been lost (which may be embedded in a higher dimensional space). One obvious solution is to increase sample sizes (e.g. via datasets such as the UK Biobank [*Sudlow et al., 2015*]). However, in parallel, it will be critical to develop neurobiologically informed feature space reduction and/or to optimize the stability of multivariate solutions via regularization methods. Another improvement would be to optimize neural data reduction sensitivity for specific symptom variation (*Shehzad et al., 2014*). We chose to use GBC for our initial geometry characterizations as it is a principled and assumption-free data-reduction metric that

captures (dys)connectivity across the whole brain and generates neural maps (where each region can be represented by a value, in contrast to full functional connectivity matrices) that are necessary for benchmarking against molecular imaging maps. However, GBC is a summary measure that by definition does not provide information regarding connectivity between specific pairs of neural regions, which may prove to be highly symptom-relevant and informative. Thus symptom-neural relationships should be further explored with higher-resolution metrics, such as restricted GBC (*Anticevic et al., 2013*), which can summarize connectivity information for a specific network or subset of neural regions, or seed-based FC using regions implicated in PSD (e.g. thalamus [*Anticevic et al., 2014*; *Woodward et al., 2011*]).

Here, we focused on the neural blood oxygen level dependent (BOLD) signal from fMRI. However, other modalities such as diffusion-weighted imaging (DWI), PET imaging, or electroencephalography (EEG) could be leveraged. Additional clinically-relevant information could be derived from genetics (such as polygenic risk scores [*Hyman, 2011*; *O'Donovan et al., 2009*; *Cao et al., 2020*]) or ecological momentary assessment (EMA) (*Corcoran et al., 2018*; *Niendam et al., 2018*), especially to parse state vs. trait biomarker variation.

Lastly, building on the proof-of-concept molecular neuroimaging comparisons, it will be imperative to eventually test such predictions in formal clinical trials. An actionable next step would be to optimize patient selection against existing treatments, which could result in higher success rates for drug development trials and potentially have massive impact for developing new interventions. Critically, the opportunity to develop, validate, and refine an individual-level quantitative framework could deliver a more rapid and cost-effective way of pairing patients with the right treatments.

## Conclusions

We show that complex and highly heterogeneous PSD symptom variation can be robustly reduced into a low-rank symptom solution that is cross-diagnostic, individually predictive, generalizable and incorporates cognitive deficits. In turn, the derived PSD symptom axes robustly mapped onto distinct yet reproducible neural patterns, which were predictive at the single-patient level. Leveraging these stable results, we show a proof-of-principle framework for relating the derived symptom-relevant neural maps, at the individual patient level, with molecular targets implicated in PSD via LSD and ketamine neuro-pharmacological manipulations. Lastly, we used AHBA gene expression maps to show that identified PSD symptom-relevant neural maps co-vary with serotonin, GABA and interneuron neural gene expression patterns. This set of symptom-neural mapping results can be iteratively and quantitatively optimized for personalized treatment segmentation endpoints.

# Materials and methods

## Overall data collection and study design

We used publicly available behavioral and neural data from the Bipolar-Schizophrenia Network on Intermediate Phenotypes (BSNIP) consortium (*Tamminga et al., 2014*). All data were obtained from the National Data Archive (NDA) repository (https://nda.nih.gov/edit_collection.html?id=2274). Participants were collected at six sites across the United States. Full recruitment details are provided in prior publications (*Tamminga et al., 2014*; *Meda et al., 2015*; *Sheffield et al., 2017*). In brief, participants were recruited through advertising in Baltimore MD, Boston MA, Chicago IL, Dallas TX, and Hartford CT. All assessments were standardized across sites. Participants were excluded if they had (i) a history of seizures or head injury resulting in >10 min loss of consciousness, (ii) positive drug screen on the day of testing, (iii) a diagnosis of substance abuse in the past 30 days or substance dependence in the past 6 months, (iv) history of serious medical or neurological disorder that would likely affect cognitive functioning, (v), history of serious medical or neurological disorder that would likely affect cognitive functioning, (vi) insufficient English proficiency, or (vii) an age-corrected Wide-Range Achievement Test (4th edition) reading test standard score <65. Additionally, participants were required to have had no change in medication and been clinically stable over the past month. Participants completed a SCID interview and were given a diagnosis via consensus from study clinicians; participants with an Axis one clinical psychosis diagnosis were additionally given assessments including the Positive and Negative Symptom Scale (PANSS)(*Kay et al., 1987*). Note that apart from the measures used in this paper, the full BSNIP dataset includes a rich characterization of measures

from multiple modalities, including electrophysiological, eye tracking, structural and diffusion neuro-imaging, as well as a number of clinical batteries, which are not quantified in this study. We used data from a total of 638 participants with complete behavioral and neural data after preprocessing and quality control, including 202 healthy controls (CON), 167 patients diagnosed with schizophrenia (SZP), 119 patients with schizoaffective disorder (SADP), and 150 patients with bipolar disorder with psychosis (BPP). For full demographic and clinical characteristics of the sample see *Appendix 1—table 2*.

## Neural data acquisition and preprocessing

Participants completed a neural magnetic resonance imaging (MRI) scan at 3T, including resting-state functional blood-oxygen-level-dependent imaging (BOLD) and a magnetization-prepared rapid gradient-echo (MP-RAGE) sequence for T1 weighted data. Full details on scanners and acquisition parameters used at each of the sites have previously been described (*Meda et al., 2012*) and are summarized here in *Appendix 1—table 3*. Neuroimaging data were preprocessed using the Human Connectome Project (HCP) minimal preprocessing pipeline (*Glasser et al., 2013*), adapted for compatibility with 'legacy' data, which are now featured as a standard option in the HCP pipelines provided by our team (https://github.com/Washington-University/HCPpipelines/pull/156, copy archived at swh:1:rev:1334b35ab863540044333bbdec70a68fb19ab611; *Repovš et al., 2021*). These modifications to the HCP pipelines were necessary as the BSNIP data did not include a standard field map and did not incorporate a T2w high-resolution image. The adaptations for single-band BOLD acquisition have previously been described in detail (*Ji et al., 2019a*).

A summary of the HCP pipelines is as follows: the T1-weighted structural images were first aligned by warping them to the standard Montreal Neurological Institute-152 (MNI-152) brain template in a single step, through a combination of linear and non-linear transformations via the FMRIB Software Library (FSL) linear image registration tool (FLIRT) and non-linear image registration tool (FNIRT) (*Jenkinson et al., 2002*). Next, FreeSurfer's recon-all pipeline was used to segment brain-wide gray and white matter to produce individual cortical and subcortical anatomical segmentations (*Reuter et al., 2012*). Cortical surface models were generated for pial and white matter boundaries as well as segmentation masks for each subcortical gray matter voxel. Using the pial and white matter surface boundaries, a 'cortical ribbon' was defined along with corresponding subcortical voxels, which were combined to generate the neural file in the Connectivity Informatics Technology Initiative (CIFTI) volume/surface 'grayordinate' space for each individual subject (*Glasser et al., 2013*). BOLD data were motion-corrected by aligning to the middle frame of every run via FLIRT in the initial NIFTI volume space. In turn, a brain-mask was applied to exclude signal from non-brain tissue. Next, cortical BOLD data were converted to the CIFTI gray matter matrix by sampling from the anatomically defined gray matter cortical ribbon and subsequently aligned to the HCP atlas using surface-based nonlinear deformation (*Glasser et al., 2013*). Subcortical voxels were aligned to the MNI-152 atlas using whole-brain non-linear registration and then the Freesurfer-defined subcortical segmentation applied to isolate the subcortical grayordinate portion of the CIFTI space.

After the HCP minimal preprocessing pipelines, movement scrubbing was performed (*Power et al., 2013*). As done in our prior work (*Anticevic et al., 2012a*), all BOLD image frames with possible movement-induced artifactual fluctuations in intensity were flagged using two criteria: frame displacement (the sum of the displacement across all six rigid body movement correction parameters) exceeding 0.5 mm (assuming 50 mm cortical sphere radius) and/or the normalized root mean square (RMS) (calculated as the RMS of differences in intensity between the current and preceding frame, computed across all voxels and divided by the mean intensity) exceeding 1.6 times the median across scans. Any frame that met one or both of these criteria, as well as the frame immediately preceding and immediately following, were discarded from further preprocessing and analyses. Subjects with more than 50% frames flagged using these criteria were excluded. Next, a high-pass filter (threshold 0.008 Hz) was applied to the BOLD data to remove low-frequency signals due to scanner drift. In-house Matlab code was used to calculate the average variation in BOLD signal in the ventricles, deep white matter, and across the whole gray matter, as well as movement parameters. These signals, as well as their first derivatives to account for delayed effects, were then regressed out of the gray matter BOLD time series as nuisance variables (as any change in the BOLD signal due to these variables are persistent and likely not reflecting neural activity) (*Power et al., 2018*). It should be noted that using global signal regression to remove spatially persistent artifact is

highly controversial in neuroimaging (*Power et al., 2017*; *Yang et al., 2017*), but it remains the field-wide gold standard (though see other recent and emerging approaches at [*Glasser et al., 2018*; *Aquino et al., 2019*]).

## Behavioral symptom and cognitive data

The behavioral measures analyzed included the Positive and Negative Syndrome Scale (PANSS), an assessment of psychosis symptom severity (*Kay et al., 1987*), and the Brief Assessment of Cognition in Schizophrenia (BACS) battery, which provided an assessment of cognitive functioning (*Keefe et al., 2004*). Only control subjects with complete BACS measures were used for analyses; PANSS symptom scores were imputed for control subjects for whom the PANSS had not been administered under the assumption that these subjects were asymptomatic. Only patient subjects with complete PANSS and BACS measures were used in analyses. The BACS scores used here are presented as standardized Z-scores normalized to the mean and standard deviation of the control group for each BSNIP site, as done in prior work (*Tamminga et al., 2014*). The full PANSS battery is conventionally divided into three sub-scales: Positive symptom scale (seven items), Negative symptom scale (seven items), and General Psychopathology symptom scale (16 items). The BACS battery consists of 6 individual sub-scores (*Keefe et al., 2004*). In total, this yielded 36 symptom variables per participant. Effects of symptom variation across assessment sites have been rigorously characterized in prior publications (*Tamminga et al., 2013*). Nevertheless, we explicitly tested for site effects in our analyses, described in detail below.

## Principal component analysis of behavioral symptom measures

The principal component analysis (PCA) of behavioral data was computed by including all 36 symptom variables across all N = 436 patients. Variables were first scaled to have unit variance across patients before running the PCA. Significance of the derived principal components (PCs) was computed via permutation testing. For each permutation, patient order was randomly shuffled for each symptom variable before re-computing PCA. This permutation was repeated 5000 times to establish the null model. PCs which accounted for a proportion of variance that exceeded chance (p<0.05 across all 5000 permutations) were retained for further analysis.

## Stability and reproducibility of the symptom PCA

Stability of the symptom PCA solution was assessed with within-sample cross-validation. To evaluate if there were site differences that uniquely drove the PCA solution, we performed a leave-site-out cross-validation analysis. Specifically, we re-ran the PCA using all patients except those from one site, which was held out. This process was repeated for each of the six sites. To further evaluate the stability of the derived PCA solutions, we performed 1000 runs of *k*-fold cross-validation for values of *k* between 2 and 10. For each *k*-fold run, the full sample of patients was randomly divided into equally-sized *k* sets and a PCA was re-computed using subjects in *k*-1 sets (as the left-out set was used as the held-out sample).

For each run of leave-site-out and *k*-fold cross-validations significance was assessed via permutation testing as described above. The number of significant PCs and the total proportion of variance explained by these significant PCs remained highly stable across all runs (see *Figure 2A* and *Appendix 1—figure 4* and *Appendix 1—figure 5*). Additionally, we compared observed and predicted PCA scores for the held-out sample of patients. Predicted scores for the held-out sample of patients were computed by weighing their raw symptom scores with the loadings from the PCA computed in all other patients. Observed scores for held-out patients were obtained from the original PCA computed on the full sample presented in the main text. The similarity between predicted and observed scores was high for all five significant PCs across all runs of leave-site-out and *k*-fold cross-validation, exceeding *r* = 0.9 in most analyses (see *Figure 2B–C* and *Appendix 1—figure 4* and *5*). Notably, the PCA solution did not show medication status or dosage effects (*Appendix 1—figure 6*). We also examined effects of age, sex, and SES in *Appendix 1—figure 7*. Briefly, we observed a significant positive relationship between age and *PC3* scores, which may be because older patients (whom presumably have been ill for a longer time) exhibit more severe symptoms along the positive *PC3 – Psychosis Configuration* dimension. We also observed a significant negative relationship between Hollingshead index of SES and *PC1* and PC2 scores. Lower *PC1* and PC2 scores indicate poorer

general functioning and cognitive performance respectively, which is consistent with higher Hollingshead indices (i.e. lower-skilled jobs or unemployment and fewer years of education). We also found significant sex differences in *PC2 – Cognitive Functioning*, *PC4 – Affective Valence*, and *PC5 – Agitation/Excitement* scores.

We also assessed the similarity of the PCA loadings using leave-site-out and 1000 runs of fivefold cross-validation frameworks (*Figure 2D*). Importantly, this cross-validation was designed to test if the observed loadings remained stable (as opposed to predicted patient-level scores). The loadings for significant PCs from each leave-site-out PCA solution as well as each run of the fivefold cross-validation were highly correlated (*Figure 2D*).

We also assessed the reproducibility of the PCA solution using independent split-half samples. The leave-site-out and *k*-fold cross-validation PCA analyses by definition use overlapping samples of patients for each iteration. Therefore, we additionally conducted a full split-half replication using entirely non-overlapping sets of patients in each iteration. For each split-half iteration, the full patient sample was randomly divided into two sets with equal proportions of each of the three diagnostic groups (BPP, SADP, SZP). Then, a PCA was computed using each of the split-half patient samples. The loadings from the two PCA solutions were then evaluated for reproducibility. This process was repeated 1000 times. The loadings for significant PCs were again highly similar even when comparing PCA solutions derived from completely non-overlapping patient samples (*Figure 2D*).

To predict individual patient PC scores for the leave-one-out analysis (*Appendix 1—figure 26A*), a PCA was computed using all patients except one held-out patient (N = 435). In turn, the derived loadings were then used to compute the predicted PC scores for the left-out patient. This process was repeated until predicted PC scores were calculated for every patient. Finally, the predicted score for each patient was evaluated for reproducibility relative to the observed score obtained from the PCA solution computed using the full N = 436 sample of patients. In addition, we computed an independent component analysis (ICA) to evaluate the consistency of the behavioral data reduction solution across methods (see *Materials and methods*).

## Global brain connectivity calculation

Following preprocessing, the functional connectivity (FC) matrix was calculated for each participant by computing the Pearson's correlation between every grayordinate in the brain with all other grayordinates. A Fisher's r-to-Z transform was then applied. Global brain connectivity (GBC) was calculate by computing every grayordinate's mean FC strength with all other grayordinates (i.e. the mean, per row, across all columns of the FC matrix). GBC is a data-driven summary measure of connectedness that is unbiased with regards to the location of a possible alteration in connectivity (*Cole et al., 2016*) and is therefore a principled way for reducing the number of neural features while assessing neural variation across the entire brain.

$$GBC(x) = \frac{1}{N}\sum_{y=1}^{N} r_{xy} \tag{1}$$

- where $GBC(x)$ denotes the GBC value at grayordinate $x$;
- where $N$ denotes the total number of grayordinates;
- where $\sum_{y=1}^{N}$ denotes the sum from $y = 1$ to $y = N$;
- where $r_{xy}$ denotes the correlation between the time-series of grayordinates $x$ and $y$;

For parcel-wise GBC maps (described below), we first computed the mean BOLD signal within each parcel (see section below for parcellation details) for each participant and then computed the pairwise FC between all parcels. Finally, to obtain the parcellated GBC metric we computed the mean FC for each parcel and all other parcels. This order of operations (first parcellating the dense data series and then computing GBC) was chosen because it resulted in stronger statistical values due to increased within-parcel signal-to-noise of the BOLD data (*Figure 4A*).

$$GBC(p) = \frac{1}{N}\sum_{q=1}^{N} r_{pq} \tag{2}$$

- where $GBC(p)$ denotes the GBC value at parcel $p$;
- where $N$ denotes the total number of parcels;
- where $\sum_{y=1}^{N}$ denotes the sum from $q = 1$ to $q = N$;
- where $r_{pq}$ denotes the correlation between the time-series of parcels $p$ and $q$;

## Neural data reduction via functional brain-wide parcellation

Here, we applied a recently developed Cole-Anticevic Brain Network Parcellation (CAB-NP) parcellation (*Ji et al., 2019b*), which defines functional networks and regions across cortex and subcortex that leveraged the Human Connectome Project's Multi-Modal Parcellation (MMP1.0) (*Glasser et al., 2016*; *Ji et al., 2019b*). The final published CAB-NP 1.0 parcellation solution can be visualized via the Brain Analysis Library of Spatial maps and Atlases (BALSA) resource (https://balsa.wustl.edu/rrg5v) and downloaded from the public repository (https://github.com/ColeLab/ColeAnticevicNetPartition, copy archived at swh:1:rev:94772010ac26f487fd6baf1d33d121d57a37e0ed; *Cole et al., 2021*). The cortex component of the parcellation solution is comprised of 180 bilateral parcels (a total of 360 across both left and right hemispheres), consistent with the Human Connectome Project's Multi-Modal Parcellation (MMP1.0) (*Glasser et al., 2016*). The subcortex component is comprised of 358 parcels defined using resting-state functional BOLD covariation with the cortical network solution (*Ji et al., 2019b*).

## Mass univariate symptom-neural mapping

Behavioral scores (a priori and PCA) were quantified in relation to individual GBC variation (either dense or parcellated) via a mass univariate regression procedure. The resulting map of regression coefficients reflected the strength of the relationship between patients' behavioral PC score and GBC at every neural location, across all 436 patients. The greater the magnitude of the coefficient for a given location, the stronger the statistical relationship between GBC and the behavioral variation across patients. The coefficients were then Z-scored for each map. Significance of the maps was assessed via nonparametric permutation testing, 2000 random shuffles with TFCE (*Smith and Nichols, 2009*) type-I error-protection computed via the Permutation Analysis of Linear Models program (*Winkler et al., 2014*).

## Cross-validation and split-half replication of mass univariate symptom-neural mapping

Stability of the symptom-neural mapping was assessed following the same cross-validation logic as described for the symptom-driven PCA solutions. Specifically, fivefold cross-validation was performed by first randomly partitioning all patients (N = 436) into five subsets. Regression of the behavioral PC scores onto GBC across patients was performed while holding out 1/5 of the patient sample (N = 349). The correlation between the resulting neural coefficient map was then computed with the neural map obtained from the full sample calculation. For leave-site-out cross-validation, all subjects except those from one site were used when calculating the symptom-neural regression model. The resulting maps were compared to the map from the full sample regression. This process was repeated six times, each time leaving out subjects in one of the six sites.

Reproducibility of the symptom-neural mapping was assessed with 1000 iterations of split-half replication. For each iteration, the full sample of subjects was first randomly split into two halves (referred to as H1 and H2) with the proportion of subjects in each diagnostic group (BPP, SADP, SZP) preserved within each half. For each iteration we used the H1 PCA solution and loadings to compute the predicted PCA scores for H2. In turn, the observed H1 scores were computed from a PCA loadings on the same H1 half-sample of patients. These H1 scores were then regressed against parcelled GBC for patients in H2. This coefficient GBC map reflects the strength of the relationship between the predicted PC score and GBC across H2 patients. Finally, the GBC coefficient maps

derived from the H1 observed and H2 predicted PCA scores were then correlated for each PC axis. This process was then repeated 1000 times and evaluated for reproducibility.

## Principal component analysis of neural data

To evaluate the consistency of the neural GBC, we computed a PCA solution for the parcellated neural GBC data for all 202 control subjects as well as for all 436 patients (*Appendix 1—figure 19*). The resulting neural GBC-derived PCs capture the striking consistency of the neural variance components irrespective of clinical status, which highlights that the bulk of the neural variance is not symptom-relevant. As with the behavioral PCA, significance of the neural PCA solution was assessed via permutation testing (1000 random shuffles of parcels within subject). To evaluate the effect of shared neural variance on the symptom-neural solution, we parsed out common neural PCs from the PSD GBC data. We first conducted a PCA using the parcellated GBC data from all 436 PSD and 202 CON (a matrix with dimensions 638 subjects x 718 parcels). This GBC-PCA resulted in 637 independent GBC-PCs. Since PCs are orthogonal to each other, we then partialled out the variance attributable to GBC-PC1 from the PSD data by reconstructing the PSD GBC matrix using only scores and coefficients from the remaining 636 GBC-PCs ($G\hat{B}C_{woPC1}$). We then reran the univariate regression as described in *Appendix 1—figure 2*, using the same five symptom PC scores across 436 PSD. We also repeated the symptom-neural regression with the first two GBC-PCs partialled out of the PSD data ($G\hat{B}C_{woPC1-2}$), with the first three PCs parsed out ($G\hat{B}C_{woPC1-3}$), and with the first four neural PCs parsed out ($G\hat{B}C_{woPC1-4}$).

## Canonical correlation analysis

Canonical correlation analysis (CCA) is a multivariate statistical technique which examines simultaneously the relationships between multiple independent variables and multiple dependent variables by computing linear combinations of each variable set that maximizes the correlations between the two sets (*Figure 5A*). Here, the two variates used were symptom and neural features across all subjects. Each feature was Z-scored prior to computing the CCA solution. Given the size of the 'dense' neural feature space reduced the number of neural features in a principled manner via the described parcellation. This reduced the number of neural features to 180 symmetrized cortical parcels (i.e. GBC was averaged for each pair of analogous parcels in the left and right hemispheres). Critically, corresponding cortical parcels in the left and right hemispheres of this parcellation have been shown to be highly similar (*Glasser et al., 2013*; *Burt et al., 2018*). First, we computed the CCA solution using all 36 item-level symptom measures as behavioral features. In turn, we computed an additional CCA solution using scores from the significant five PCs.

Each symptom PC is a weighted linear composite of the original symptom measures and each behavioral CV is a weighted linear composite of the symptom PCs. Therefore, to compute the loadings of the 36 symptom measures on each of the behavioral CVs (computed using the symptom PCs), we multiplied the matrix of loadings from the CCA with the matrix of loadings from the PCA (*Figure 5J*).

## CCA cross-validation and split-half replication

Stability of the CCA solution was assessed via cross-validation. Five-fold cross-validation of the CCA was performed by first randomly partitioning all patients (N = 436) into five subsets. The CCA was then performed between neural and behavioral features using all but one of the subsets. The results of this CCA was then compared to the full sample CCA. *Appendix 1—figure 17A-D* shows the comparisons across all five five-fold cross-validation runs of the CCA compared to the full model, including neural factor loadings, behavioral factor loadings, and projected symptom measure loadings. For leave-site-out cross-validation, all subjects except those from one site were used in the CCA. The resulting outputs were then compared to those from the full sample CCA. This process was repeated for all six sites. These data are shown in *Appendix 1—figure 17E-H*.

For leave-one-out cross-validation of the CCA, one subject was held out as a CCA was performed using neural and behavioral features from the other 435 subjects. The loadings matrices  and  from the CCA were then used to calculate the 'predicted' neural and behavioral latent scores for all five canonical modes for the holdout subject. This was repeated for every subject, such that predicted neural and behavioral latent score matrices ($\hat{N}$ and $\hat{B}$) were computed, of the same dimensions as ($N$

and $B$) respectively. The corresponding CVs in $\hat{N}$ and $\hat{B}$ were then correlated across subjects, as shown in *Figure 5M*. If the CCA solution is stable, these correlations should be comparable to the canonical correlations of the full CCA (*Figure 5D*).

Reproducibility of the CCA solution was assessed via split-half replication. This was performed by first randomly splitting the full sample into two halves (H1 and H2) with the proportion of subjects in each diagnostic group (BPP, SADP, SZP) preserved within each half. A CCA was then performed separately in each half and the resulting outputs were then compared to each other. This process was repeated 1000 times to obtain mean and standard deviation performance metrics. These data are shown in *Figure 5L*.

## Multivariate CCA power analysis

Multivariate power analyses to estimate the minimum sample size needed to sufficiently power a CCA were computed using methods described in *Helmer et al., 2020*, using the Generative Modeling of Multivariate Relationships tool (*gemmr*, https://github.com/murraylab/gemmr [v0.1.2, copy archived at swh:1:rev:672c2fbb8ff8f5b08afb16fe9b790536c69f2cf3; *Helmer, 2021*]). Briefly, a model was built by: (1) Generating synthetic datasets for the two input data matrices, by sampling from a multivariate normal distribution with a joint covariance matrix that was structured to encode CCA solutions with specified properties; (2) Performing CCAs on these synthetic datasets. Because the joint covariance matrix is known, the true values of estimated association strength, weights, scores, and loadings of the CCA, as well as the errors for these four metrics, can also be computed. In addition, statistical power that the estimated association strength is different from 0 is determined through permutation testing; (3) Varying parameters of the generative model (number of features, assumed true between-set correlation, within-set variance structure for both datasets) the required sample size $n_{req}$ is determined in each case such that statistical power reaches 90% and all of the above described error metrics fall to a target level of 10%; and (4) Fitting and validating a linear model to predict the required sample size $n_{req}$ from parameters of the generative model. This linear model was then used to calculate $n_{req}$ for CCA in three data scenarios: (i) 718 neural vs. 5 symptom features; (ii) 180 neural vs. 5 symptom features; (iii) 12 neural vs. 5 symptom features.

## Independent replication dataset

To illustrate the generalizability of the single-subject prediction results across independent datasets and DSM disorders, we use an independently collected dataset consisting of 30 patients with a formal diagnosis of schizophrenia (SZP) and 39 patients diagnosed with obsessive compulsive disorder (OCD). These patients were recruited via clinician referral and regional advertising and assessed at the Connecticut Mental Health Center in New Haven, CT. Demographic and symptom data are shown in *Appendix 1—table 4*.

The behavioral assessment for the replication dataset included the PANSS and the Penn Computerized Neurocognitive Battery (PennCNB)(*Moore et al., 2015*). Items from the PennCNB were matched to items from the BACS as follows: BACS digital sequencing (working memory) to LNB (N-back Total score); BACS symbol coding (attention) -to Penn Continuous Performance Test; BACS token motor task (motor) to CTAP; BACS Tower of London (abstraction and mental flexibility) to PCET or RAVEN (Penn Conditional Exclusion Test); BACS verbal fluency (verbal ability) to SPVRT (Penn Verbal Reasoning Test); BACS verbal memory test to PLLT (Penn List Learning Test) (*Gur et al., 2010*).

Neural data were collected using a Siemens 3T scanner with a 64-channel head coil at the Yale Center for Biomedical Imaging. Imaging acquisition parameters were aligned with those of the Human Connectome Project (HCP) (*Van Essen et al., 2013*). High-resolution T1w and T2w structural images were acquired in 224 AC-PC aligned slices, 0.8 mm isotropic voxels. T1w images were collected with a magnetization-prepared rapid gradient-echo (MP-RAGE) pulse sequence (TR = 2400 ms, TE = 2.07 ms, flip angle = 8°, field of view = 256×256 mm). T2w images were collected with a SCP pulse sequence [TR = 3200 ms, TE = 564 ms, flip angle mode = T2 var, field of view = 256×256 mm]. Resting-state BOLD images were collected with a multi-band accelerated fast gradient-echo, echo-planar sequence (acceleration factor = 6, time repetition (TR) = 800 ms, time echo (TE) = 31.0 ms, flip angle = 55°, field of view = 210×210 mm, matrix = 84×84, bandwidth = 2290 Hz); 54 interleaved axial slices aligned to the anterior-posterior commissure (AC-PC) with 2.5 mm isotropic

voxels. Additionally, a pair of reverse phase-encoded spin-echo field maps (anterior-to-posterior and posterior-to-anterior) were acquired (voxel size = 2.5 mm isotropic, TR = 7220 ms, TE = 73 ms, flip angle = 90°, field of view = 210×210 mm, bandwidth = 2290 Hz). These neural data were preprocessed using the HCP minimal preprocessing pipeline as described in the section 'Neural data acquisition and preprocessing' above, with the same motion scrubbing and nuisance signal regression parameters as were used for the BSNIP dataset.

## Neural parcel ΔGBC feature selection and prediction

For the purposes of patient selection, we were focused on individual differences in neural features as they co-vary in relation to individual symptom scores. However, this individual variation in neural features may be small compared to the overall group mean. Hence for each patient, we compute the difference in each brain location (parcel) between the patient's actual GBC and the group mean GBC for that location. This is denoted by ΔGBC. Importantly, using a de-meaned GBC metric 'standardizes' the data and helps to correct for possible differences in scanners/protocols across different datasets. Next, we developed an optimized univariate regression framework leveraging a 'dot product' metric to relate a vector of neural features with a single symptom scalar value. This process for neural feature selection (results in *Appendix 1—figure 26*) is shown as a systems flow diagram in *Appendix 1—figure 24*. The observed dot product GBC metric ($dpGBC^{obs}$) is computed as follows:

$$dpGBC_i^{obs} = \begin{pmatrix} \Delta GBC_1^{obs} \\ \vdots \\ \Delta GBC_{P_{select}}^{obs} \end{pmatrix}_i \bullet \begin{pmatrix} \beta GBC_1^{obs} \\ \vdots \\ \beta GBC_{P_{select}}^{obs} \end{pmatrix}_{N-i} \tag{3}$$

- where for a given subject $i$, $dpGBC_i^{obs}$ denotes the dot product GBC value of the two vectors $\Delta GBC_i^{obs} \cdot \beta GBC_{N-1}^{obs}$ across all $P_{select}$ parcels,
- $\Delta GBC_i^{obs}$ is a vector of length $P_{select}$ denoting a difference map of subject $i$'s $GBC^{obs}$ map relative to the group mean $GBC^{obs}$ map, within a given number of parcels $P_{select}$,
- the $\beta GBC_{N-i}^{obs}$ vector denotes the PCA-to-GBC statistical group-level β map for a low-dimensional PC symptom score across selected parcels $P_{select}$ for $N$ subjects excluding subject $i$.

This calculation is then repeated for each subject $i$, resulting in a final vector $dpGBC^{obs} = [dpGBC_1^{obs}, \ldots, dpGBC_N^{obs}]$ for $N$ subjects. There are several key properties of the $dpGBC^{obs}$ statistic: (i) it is not inflated by individual GBC map similarity to the group map because each subject's $\Delta GBC^{obs}$ map is demeaned relative to the reference group computed independently of the left-out subject; (ii) this statistic is not biased by the parcel number (which drops with iterative selection) because the resulting $dpGBC^{obs}$ value variation is quantified relative to the low-dimensional symptom score across all subjects (see *Equation 2*). Put differently, the final evaluation considers the relationship between $dpGBC^{obs}$ and the low-dimensional PC symptom scores across individuals; (iii) The dot product statistic can yield both positive and negative values – a property which some map similarity measures lack (e.g. $\eta^2$); (iv) It is unbounded (unlike a correlation), which is key to maximize co-variation with low-dimensional symptom scores across individuals (see *Equation 2*); (v) The $\Delta GBC_i^{obs}$ map for a given individual is projected onto the basis set of the $\beta GBC_{N-i}^{obs}$ map, which is independent of the left-out individual but directly related to the low-dimensional PC symptom score variance, thus maximizing the dot product optimization.

Next, we select $N - i$ individuals and compute a univariate regression where participants' low dimensional symptom scores are regressed onto the $dpGBC_{N-i}^{obs}$ values:

$$\begin{pmatrix} dpGBC_1^{obs} \\ \vdots \\ dpGBC_{N-i}^{obs} \end{pmatrix} = \alpha \begin{pmatrix} S_1^{obs} \\ \vdots \\ S_{N-i}^{obs} \end{pmatrix} + \epsilon \tag{4}$$

- where for $dpGBC^{obs}$ each element denotes the dot product value of the $\Delta GBC_i^{obs} \cdot \beta GBC_{N-1}^{obs}$ vectors per subject across $P$ parcels,
- α denotes the regression coefficient in the univariate linear model,

- where for $S^{obs}$ each element denotes the observed low-dimensional symptom score (e.g. *PC3* score) per subject,
- $\epsilon$ denotes the error term in the univariate linear model.

After the $dpGBC^{obs} = \alpha S^{obs} + \epsilon$ regression is computed on $N - i$ subjects, it is applied to the left-out-subject $i$. This is repeated for all $N$ subjects and the model is evaluated for the number of parcels that maximize two key dot product evaluation metrics (***Figure 6B***):

Metric A

$$For\ p = \{1, \ldots, P\}\ find\ p\ where$$
$$\Rightarrow A = \max_{r} \left[ \begin{pmatrix} dpGBC_1 \\ \vdots \\ dpGBC_N \end{pmatrix}^{pred}, \begin{pmatrix} S_1 \\ \vdots \\ S_N \end{pmatrix}^{pred} \right] \tag{5}$$

Metric B

$$For\ p = \{1, \ldots, P\}\ find\ p\ where$$
$$\Rightarrow B = \max_{r} \left[ \begin{pmatrix} dpGBC_1 \\ \vdots \\ dpGBC_N \end{pmatrix}^{obs}, \begin{pmatrix} dpGBC_1 \\ \vdots \\ dpGBC_N \end{pmatrix}^{pred} \right] \tag{6}$$

- where $A$ denotes the maximum $r$ correlation value for the two vectors of $dpGBC^{pred}$ values and the predicted $S^{pred}$ low-dimensional symptom scores (e.g. *PC3* axis) for $N$ subjects from the leave-one-out cross-validation,
- where $B$ denotes the maximum $r$ correlation value for the two vectors of *obs* and *pred* $dpGBC$ values for $N$ subjects.

In the initial step in the step-down model, all $P = 718$ parcels are retained in the initial dot product calculation. For each iteration of $P$ selected parcels, the least predictable parcel $p$ (i.e. the parcel with the weakest value in the *PC3* map) is eliminated from the map. Then, the step-down regression is repeated until $P = 1$.

## Pharmacological neuroimaging acquisition in healthy volunteers - LSD

Methods for the lysergic acid diethylamide (LSD) neuroimaging study are described in detail in prior publications (***Preller et al., 2018***). The use of LSD in humans was authorized by the Swiss Federal Office of Public Health, Bern, Switzerland. The study protocol was approved by the Cantonal Ethics Committee of Zurich (KEK-ZH_No: 2014_0496). Participants received both written and oral descriptions of the study procedures and the effects and possible risks of LSD. All participants provided written informed consent in accordance with the Declaration of Helsinki. The study employed a fully double-blind, randomized, within-subject cross-over design with three conditions: (1) placebo + placebo (Pla) condition: placebo (179 mg Mannitol and Aerosil 1 mg po) after pretreatment with placebo (179 mg Mannitol and Aerosil 1 mg po); (2) Pla+LSD (LSD) condition: LSD (100 µg po) after pretreatment with placebo (179 mg Mannitol and Aerosil 1 mg po), or (3) Ketanserin+LSD (Ket+LSD) condition: LSD (100 µg po) after pretreatment with the 5-HT2A antagonist Ket (40 mg po). Data were collected for all subjects in a randomized counterbalanced order at three different sessions each two weeks apart. For all conditions, the first substance was administered 60 min before the second substance, and the first neural scan was conducted 75 min after the second administration, with a second scan conducted at 300 min post-administration. In the present study, only data from the two neural scans for the LSD and Pla conditions were evaluated.

Briefly, neuroimaging data acquisition details for the LSD study are as follows. MRI data were acquired on a Philips Achieva 3.0T whole-body scanner (Best, The Netherlands). A 32-channel receiver head coil and MultiTransmit parallel radio frequency transmission was used. Images were acquired using a whole-brain gradient-echo planar imaging (EPI) sequence (repetition time = 2,500 ms; echo time = 27 ms; slice thickness = 3 mm; 45 axial slices; no slice gap; field of view = 240 mm$^2$; in-plane resolution = 3 mm × 3 mm; sensitivity- encoding reduction factor = 2.0). 240 volumes were acquired per resting state scan resulting in a scan duration of 10 mins. Additionally, two high-

resolution anatomical images were acquired using T1-weighted and T2-weighted sequences. T1-weighted images were collected via a 3D magnetization-prepared rapid gradient-echo sequence (MP-RAGE) with the following parameters: voxel size = 0.7 mm$^3$, time between two inversion pulses = 3123 ms, inversion time = 1055 ms, inter-echo delay = 12 ms, flip angle = 8°, matrix = 320 × 335, field of view = 224 × 235 mm, 236 sagittal slices. Furthermore T2-weighted images were collected using via a turbo spin-echo sequence with the following parameters: voxel size = 0.7 mm$^3$, repetition time = 2500 ms, echo time = 415 ms, flip angle = 90°, matrix = 320 × 335, field of view = 224 mm × 235 mm, 236 sagittal slices.

## Pharmacological neuroimaging acquisition in healthy volunteers - ketamine

Similar to the LSD study, the ketamine pharmacological neuroimaging protocol employed a within-subject design where all healthy volunteer participants underwent a single scanning session consisting of two infusions: (i) placebo (saline solution) followed by (ii) ketamine infusion. The study was approved by the Yale University institutional review board. Participants received both written and oral descriptions of the study procedures and the effects and possible risks of ketamine, and all participants provided written informed consent before beginning the study. Healthy volunteers were informed prior to scanning that they would undergo one placebo run and one ketamine run but were blinded to the order of administration. Because of the sustained effects of ketamine, this infusion was always the second of the two runs, consistent with prior work (*Anticevic et al., 2012a*). A doctor and two nurses as well as a study coordinator remained present for the duration of the scan. A bolus of ketamine (0.3 mg/kg of bodyweight) or saline were delivered via infusion 5 s after the start of the run and then continuously at a rate of 0.65 mg/kg per hour through the duration of the session. The sequence of scans in each for either the placebo or ketamine infusion was as follows: (i) resting state (4.67 min); (ii) blood draw (sham if saline condition); (iii) a cognitive working memory task (total 14 min); (iv) blood draw (sham if saline condition); (v) a cognitive working memory task (total 14 min); (vii) blood draw (sham if saline condition); (vii) a cognitive working memory task(total 8.63 min). Data from the the cognitive working memory task were not used in the present study and are actively undergoing a distinct analysis. Participants were scanned at Yale University on a Siemens Trio 3T whole-body scanner and 32-channel receiver head coil. High-resolution structural T1-weighted images were acquired using an MP-RAGE sequence and the following parameters: voxel size = 0.8 mm$^3$, time between two inversion pulses = 3123 ms, inversion time = 1055 ms, inter-echo delay = 12 ms, flip angle = 8°, matrix = 320 × 335, field of view = 227 × 272 mm, 227 sagittal slices. T2-weighted images were acquired with the following parameters:voxel size = 0.8 mm$^3$, repetition time = 2500 ms, echo time = 415 ms, flip angle = 90°, matrix = 320 × 335, field of view = 227 × 272 mm, 227 sagittal slices. BOLD images were acquired using a whole-brain gradient-echo planar imaging (EPI) sequence (400 frames at TR = 0.7 ms; TE = 30 ms; slice thickness = 2.5 mm; 54 axial slices; no slice gap; field of view = 250 mm$^2$; in-plane resolution = 2.5 mm × 2.5 mm). In addition, a field map and pair of spin-echo gradients were collected at the end of every scanning session.

## Pharmacological neuroimaging processing and analysis for LSD and ketamine samples

All data were preprocessed to be consistent with the BSNIP and replication dataset processing steps. Specifically, we used the HCP minimal preprocessing pipeline and with the same motion scrubbing and nuisance signal regression parameters as were used for the BSNIP dataset as described in the section 'Neural data acquisition and preprocessing' above. Parcel-wise GBC maps were computed as described in the section 'Resting-state functional connectivity and global brain connectivity' above, by first parcellating the dense time-series for each subject and then computing the parcel-level GBC. As with the BSNIP and replication analyses, parcel GBC was first calculated by parcellating the dense time-series for every subject and then computing GBC across all 718 parcels from the whole-brain functional parcellation (*Ji et al., 2019b*; *Glasser et al., 2016*). Group-level GBC maps were computed for every participant for all conditions. We computed the contrast map for 'LSD-Placebo' as well as 'Ketamine-Placebo' conditions as a t-test between both the pharmacological scans versus placebo scans. The Z-scored t-contrast maps between pharmacological and

placebo conditions were used as a pharmacological target map in relation to the PC-derived neural GBC maps (*Figure 7* and *Appendix 1—figure 29*).

## Neural gene expression mapping

The gene mapping analyses in this study utilize the procedure described in *Burt et al., 2018*. Briefly, we used cortical gene expression data from the publicly available Allen Human Brain Atlas (AHBA, RRID:SCR-007416) , mapped to cortex (*Burt et al., 2018*). Specifically, the AHBA quantified expression levels across 20,737 genes obtained from six postmortem human brains using DNA microarray probes sampled from hundreds of neuroanatomical loci. Recent studies demonstrated the ability to map expression of each gene onto neuroimaging-compatible templates (*Hawrylycz et al., 2012*; *Burt et al., 2018*). Building on these innovations we mapped gene expression on to 180 symmetrized cortical parcels from the HCP atlas (*Glasser et al., 2016*) in line with recently published methods (*Burt et al., 2018*). This yielded a group-level map for each gene where the value in each parcel reflected the average expression level of that gene in the AHBA dataset. These group-level maps were in turn as a gene expression target map in relation to the PC-derived neural GBC maps.

## Code availability

Code and core results presented in this paper are available at https://github.com/AnticevicLab/bbs_manuscript/ (copy archived at swh:1:rev:4776b571bbf9a27a2f43146728a6834edbb552ce; *Ji, 2021*).

# Acknowledgements

We thank JH Krystal and I Levy for useful discussions. This work was supported by the Heffter Research Institute (Grant No. 1–190420), Swiss Neuromatrix Foundation (Grant No. 2016–0111), Swiss National Science Foundation under the framework of Neuron Cofund (Grant No. 01EW1908), Slovenian Research Agency (ARRS) (Grant Nos. J7-8275, J7-6829, P3-0338 to GR), 1U01MH121766 (to AA), R01MH112746 (to JDM), SFARI Pilot Award (to JDM and AA), 5R01MH112189 (to AA), 5R01MH108590 (to AA). Data were provided by the Bipolar- Schizophrenia Network on Intermediate Phenotypes (BSNIP) consortium via the National Institute of Mental Health Data Archive (NDA).

# Additional information

### Competing interests

Jie Lisa Ji: consults for RBNC (formerly BlackThorn Therapeutics) and is a co-inventor for the following pending patent: Anticevic A, Murray JD, Ji JL: Systems and Methods for Neuro-Behavioral Relationships in Dimensional Geometric Embedding (N-BRIDGE), PCT International Application No. PCT/US2119/022110, filed March 13, 2019. Clara Fonteneau, Zailyn Tamayo, Jure Demšar: consults for RBNC (formerly BlackThorn Therapeutics). Joshua B Burt: consulted for BlackThorn Therapeutics in 2019 and is currently an employee of RBNC Therapeutics. Katrin H Preller: is currently an employee of F.Hoffmann-La Roche AG. William J Martin: is an employee of The Janssen Pharmaceutical Companies of Johnson & Johnson and is a co-inventor for the following pending patent: Murray JD, Anticevic A, Martin, WJ: Methods and tools for detecting, diagnosing, predicting, prognosticating, or treating a neurobehavioral phenotype in a subject, U.S. Application No. 16/149,903 filed on October 2, 2018. Grega Repovš: consults for and holds equity with RBNC (formerly BlackThorn Therapeutics). Christopher Pittenger: has consulted in the past 3 years to Biohaven Pharmaceuticals, Lundbeck Parmaceuticals, Ceruvia Therapeutics, Transcend Therapetics, Freedom Biosciences, Teva Pharmaceuticals and Brainsway Therapeutics and receives research funding from Biohaven Pharmaceuticals, the Usona Institute, and Transcend Pharmaceuticals. None of these sources of support are related to the present work. CP is an inventor on two pending patent applications, 16/304,925 and 63/074,275, neither of which is relevant to the current work. John D Murray: consults for and holds equity with RBNC (formerly BlackThorn Therapeutics) and is a coinventor on the following patents: Anticevic A, Murray JD, Ji JL: Systems and Methods for Neuro-Behavioral Relationships in Dimensional Geometric Embedding (N-BRIDGE), PCT International Application No. PCT/US2119/022110, filed March 13, 2019 and Murray JD, Anticevic A, Martin, WJ: Methods and tools for detecting, diagnosing, predicting, prognosticating, or treating a neurobehavioral phenotype in a subject, U.S.

Application No. 16/149,903 filed on October 2, 2018, U.S. Application or PCT International Application No. 18/054,009 filed on October 2, 2018. Alan Anticevic: consults for and holds equity with RBNC (formerly BlackThorn Therapeutics) and is a coinventor on the following patents: Anticevic A, Murray JD, Ji JL: Systems and Methods for Neuro-Behavioral Relationships in Dimensional Geometric Embedding (N-BRIDGE), PCT International Application No. PCT/US2119/022110, filed March 13, 2019 and Murray JD, Anticevic A, Martin, WJ: Methods and tools for detecting, diagnosing, predicting, prognosticating, or treating a neurobehavioral phenotype in a subject, U.S. Application No. 16/149,903 filed on October 2, 2018, U.S. Application for PCT International Application No. 18/054,009 filed on October 2, 2018. The other authors declare that no competing interests exist.

## Funding

| Funder | Grant reference number | Author |
|---|---|---|
| Heffter Research Institute | 1-190420 | Katrin H Preller<br>Franz X Vollenweider |
| Swiss Neuromatrix Foundation | 2016-0111 | Katrin H Preller<br>Franz X Vollenweider |
| Swiss National Science Foundation | 01EW1908 | Katrin H Preller<br>Franz X Vollenweider |
| Slovenian Research Agency | J7-8275 | Grega Repovs |
| Slovenian Research Agency | J7-6829 | Grega Repovs |
| Slovenian Research Agency | P3-0338 | Grega Repovs |
| National Institutes of Health | 1U01MH121766 | Alan Anticevic |
| National Institutes of Health | R01MH112746 | John D Murray |
| National Institutes of Health | 5R01MH112189 | Alan Anticevic |
| National Institutes of Health | 5R01MH108590 | Alan Anticevic |
| Simons Foundation | Pilot Award | John D Murray<br>Alan Anticevic |

The funders had no role in study design, data collection and interpretation, or the decision to submit the work for publication.

## Author contributions

Jie Lisa Ji, Conceptualization, Data curation, Formal analysis, Validation, Visualization, Methodology, Writing - original draft, Writing - review and editing; Markus Helmer, Conceptualization, Formal analysis, Methodology, Writing - review and editing; Clara Fonteneau, Visualization, Writing - review and editing; Joshua B Burt, Resources, Software, Methodology; Zailyn Tamayo, Jure Demšar, Resources, Software; Brendan D Adkinson, Data curation, Project administration; Aleksandar Savić, William J Martin, Conceptualization, Writing - review and editing; Katrin H Preller, Conceptualization, Resources, Data curation, Funding acquisition, Methodology, Writing - review and editing; Flora Moujaes, Conceptualization, Data curation, Software, Methodology, Writing - review and editing; Franz X Vollenweider, Resources, Funding acquisition, Methodology; Grega Repovš, Conceptualization, Resources, Software, Formal analysis, Funding acquisition, Methodology, Writing - review and editing; Youngsun T Cho, Christopher Pittenger, Conceptualization, Resources, Data curation, Supervision, Funding acquisition, Investigation, Methodology; John D Murray, Conceptualization, Resources, Software, Formal analysis, Supervision, Funding acquisition, Methodology, Writing - review and editing; Alan Anticevic, Conceptualization, Resources, Software, Formal analysis, Supervision, Funding acquisition, Validation, Investigation, Visualization, Methodology, Writing - original draft, Project administration, Writing - review and editing

## Author ORCIDs

Jie Lisa Ji https://orcid.org/0000-0002-6280-9070
Markus Helmer http://orcid.org/0000-0001-9680-0595
Brendan D Adkinson http://orcid.org/0000-0003-3196-8674

Katrin H Preller [ID] https://orcid.org/0000-0003-0413-7672
Franz X Vollenweider [ID] https://orcid.org/0000-0001-9053-6164
William J Martin [ID] https://orcid.org/0000-0002-2749-3365
John D Murray [ID] https://orcid.org/0000-0003-4115-8181
Alan Anticevic [ID] https://orcid.org/0000-0002-4324-0536

### Ethics

Human subjects: All data analyzed in this manuscript were collected in prior studies. All studies were approved by the IRBs of the respective institutions (see Methods for details) and participants provided written informed consent in accordance with the Declaration of Helsinki.

### Decision letter and Author response

Decision letter https://doi.org/10.7554/eLife.66968.sa1
Author response https://doi.org/10.7554/eLife.66968.sa2

## Additional files

### Supplementary files

• Transparent reporting form

### Data availability

Data and code for all core results can be found at https://github.com/AnticevicLab/bbs_manuscript/ (copy archived at https://archive.softwareheritage.org/swh:1:rev:200c4c24fcaf43632fcbd4f3d65bffd8094b00aa).

The following previously published dataset was used:

| Author(s) | Year | Dataset title | Dataset URL | Database and Identifier |
| --- | --- | --- | --- | --- |
| Tamminga C | 2013 | Bipolar & Schizophrenia Consortium for Parsing Intermediate Phenotypes (B-SNIP 1) | https://nda.nih.gov/edit_collection.html?id=2274 | NIMH Data Archive, 2274 |

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

## Appendix 1

### Note 1 - ICA of behavioral symptom data

We performed an independent component analysis (ICA) on the symptom data as a comparison to our PCA data-reduction solution. Specifically, an ICA was conducted on the behavioral data as an alternative dimensionality reduction method to the PCA. Five components were selected to allow for a direct comparison to be made to the results of the PCA (*Appendix 1—figure 3*). ICA decomposition was computed on the *436 × 36* (patients x symptom variables) data matrix using Hyvarinen's (*Hyvärinen, 1999*) FastICA algorithm, executed via the 'icafast' package in R (*Helwig and Hong, 2013*). Data were mean-centered before decomposition. After decomposition, the source signal estimate ('ICA score') for all subjects for each independent component (IC) was regressed on to the neural data across patients, as with a priori and PCA scores, described in the methods section. The proportion of variance explained by each independent component (IC) is shown in *Appendix 1—figure 3A*. These percentages are comparable to those observed across all the PCs in *Figure 1C*. *Appendix 1—figure 3B* shows the correlations between subject scores across all five ICs and five PCs (N = 436). Notably, certain pairs of components between the two solutions appear to be highly similar and exclusively mapped to each other (*IC5* and *PC4; IC4* and *PC5*). The correspondence between PCs 1–3 and ICs 1–3 appears to be more distributed as opposed to following a one-to-one mapping, likely because PCs are orthogonal to each other whereas ICs are not necessarily orthogonal. For example, *PC3* appears to correspond to *IC2*, but is negatively related to *IC3*, suggesting that these two ICs are oblique to the PC and perhaps share behavioral symptom variation in the ICA solution that is explained by a single *PC3*. Of note, *PC3* (the 'Psychosis Configuration' axis) is a bi-directional axis whereby individuals who score either highly positively or highly negatively are highly symptomatic in different ways. The diverging nature of this axis may be not be captured well in the ICA solution, when orthogonality is not enforced. *Appendix 1—figure 12F* shows the correlations across parcels (N = 718 parcels) between the five IC maps (*Appendix 1—figure 12A-E*) and five PC maps. Again, the neural pattern resembles that in *Appendix 1—figure 12*, with certain pairs corresponding highly between the the ICA and PCA solutions. The coefficient maps for each of the five ICs are shown in *Appendix 1—figure 12*. To compare the ICA solution to the PCA solution, we performed a correlation between the individual PCA and ICA scores across subjects, as well as between parcel coefficients of the five PCA maps and five ICA maps across parcels.

### Note 2 - Establishing the reproducibility of the data-reduced symptom-neural mapping solution

Here, we expand on the specific details regarding the cross-validation and reproducibility of the symptom-neural mapping result described in *Figure 4*. Specifically, *Figure 4D* shows a summary of a fivefold bootstrapping symptom-neural map validation. Here, the patient sample was randomly split into five folds and each fold was held out once as the regression was performed between symptom scores and parcel-level GBC in the other four folds. The resulting GBC coefficient map was then correlated with the full sample map performed in all 436 patients. Notably, the correlations between maps for each of the *k*-fold runs and the full model reached effects between *r* = 0.8–0.9. The coefficient map for one such fold is shown in *Figure 4E* for and the correlation between the coefficients in this map and the coefficients of the full regression model (performed in all 436 subjects) is shown in *Figure 4F*. *Figure 4G* shows a similar summary for leave-one-site-out validation for the regression of all five PC dimensions and traditional symptom scales onto neural GBC. Here, patients from one of the sites were left out while a regression was performed in all remaining patients. In turn, this effect was correlated with the coefficient map from the full patient sample. Again, the maps are highly similar for all sites, showing that the relationships between behavior and neural data are not driven by any one site in particular. Here, the coefficient map for one example is shown in *Figure 4H* for PC3. Specifically, the results performed without Site three are shown because this is the site with the greatest number of patients and therefore may be the most likely to impact the final effect. The correlation between the coefficients in the resulting map without Site three and the coefficients of the full sample regression model are shown in the scatterplot in *Figure 4I*. Finally, we performed a stratified split-half replication. Here, the full sample of patients was randomly divided into two halves (H1 and H2) with the proportion of each diagnostic group (BPP, SADP, SZP) preserved within each half.

Then, PCA on the symptom data was computed in each half independently. As noted in the main text, this resulted in the 'observed' PCA values for H1 and H2. In turn, we used loadings from H1 PCA to compute the 'predicted' PC scores for subjects in H2, and vice versa. These 'predicted' and 'observed' PC scores were then independently regressed against parcellated GBC for subjects a given half. This yielded two maps for each PC: one for the 'predicted' PC scores based on H1 data; and one for the 'observed' PC scores from H1 data. These two symptom-neural maps were then correlated to evaluate reproducibility of the effects. This process was performed for both H1-to-H2 reproducibility as well as H2-to-H1 reproducibility. We performed 1000 runs of the entire split-half replication analysis for the PC1-5 maps (*Figure 4J*. Example data for *PC3* H1 and H2 are shown in *Figure 4K–M*). These results highlight that the behavioral PC-to-GBC mapping yields a highly reproducible 'brain-behavior' effect.

## Note 3 - Multivariate canonical correlation analysis of symptom and neural features in PSD

CCA solves for transformation matrices  and  such that the correlations between the linearly transformed 'latent'' matrices *U* and *V* are maximal (see *Materials and methods* for computational details). Matrix *U* contains the latent 'behavioral' subject scores for all CV and is a linear composite of behavioral feature matrix *B* scaled by the loadings in ; similarly, *V* is the latent neural score matrix and is the linear composite of neural features *N* transformed by . Each column in *U* and *V* is known as a canonical variate (CV); each corresponding pair of canonical variates (e.g. *U1* and *V1*) is referred to as a canonical mode. The correlations between the transformed latent matrices *U* and *V* are maximized (*Figure 5B*), under the constraint that the canonical modes are orthogonal and independent to each other. Hence the correlation between each subsequent pair of CVs is computed from the residuals of the previous pair, until the number of columns in the smaller of the two matrices is reached.

In the main text we show that our CCA solution is not reproducible. We also examined whether the size of the neural feature space affects the CCA solution. To this end, we computed CCA using: (i) 180 symmetrized cortical parcels; (ii) 359 bilateral subcortex-only parcels; (iii) 192 symmetrized subcortical parcels; (iv) 12 functional whole-brain networks, from the recently developed cortico-subcortical CAB-NP parcellation (*Glasser et al., 2016*; *Ji et al., 2019b*). We show results for subcortical (192 neural features) in *Appendix 1—figure 13*, as well as network-level (12 features neural features) in *Appendix 1—figure 14*. We evaluated CCA significance via permutation testing whereby patient order was randomly shuffled 5000 times (such that the association between behavioral and neural features was violated) and a CCA solution was computed for each shuffle; (see *Materials and methods*).

For each CCA solution, we examined how much symptom variance can be accounted for by the neural feature variance. Specifically, the correlation between behavioral features *B* and the latent neural matrix *V* (i.e. *N*) reflects the amount of variance in *B* that can be explained by the neural CVs in *V* (*Figure 5E*). In the main text, *Figure 5F* shows the proportion of behavioral variance explained by each of the neural CVs in a CCA performed between 180 neural features and all 36 symptom measures. *Figure 5G* shows the result for the five PC-derived symptom dimensions. Given that the symptom PCs accounted for ~50.93% of total symptom variance (see *Figure 1*), the percentages here were scaled to reflect the total proportion of behavioral variance explained by each CV. The amount of neural variance explained by each behavioral CV is shown in *Appendix 1—figure 16*.

Next, we examined the proportion of symptom variance explained by each neural canonical variate (CV). Because the PCs accounted for ~50.93% of total symptom variance (*Figure 1*), the percentages here were scaled accordingly. While the CCA using PC scores has fewer features than the one using item-level symptom measures, each neural CV explained more behavioral variance (*Figure 5G* versus 5F), suggesting that the PCs may capture neurally-relevant symptom variation with fewer features. Each CV has a pair of associated symptom and neural profiles, which can inform interpretation. This is shown for *CV3* in *Figure 5H–K*. Specifically, the latent scores for an example behavioral canonical variate (CV3) across diagnostic groups normalized to controls are shown in *Figure 5H*. The profile of PC loadings onto *CV3* are shown in *Figure 5I*. Full results are shown in *Appendix 1—figure 16*. CCA loadings are computed as the correlation of the latent matrix *U* (or *V*) and the input data matrix *B* (or *N*). These CCA loadings reflect the amount of variance in the original data that is

extracted by the canonical variates. The high negative loadings of *CV3* onto *PCs 1,4,5* versus the high positive loadings of *CV3* onto *PC3* in *Figure 5I* indicate that *CV3* captures complex symptom variation. This can also be seen by the projecting 36 symptom measure loadings from a given PC onto *CV3* (*Figure 5J*). Here the most positively loaded items were #7 'Delusions' (PANSS P1), #9 'Hallucinations' (PANSS P3), #21 'Somatic Concern' (PANSS G1) and #29 'Unusual Thought Content' (PANSS G9), while negatively loaded items include #16 'Poor Rapport' (PANSS N3), #19 'Lack of Spontaneity' (PANSS N6), and #28 'Uncooperativeness' (PANSS G8). Conversely, the corresponding neural factor loadings for *CV3* are shown in *Figure 5K*, illustrating GBC patterns associated with the *CV3* profile. Data for all five CVs are shown in *Appendix 1—figure 15*.

Interestingly, the cross-validation of the CCA effect was reproducible when using overlapping cross-validation strategies *Appendix 1—figure 17*. However, as noted in the main text, the CCA solution was not reproducible with out-sample cross-validation strategies even though the effect passed p<0.05 permutation testing. This highlights the importance of rigorously testing for out-of-sample reproducibility of brain-behavioral effects across independent sub-samples (*Helmer et al., 2020*).

## Note 4 - Establishing the Reproducibility of the CCA Solution

Here, we outline the details of the split-half replication for the CCA solution. Specifically, the full patient sample was randomly split (referred to as 'H1' and 'H2' respectively), while preserving the proportion of patients in each diagnostic group. Then, CCA was performed independently for H1 and H2. While the loadings for behavioral PCs and original symptom measures are somewhat similar (mean $r \approx 0.5$) between the two CCAs in each run, the neural loadings were not stable across H1 and H2 CCA solutions. Critically, CCA results did not perform well for leave-one-subject-out cross-validation (*Figure 5M*). Here, one patient was held out while CCA was performed using all data from the remaining 435 patients. The loadings matrices and from the CCA were then used to calculate the 'predicted' neural and behavioral latent scores for all five CVs for the patient that was held out of the CCA solution. This process was repeated for every patient and the final result was evaluated for reproducibility. As described in the main text, this did not yield reproducible CCA effects (*Figure 5M*).

Of note, CCA may yield higher reproducibility if the neural feature space were to be further reduced. As noted, our approach was to first parcellate the BOLD signal and then use GBC as a data-driven method to yield a neurobiologically and quantitatively interpretable neural data reduction, and we additionally symmetrized the result across hemispheres. Nevertheless, in sharp contrast to the PCA univariate feature selection approach, the CCA solutions were still not stable in the present sample size of $N = 436$. Indeed, a multivariate power analysis (*Helmer et al., 2020*) estimates that the following sample sizes will be required to sufficiently power a CCA between 180 neural features and five symptom features, at different levels of true canonical correlation ($r_true$):

- if $r_{true} = 0.1$: $n_{required} = 103,545$
- if $r_{true} = 0.3$: $n_{required} = 8,145$
- if $r_{true} = 0.5$: $n_{required} = 2,497$

To test if further neural feature space reduction may be improve reproducibility, we also evaluated CCA solutions with neural GBC parcellated according to 12 brain-wide functional networks derived from the recent HCP-driven network parcellation (*Ji et al., 2019b*). Again, we computed the CCA for all 36 symptom measures as well as five *PCs* (*Appendix 1—figure 14*). As with the parcel-level effects, the network-level CCA analysis produced significant results (for *CV1* when using 36 symptom measure scores and for all 5 CVs when using the five PC-derived scores). Here, the result produced much lower canonical correlations (~0.3–0.5); however, these effects (for *CV1*) clearly exceeded the 95% confidence interval generated via random permutations, suggesting that they may reflect the true canonical correlation. We observed a similar result when we evaluated CCAs computed with neural GBC from 192 symmetrized subcortical parcels and 36 symptoms or 5 PCs (*Appendix 1—figure 13*). In other words, data-reducing the neural signal to 12 functional networks likely averaged out parcel-level information that may carry symptom-relevant variance, but may be closer to capturing the true effect. Indeed, the power analysis suggests that the current sample size is closer to that needed to detect an effect with 12 vs. 5 features:

- if $r_{true} = 0.1$: $n_{required} = 8,855$
- if $r_{true} = 0.3$: $n_{required} = 696$
- if $r_{true} = 0.5$: $n_{required} = 213$

Note that we do not present a CCA conducted with parcels across the whole brain, as the number of variables would exceed the number of observations. However, the multivariate power analysis using 718 neural features and five symptom features estimates that the following sample sizes would be required to detect the following effects:

- if $r_{true} = 0.1$: $n_{required} = 421,608$
- if $r_{true} = 0.3$: $n_{required} = 33,166$
- if $r_{true} = 0.5$: $n_{required} = 10,168$

This analysis suggests that even the lowest bound of ~10 k samples exceeds the present available sample size by two orders of magnitude.

## Note 5 - Evaluating individual patients in discovery and replication samples via similarity to target brain-behavior spaces

We tested a hybrid neurobehavioral patient selection strategy for single *PC* symptom axes by first imposing a PC-based symptom threshold, followed by a target neural similarity threshold driven by the most highly predictive symptom-neural map features. Specifically, the 'neural similarity prediction index (NSPI)' computes a patient-specific Spearman's ρ between that patient's $\Delta GBC^{obs}$ and the group reference $\beta_{PC3} GBC^{obs}$ map using the maximally predictive $P = 39$ parcels (see *Appendix 1—figure 27* for whole-brain results and alternative similarity metrics). *Appendix 1—figure 26A* shows a significant relationship between each patient's *PC3* symptom score (X-axis) and the neural similarity index (Y-axis). In turn, *Appendix 1—figure 26B* shows binned results, which provides a visual intuition for patient segmentation across both the neural and behavioral indices (*Appendix 1—figure 26A*, right: binned by $\rho = 0.1$ and $PC3_{score} = 0.5$). For patients at either tail, the neurobehavioral relationship was robust. Conversely, patients with a low absolute *PC3* score showed a weak relationship with symptom-relevant neural features. This is intuitive because these individuals do not vary along the selected PC symptom axis.

*Appendix 1—figure 26C* shows the mean NSPI across subjects within each PC symptom bin along the X-axis. The resulting sigmoid captures that patients exhibit greater neural similarity if their PC symptom scores are more extreme. To evaluate if this relationship can yield a personalized patient selection, we computed the absolute NSPI (*Appendix 1—figure 26D*). The effect was approximated by a quadratic function, highlighting that patients with extreme *PC3* scores (either positive or negative) exhibited a stronger NSPI (i.e. personalized neural effects that strongly resembled the reference $(\beta_{PC3} GBC^{obs})^{P=39}$ map). *Appendix 1—figure 26E* shows the application of this neurobehavioral selection procedure, demonstrating that PSD patients with extreme *PC3* scores (defined at the top/bottom 10th percentile of the 'discovery' sample, $+2.17 < PC3_{score} < -2.41$) exhibit high NSPI values. We observed an inherent trade-off such that if the PC score threshold was raised then neural target similarity confidence goes up, but fewer patients will be selected. In the discovery PSD sample, all patients were accurately selected above the following neurobehavioral thresholds: $90^{th}\%tile < PC3_{score} < 10^{th}\%tile$ and $|\rho| > 0.4$ (*Appendix 1—figure 26E*, 34/436 patients selected, green line). We show consistent effects for when the selection was applied to *PC5* (*Appendix 1—figure 26F*; results for all PCs in *Appendix 1—figure 28*).

To test if the neurobehavioral selection is generalizable, we used an independent cross-diagnostic sample of 30 patients diagnosed with SZP and 39 diagnosed with obsessive-compulsive disorder (OCD) (*Appendix 1—figure 26G*, see **Materials and methods** and *Appendix 1—table 3* for details). Applying the 'discovery' selection thresholds yielded similar results for ~6% of the cross-diagnostic 'replication' sample for *PC3* (*Appendix 1—figure 26G*, full analyses in *Appendix 1—figure 27C*). Notably, no replication sample patients were selected along the neurobehavioral thresholds for *PC5* (*Appendix 1—figure 26H*). Although there are SZP patients in the replication cross-diagnostic sample, few scored highly on *PC5* and none met the neural similarity threshold, emphasizing that not all patients within the same DSM-based diagnosis will exhibit variation along the same neurobehavioral axis. Collectively, these results show that data-driven symptom scores can pinpoint

individual patients for whom their neural variation strongly maps onto a target neural reference map. These data also highlight that both symptom and neural information for an independent patient can be quantified in the reference 'discovery' BBS using their symptom data alone.

**Appendix 1—table 1.** Glossary of the key terms and abbreviations used in the study.

| Abbreviation/ Term | Definition |
|---|---|
| **General and Behavioral Terms** | |
| *PSD* | Psychosis spectrum disorder; or patients diagnosed with a PSD. |
| *BPP* | Bipolar disorder with psychosis; or patients diagnosed with BPP. |
| *SADP* | Schizoaffective disorder with psychosis; or patients diagnosed with SADP. |
| *SZP* | Schizophrenia with psychosis; or patients diagnosed with SZP. |
| *CON* | Control subjects. |
| *BACS* | Brief Assessment of Cognition in Schizophrenia. |
| *PANSS* | Positive and Negative Syndrome Scale. |
| *Symptom measures* | The 36 behavioral items from the BACS and PANSS. |
| *PCA, PC* | Principal component analysis, and principal component. |
| *ICA, IC* | Independent component analysis, and independent component. |
| **Neural Terms** | |
| *FC* | Functional connectivity (Fisher's r-to-Z transformed Pearson correlation values). |
| *GBC* | Global brain connectivity; computed for each brain location as the the mean FC across the whole brain. |
| $\beta_{PC}GBC$ | The beta coefficient map from a mass univariate regression of PC scores on to GBC across subjects. |
| $\beta_{PC}GBC^{obs}$ | The beta coefficient map from a mass univariate regression of observed (measured) symptom PC scores on to observed (measured) GBC across subjects. |
| $\beta_{PC}GBC^{pred}$ | The beta coefficient map from a mass univariate regression of predicted symptom PC scores on to observed (measured) GBC across subjects. |
| $GBC - PCA$ | A PCA performed on the neural GBC data across all N = 638 PSD and CON subjects. |
| $G\hat{B}C_{woPC1}$ | The GBC matrix reconstructed without the first PC of the GBC-PCA. |
| $G\hat{B}C_{woPC1-2}$ | The GBC matrix reconstructed without the first two PCs of the GBC-PCA. |
| $G\hat{B}C_{woPC1-3}$ | The GBC matrix reconstructed without the first three PCs of the GBC-PCA. |
| $G\hat{B}C_{woPC1-4}$ | The GBC matrix reconstructed without the first four PCs of the GBC-PCA. |
| **Multivariate Analyses Terms** | |
| *CCA* | Canonical correlation analysis. |
| *CV* | Canonical variate from a CCA solution. |
| $180\,vs.36\,CCA$ | The CCA computed between 180 neural features and 36 symptom measures. |
| $5180\,vs.5\,CCA$ | The CCA computed between 180 neural features and five symptom PCs. |
| $N, B$ | The input neural and behavioral data matrices in a CCA, respectively. |
| $\hat{N}, \hat{B}$ | The predicted neural and behavioral data matrices in the CCA cross-validation, respectively. |
| $\Theta, \Psi$ | The neural and behavioral transformation matrices in a CCA, respectively. |
| $U, V$ | The transfomed neural and behavioral data matrices in a CCA, respectively. |
| **Single-Subject Prediction Analyses Terms** | |
| $\Delta GBC$ | The difference between an individual's measured GBC map and the mean GBC map computed for the whole group; reflects individual differences in GBC, which may otherwise be conflated by the group mean. |
| $\beta_{PC}GBC^{obs}_{N-1}$ | The beta coefficient map computed across $N - 1$ subjects, using observed symptom PC and GBC data. |

*Continued on next page*

*Appendix 1—table 1 continued*

| Abbreviation/ Term | Definition |
|---|---|
| $S^{obs}$ | The vector of observed symptom scores for a single PC (i.e. from a PCA computed using symptom measures), with length N-1 (as subject i is left out). |
| $S^{pred}$ | The predicted symptom PC for subject $i$, using the loadings from a leave-one-out PCA computed on N-i subjects. |
| $dpGBC$ | Dot product between a subject's ΔGBC map and a $\beta_{PC}$GBC map. |
| $dpGBC_i^{obs}$ | Dot product between a subject $i$'s ΔGBC map and the $\beta_{PC}GBC_{N-1}^{obs}$ map computed without that subject. |
| $dpGBC_i^{pred}$ | The predicted dpGBC for a subject $i$, predicted from a regression model using the dpGBCobs from all other subjects and the predicted Spred for subject $i$. |
| $P_{select}$ | The number of parcels in the subset of the brain map used in a particular analysis. $P_{select} = 718$ for the full whole-brain map. |
| $BBS$ | Brain-Behavior Space; the symptom and neural mapping for a particular dimension. |

**Appendix 1—table 2.** Clinical and demographic details for the BSNIP sample.
Clinical and demographic measures are shown at the group level for the BSNIP sample used for initial 'discovery' analyses. Means and standard deviations were computed for each of the diagnostic groups. Abbreviations: CON, controls; PSD, psychosis spectrum disorder patients; BPP, bipolar disorder with psychosis; SADP, schizoaffective disorder with psychosis; SZP, schizophrenia; CPZ, chlorpromazine; PANSS, Positive and Negative Syndrome Scale for Schizophrenia, BACS, Brief Assessment of Cognition in Schizophrenia.

| Characteristic | CON (N=202) | | PSD (N=436) | | BPP (N=150) | | SADP (N=119) | | SZP (N=167) | |
|---|---|---|---|---|---|---|---|---|---|---|
| | Mean | SD | Mean | SD | Mean | SD | Mean | SD | Mean | SD |
| Age (years) | 37.18 | 12.18 | 35.33 | 12.31 | 36.29 | 12.94 | 35.86 | 11.87 | 34.16 | 11.99 |
| Sex (% male) | 42.08 | - | 48.62 | - | 31.33 | - | 44.54 | - | 67.07 | - |
| Parental Education (years) | 13.35 | 3.42 | 13.67 | 3.43 | 14.16 | 3.44 | 13.05 | 3.95 | 13.62 | 2.93 |
| Participant's Education | 14.76 | 2.27 | 13.47 | 2.34 | 14.22 | 2.31 | 13.25 | 2.22 | 12.93 | 2.28 |
| Handedness (% right) | 85.64 | - | 85.78 | - | 83.33 | - | 88.24 | - | 86.23 | - |
| Signal-to-Noise (SNR) | 231.03 | 82.70 | 218.34 | 93.44 | 229.78 | 100.09 | 230.93 | 93.83 | 198.10 | 83.13 |
| % Frames Flagged | 2.81 | 3.99 | 6.17 | 9.09 | 6.41 | 9.67 | 5.32 | 9.32 | 6.57 | 8.35 |
| Medication (CPZ equivalents) | - | - | 443.20 | 402.92 | 318.61 | 321.06 | 514.43 | 462.70 | 490.64 | 395.76 |
| PANSS Positive Symptoms | 7.03 | 0.30 | 15.66 | 5.35 | 12.87 | 4.35 | 18.19 | 5.20 | 16.37 | 5.15 |
| PANSS Negative Symptoms | 7.01 | 0.16 | 14.59 | 5.10 | 12.01 | 3.66 | 15.55 | 4.52 | 16.22 | 5.68 |
| PANSS General Psychopathology | 16.04 | 0.36 | 31.45 | 8.67 | 28.72 | 8.20 | 34.57 | 8.70 | 31.67 | 8.30 |
| PANSS Total Psychopathology | 30.08 | 0.61 | 61.67 | 16.32 | 53.59 | 13.69 | 68.22 | 15.72 | 64.26 | 16.04 |
| BACS Cognitive Score (Z) | 0.02 | 1.10 | −1.22 | 1.20 | −0.83 | 1.19 | −1.30 | 1.10 | −1.52 | 1.19 |

**Appendix 1—table 3.** Scan acquisition parameters for the BSNIP dataset across sites.
Scan acquisition details are shown for the 6 BSNIP acquisition sites. Adapted from *Meda et al., 2015*.

| BOLD | TR (ms) | TE (ms) | FA (deg.) | Slices (N) | Slice order | Matrix (mm) | Voxel (mm) | Vendor |
|---|---|---|---|---|---|---|---|---|
| Baltimore | 2210 | 30 | 70 | 36 | I-A | 64 × 64 | 3.4 × 3.4 × 3 | Siemens Trio |
| Hartford | 1500 | 27 | 70 | 29 | S-A | 64 × 64 | 3.4 × 3.4 × 5 | Siemens Allegra |

*Continued on next page*

*Appendix 1—table 3 continued*

| BOLD | TR (ms) | TE (ms) | FA (deg.) | Slices (N) | Slice order | Matrix (mm) | Voxel (mm) | Vendor |
|---|---|---|---|---|---|---|---|---|
| Detroit | 1570/1720 | 22 | 60 | 29 | S-A | 64 × 64 | 3.4 × 3.4 × 4 | Siemens Trio |
| Dallas | 1500 | 27 | 60 | 29 | S-A | 64 × 64 | 3.4 × 3.4 × 4 | Philips |
| Chicago | 1775 | 27 | 60 | 29 | S-A | 64 × 64 | 3.4 × 3.4 × 4 | GE Signa HDX |
| Boston | 3000 | 27 | 60 | 30 | S-A | 64 × 64 | 3.4 × 3.4 × 5 | GE Signa HDX |

| T1w | TR (ms) | TE (ms) | FA (deg.) | Slices (N) | Slice order | Matrix (mm) | Voxel (mm) | Vendor |
|---|---|---|---|---|---|---|---|---|
| Baltimore | 2300 | 2.91 | 9 | 160 | — | 256 × 240 | 1 × 1 × 1.2 | Siemens Trio |
| Hartford | 2300 | 2.91 | 9 | 160 | — | 256 × 240 | 1 × 1 × 1.2 | Siemens Allegra |
| Detroit | 2300 | 2.94 | 9 | 160 | — | 256 × 240 | 1 × 1 × 1.2 | Siemens Trio |
| Dallas | 6.6 | 2.8 | 8 | 170 | — | 256 × 256 | 1 × 1 × 1.2 | Philips |
| Chicago | 6.98 | 2.84 | 8 | 166 | — | 256 × 256 | 1 × 1 × 1.2 | GE Signa HDX |
| Boston | 6.98 | 2.84 | 8 | 166 | — | 256 × 256 | 1 × 1 × 1.2 | GE Signa HDX |

**Appendix 1—table 4.** Clinical and demographic details for the independent replication cross-diagnostic sample.

Clinical and demographic measures are shown at the group level for the independent clinical sample used for replication analyses presented in main text (*Figure 7*). Means and standard deviations were computed for each of the diagnostic groups. Abbreviations: CON, controls; PTT, all patients; OCD, obsessive-compulsive disorder; SZP, schizophrenia; PANSS, Positive and Negative Syndrome Scale for Schizophrenia, BACS, Brief Assessment of Cognition in Schizophrenia.

| Characteristic | CON (N=34) | | PTT (N=69) | | OCD (N=39) | | SZP (N=30) | |
|---|---|---|---|---|---|---|---|---|
| | Mean | SD | Mean | SD | Mean | SD | Mean | SD |
| Age (years) | 29.91 | 8.11 | 33.7 | 12.89 | 32.23 | 12.32 | 35.63 | 13.63 |
| Sex (% male) | 61.7 | - | 59.4 | - | 46.1 | - | 76.7 | - |
| Parental Education (years) | 15.05 | 3.31 | 13.98 | 3.45 | 14.02 | 3.78 | 13.93 | 3.01 |
| Participant's Education | 16.37 | 2.34 | 14.16 | 2.12 | 15.22 | 1.95 | 12.79 | 2.34 |
| Handedness (% right) | 93.9 | - | 91.3 | - | 84.6 | - | 100.0 | - |
| Signal-to-noise (SNR) | 53.45 | 5.59 | 51.86 | 5.63 | 52.29 | 5.28 | 51.33 | 6.08 |
| % Frames Flagged | 0.21 | 0.53 | 7.27 | 15.53 | 5.11 | 13.81 | 10.07 | 17.37 |
| PANSS Positive Symptoms | 7.58 | 1.06 | 11.40 | 4.23 | 8.97 | 2.74 | 14.56 | 6.16 |
| PANSS Negative Symptoms | 7.75 | 1.57 | 11.06 | 3.59 | 8.80 | 2.21 14.00 | 5.39 | |
| PANSS General Psychopathology | 32.62 | 2.62 | 48.80 | 10.96 | 42.71 | 9.07 56.72 | 13.42 | |
| PANSS Total Psychopathology | 47.95 | 5.25 | 71.26 | 18.78 | 60.48 | 14.02 85.28 | 24.97 | |
| BACS Cognitive Score (Z) | 0.00 | 1.00 | −0.67 | 1.45 | −0.07 | 1.03 | −1.45 | 1.55 |

**Appendix 1—table 5.** Summary of the key questions, results, and supporting data in this study.

| Question | Results | Corresponding Figures |
|---|---|---|
| Can data-reduction methods (e.g. principal component analysis, PCA) reliably map symptom axes across PSD that include both canonical symptoms and cognitive deficits? | The PCA on 36 behavioral measures and 436 PSD patients produced five significant components (PCs) spanning key symptom domains and oblique to conventional symptom factors and diagnostic categories. | *Figure 1*<br><br>*Appendix 1—figure 2–3* |
| Are the derived dimensionality-reduced symptom axes in PSD stable and replicable? | The behavioral PCA solution was stable across split-half replication, leave-site-out, and k-fold cross-validations, and not driven by medication effects. | *Figure 2*<br><br>*Appendix 1—figure 4, 5, 6, 7* |
| Do dimensionality-reduced symptom axes yield stronger statistical neural maps across PSD, relative to conventional scales & diagnoses? | PC symptom scores yielded a much stronger statistical effect with neural features (as measured by global brain connectivity, GBC) across patients than did conventional symptom scores. | *Figure 3*<br><br>*Appendix 1—figures 7, 8, 9, 10* |
| Are the PC brain-behavior maps stable and replicable? | PC brain-behavior maps are stable and reproducible across split-half, leave-site-out and k-fold cross-validations. | *Figure 4*<br><br>*Appendix 1—figure 11–12* |
| Can a multivariate approach (e.g. canonical correlation analysis, CCA) be used to derive a stable neuro-behavioral mapping? | CCA produced a stable within-sample solution, but performed poorly on out-of-sample cross-validations. | *Figure 5*<br><br>*Appendix 1—figures 13, 14, 15, 16, 17, 18* |
| 8Is the computed symptom-to-neural mapping actionable for individual patient selection? | A majority of neural variance in PSD patients is not symptom-relevant. However, brain-behavioral maps can be optimized to select specific patients along PC axes. This method can also be applied to select patients from independent datasets. | *Figures 6* and *7*<br><br>*Appendix 1—figures 24, 25, 26, 27, 28* |
| 8Can we inform molecular mechanism and treatment decisions by relating patient-specific brain-behavioral maps to independently-acquired pharmacological neuroimaging maps? | Pharmacological neuroimaging maps targeting mechanisms of interest (e.g. ketamine and LSD) can be used as independent benchmarks against which patient-specific brain-behavioral maps can be evaluated. | *Figure 7*<br><br>*Appendix 1—figure 29* |
| 9Can we inform molecular mechanism and potential novel therapeutic targets by relating brain-behavior maps to gene expression maps? | Gene expression maps for genes implicated in PSD neuropathology (e.g. serotonin and GABA receptors, interneuron markers) can inform mechanism and potential novel targets relating to specific axes of neuro-behavioral variation. | *Figure 8*<br><br>*Appendix 1—figure 30* |

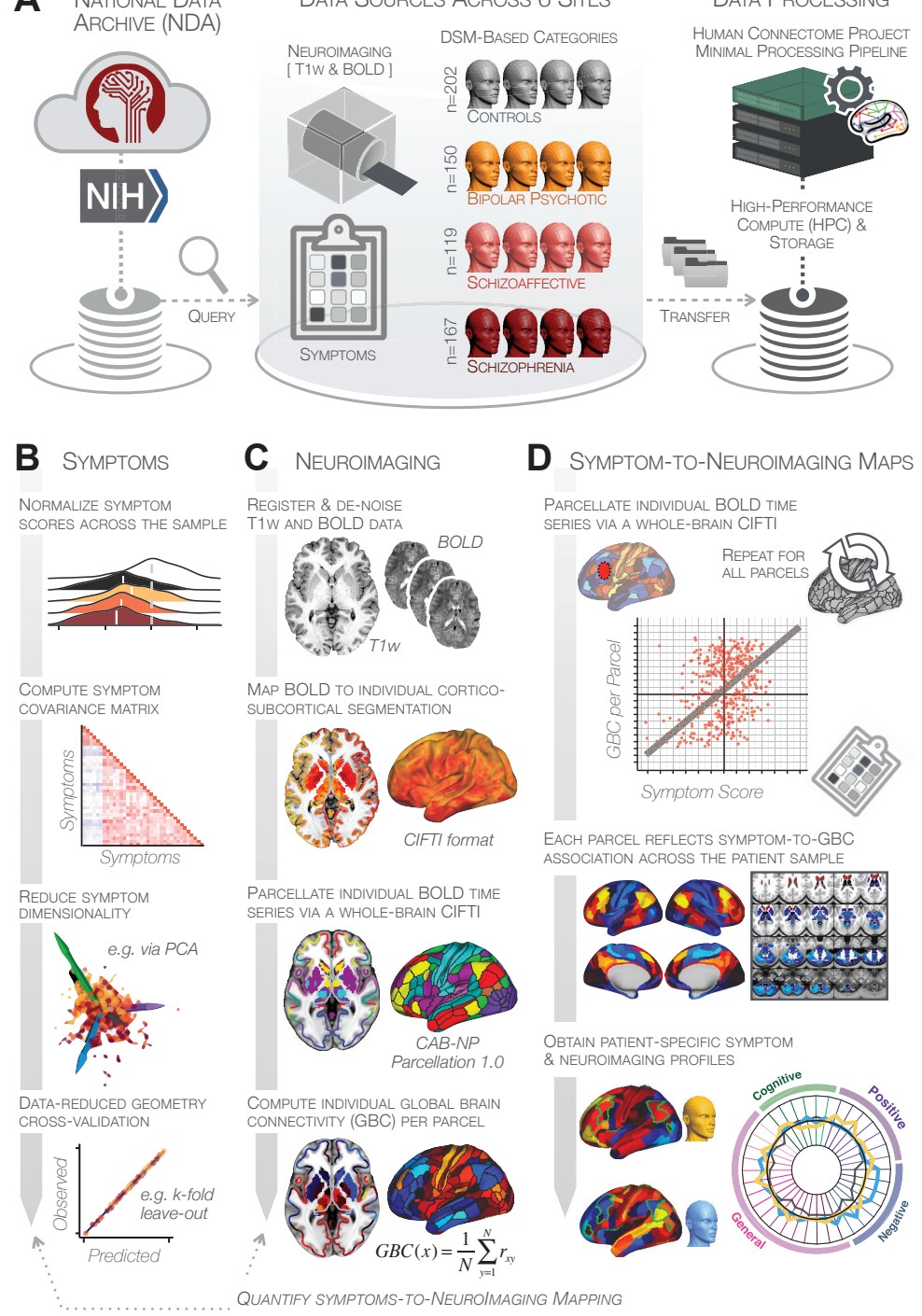

**Appendix 1—figure 1.** Study workflow used study to quantify shared neural and behavioral variation in individuals diagnosed with PSD. (**A**) Data from the BSNIP study were acquired from the National Institute of Health Data Archive (NDA). T1-weighted structural and resting-state BOLD neuroimaging data were obtained for a total of 638 individuals (202 controls, 150 patients with a diagnosis of bipolar disorder with psychosis, 119 with a diagnosis of schizoaffective disorder, and 167 patients with schizophrenia). Data were processed through the Human Connectome Project's (HCP) Minimal Preprocessing Pipeline with modifications made for 'legacy' BOLD and T1w data,

*Appendix 1—figure 1 continued on next page*

which are now featured as a standard option in the HCP pipelines provided by our team (https://github.com/Washington-University/HCPpipelines/pull/156) using Yale High Performance Computing clusters. (**B**) Symptom data were first normalized across the sample. The correlation matrix across all 436 PSD patients and 36 symptom measures was computed followed by dimensionality reduction (e.g. using PCA or ICA). The dimensionality-reduced solution was then cross-validated to assess stability and reproducibility across sites, *k*-fold cross-validations, leave-subject-out and split-half approaches. (**C**) In parallel, all neuroimaging data were processed as noted above. T1w and resting-state BOLD images were preprocessed using a modified version of the HCP Minimal Preprocessing Pipeline, including individual-subject registration of structural and function data, de-noising, and mapping of BOLD data on a hybrid surface-volume cortico-subcortical format (Connectivity Informatics Technology Initiative [CIFTI] format, see *Materials and methods* for details). After registration to a standard CIFTI template BOLD data were parcellated at the individual subject level using the Cole-Anticevic Brain-wide Network Partition (CAB-NP), which is a functionally-defined network and parcel-level partition in the CIFTI space encompassing both cortex and subcortex (*Glasser et al., 2016*; *Ji et al., 2019b*). Lastly, a global brain connectivity (GBC) map for each subject was computed by taking the mean functional connectivity of each parcel with all other parcels in the brain at the single subject level. (**D**) After symptom and neural data were fully processed in tandem, the symptom-to-neuroimaging mapping was quantified across subjects. Specifically, the relationship between data-reduced symptom scores and GBC was computed for each parcel across all patients. This produced a group-level symptom-neural map, which was subsequently cross-validated using leave-site-out, *k*-fold, leave subject-out as well as split-half approaches to assess reproducibility of the effect. Finally, the stable symptom-neural mapping result was further feature-optimized for single-subject prediction. This yielded a set of parcels for quantifying patient-specific symptom-neural prediction based on a cross-validated group reference map as well as comparison of the selected parcels with independent molecular neuroimaging maps (i.e. pharmacological [*Anticevic et al., 2012a*; *Preller et al., 2018*] and gene expression maps [*Burt et al., 2018*]).

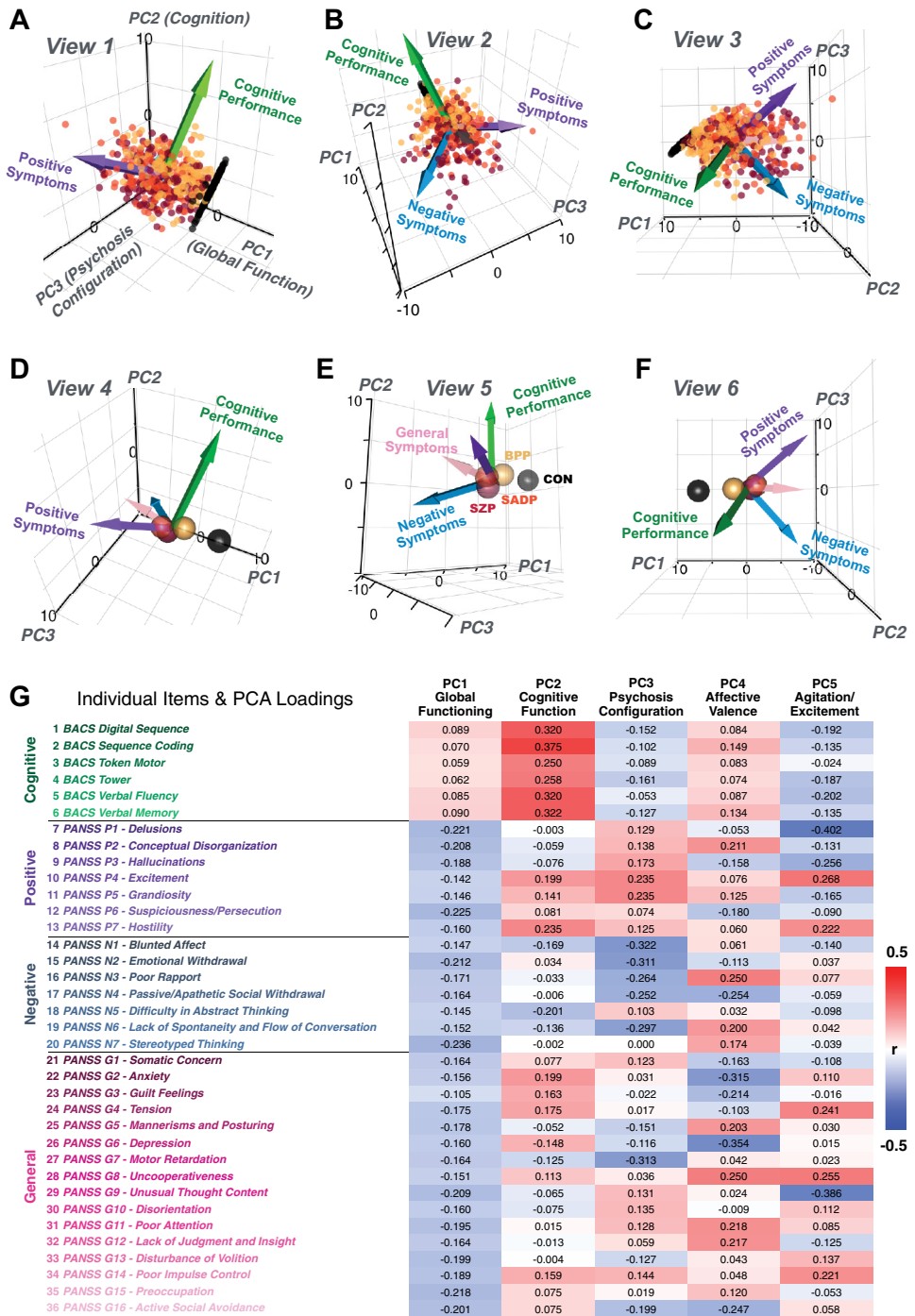

**Appendix 1—figure 2.** Alternative views of the behavioral PCA triplot. (**A–C**) Alternative views of the triplot in **Figure 1F** showing the relationship between the three principal axes of variation in behavior and standard clinical symptom factors. Each point represents an individual subject projected into the geometry defined by the first three principal components (PC). Vectors show the projections of standard symptoms factors. (**D–F**) Alternative views of the triplot in panels A–C, where each sphere represents the mean of each a priori clinical group. Vectors show the projections of standard symptoms factors [PANSS positive (purple), negative (blue), general (pink) symptoms and BACS cognitive performance (green)]. BPP, bipolar disorder; SADP, schizoaffective disorder; SZP, schizophrenia; CON, controls. (**G**) Heatmap of the loadings of each of the 36 symptom

*Appendix 1—figure 2 continued on next page*

*Appendix 1—figure 2 continued*

measures on the five significant PCs (also seen in radarplot in *Figure 1E*). Positive loadings are indicated in red; negative loadings are shown in blue. Each PC is named based on its most strongly loaded items.

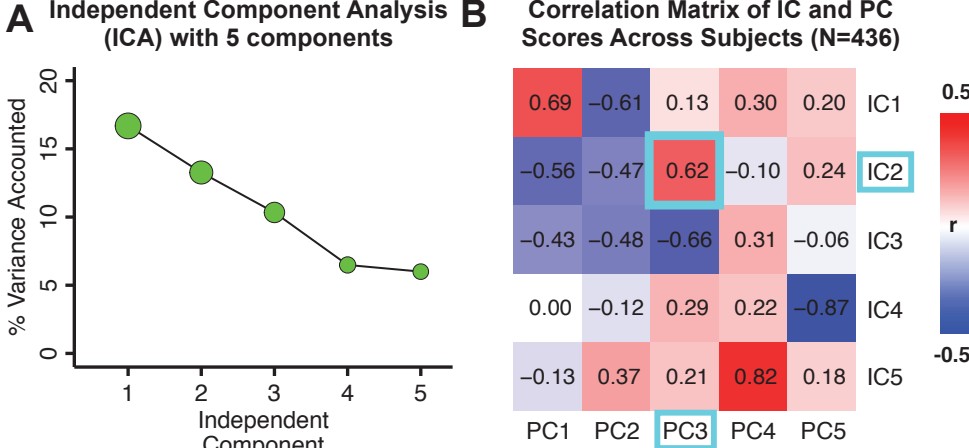

**Appendix 1—figure 3.** Independent component analysis (ICA) as an alternative method of dimensionality-reduction of symptom data. (**A**) Screeplot showing the total proportion of variance explained by each independent component (IC) in a five-component solution performed across all 36 behavioral measures in 436 patients. The size of each point is proportional to the variance explained by that IC. (**B**) Correlation matrix showing correlations of individual subject scores for the five significant principal components (PCs) from the PCA solution shown in *Figure 1* and the five ICs from the ICA solution, across all 436 subjects. Neural maps for each IC are shown in *Appendix 1—figure 12*.

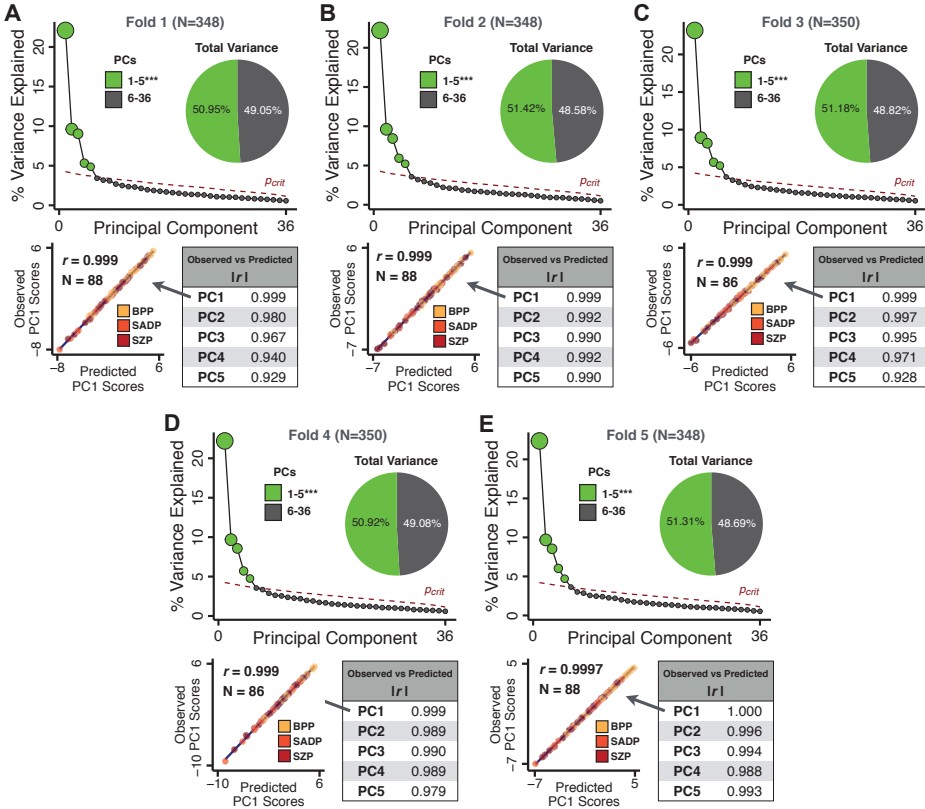

**Appendix 1—figure 4.** k-Fold cross-validation for symptom-driven principal component analysis (PCA). These results show a fivefold cross-validation analysis to test the stability of the PCA solution. The full patient sample was first randomly split into five sets and patients were randomly assigned to one of five subsets. Each subset of patients was then used as an independent 'test sample' in a PCA that was derived from the other four subsets. Screeplot shows proportion of variance explained by each of the PCs in a PCA of all 36 behavioral measures, excluding a subset of 88 patients. The number of significant PCs determined via a permutation test and the total proportion of variance explained by these PCs are all comparable to the full model shown in the main text. To obtain a 'predicted' PC score for the 88 patients in the excluded subset, the loadings from the model obtained from the other 348 patients were used. The 'observed' PC scores are the scores from the full model of the same 88 patients. The scatterplot shows that the predicted and observed scores for *PC1* are highly correlated (*r* = 0.999), suggesting that the PCA solution is stable and predictive at the individual-subject level. Similarly, predicted and observed scores are highly correlated for all five PCs. (**B–E**) The results of the PCA are also highly comparable and predictive for the other four folds.

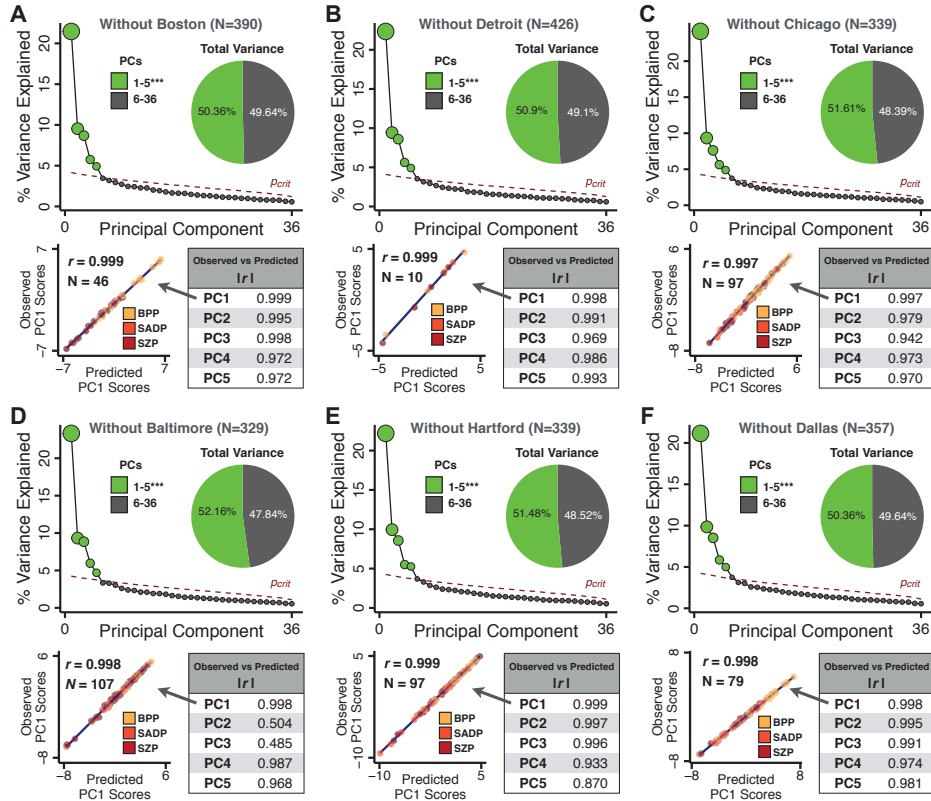

**Appendix 1—figure 5.** Leave-site-out cross-validation for symptom-driven principal component analysis (PCA). These results show a sixfold leave-site-out cross-validation analysis to test the stability of the PCA solution when a given site is excluded from the model. The full patient sample was first split into six sets according to data collection site. Each held-out site was then used as an independent 'test sample' in a PCA that was derived from the other five sites. (**A**) Proportion of variance explained by each of the PCs in a PCA of all 36 behavioral measures, excluding one of the six sites at which data was collected (here we excluded the Boston site). The number of significant PCs determined via a permutation test and the total proportion of variance explained by these PCs are all comparable to the full model shown in *Figure 1C*. To obtain a 'predicted' PC score for the 46 patients in the excluded site, the loadings from the model obtained from the other 390 patients were used. The 'observed' PC scores are the scores from the full model of the same 88 patients. The scatterplot shows that the predicted and observed scores for *PC1* are highly correlated (*r* = 0.999), suggesting that the PCA solution is stable and robustly predictive at the individual patient level. Predicted and observed scores are highly correlated for all five PCs. (**B–F**) The results of the PCA are also highly comparable for the other five sites, suggesting that possible site differences in evaluating patient symptoms or patient sample composition are not impacting the obtained PCA solutions.

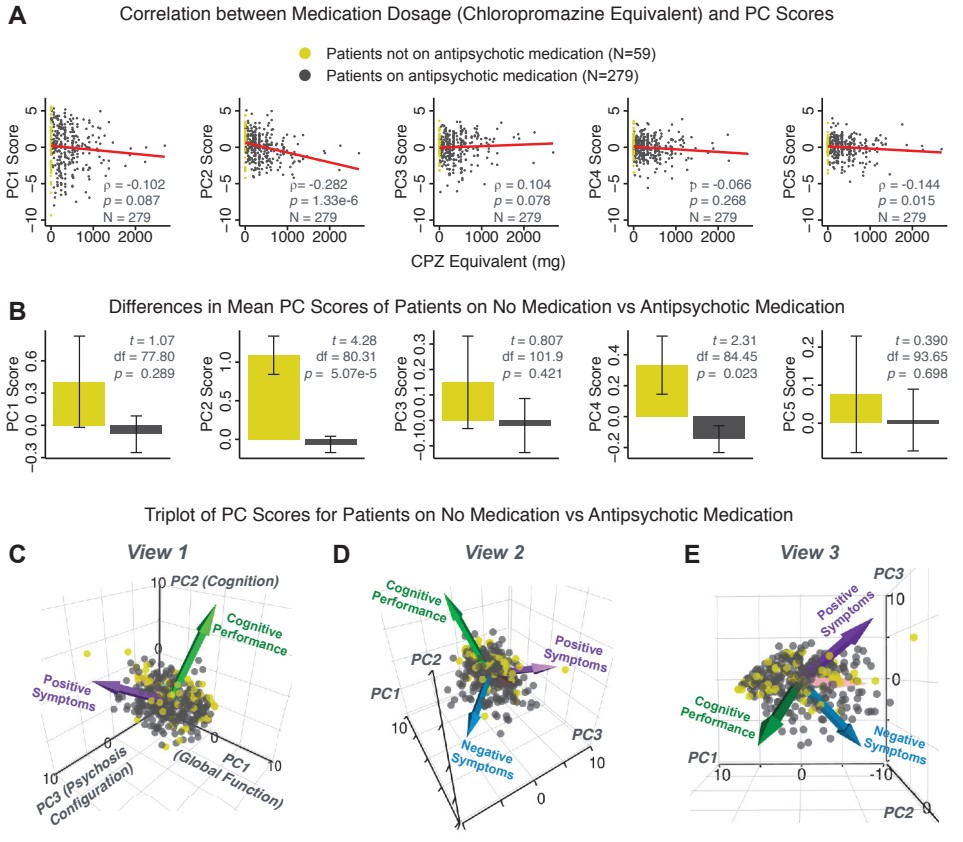

**Appendix 1—figure 6.** PCA solution is not driven by medication status or dosage. Antipsychotic medication dosages were available for N = 338 out of 436 PSD patients, including 59 patients not on antipsychotic medication. Antipsychotic dosages were converted to chlorpromazine (CPZ) equivalents (*Lencer et al., 2015*). (**A**) Spearman's ρ between medication dosage (CPZ equivalents) and PC scores for each of the five significant principal components (PCs), for medicated patients (gray points). Patients not on antipsychotic medication are also shown (yellow points, CPZ = 0 mg); however, they were not included in the calculation of Spearman's ρ as they contain no rank information. (**B**) Bar plots show the mean PC scores of unmedicated (yellow) versus medicated (gray) PSD patients for each of the five PCs. Error bars show standard error of the mean. Note that only PC2 'Cognitive Performance' scores appear to show a significant relationship with medication consistently; this could be because antipsychotic medication (particularly first-generation antipsychotics) are related to symptom variance across some PCA-derived symptom dimensions but do not effectively treat cognitive deficits. Also, cognitive deficits (such as reaction time and fluency) may be exacerbated due to the neuroleptic effects of first-generation antipsychotics. (**C–E**) Alternative views of the triplot (as seen in *Appendix 1—figure 2A–C*) showing the relationship between the three principal axes of variation in symptoms and aggregate scores from the PANSS and BACS symptom scales (i.e. vectors show the projections of standard symptoms factors). Each point represents an individual patient projected into the geometry defined by the first three principal components (PC). Each patient is colored according to medication status (yellow = unmedicated, gray = with medication). As evident from this plot there is no apparent clustering of patients according to their medication status in the 3D PCA-derived space.

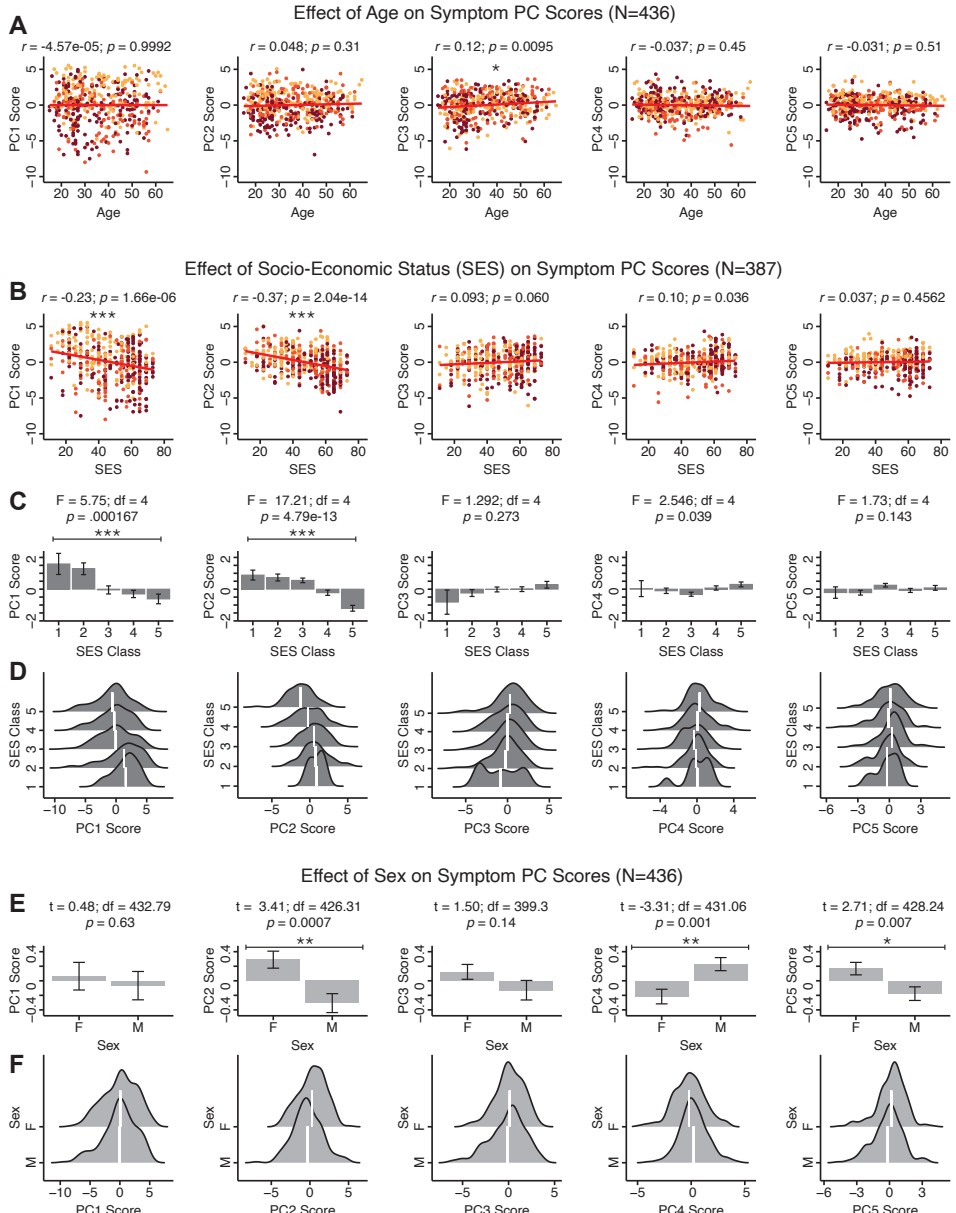

**Appendix 1—figure 7.** Effects of age, socio-economic status, and sex on symptom PCA solution. (**A**) Correlations between symptom PC scores and age (years) across N = 436 PSD. Pearson's correlation value and uncorrected p-values are reported above scatterplots. After Bonferroni correction, we observed a significant positive relationship between age and *PC3* score. This may be because older patients have been ill for a longer period of time and exhibit more severe symptoms along the positive *PC3* dimension. (**B**) Correlations between symptom PC scores and socio-economic status (SES) as measured by the Hollingshead Index of Social Position (*Hollingshead, 1975*), across N = 387 PSD with available data. The index is computed as (Hollingshead occupation score * 7) + (Hollingshead education score * 4); a higher score indicates lower SES (*Padmanabhan et al., 2015*). We observed a significant negative relationship between Hollingshead index and *PC1* and PC2 scores. Lower *PC1* and *PC2* scores indicate poorer general functioning and cognitive performance respectively, which is consistent with higher Hollingshead indices (i.e. lower-skilled jobs or unemployment and fewer years of education). (**C**) The Hollingshead index can be split into five classes, with one being the highest and five being the lowest SES class

*Appendix 1—figure 7 continued on next page*

*Appendix 1—figure 7 continued*

(*Hollingshead, 1975*). Consistent with (**B**) we found a significant difference between the classes after Bonferroni correction for *PC1* and *PC2* scores. (**D**) Distributions of PC scores across Hollingshead SES classes show the overlap in scores. White lines indicate the mean score in each class. (**E**) Differences in PC scores between (M)ale and (F)emale PSD subjects. We found a significant difference between sexes in *PC2 – Cognitive Functioning*, *PC4 – Affective Valence*, and *PC5 – Agitation/Excitement* scores. (**F**) Distributions of PC scores across M and F subjects show the overlap in scores. White lines indicate the mean score for each sex.

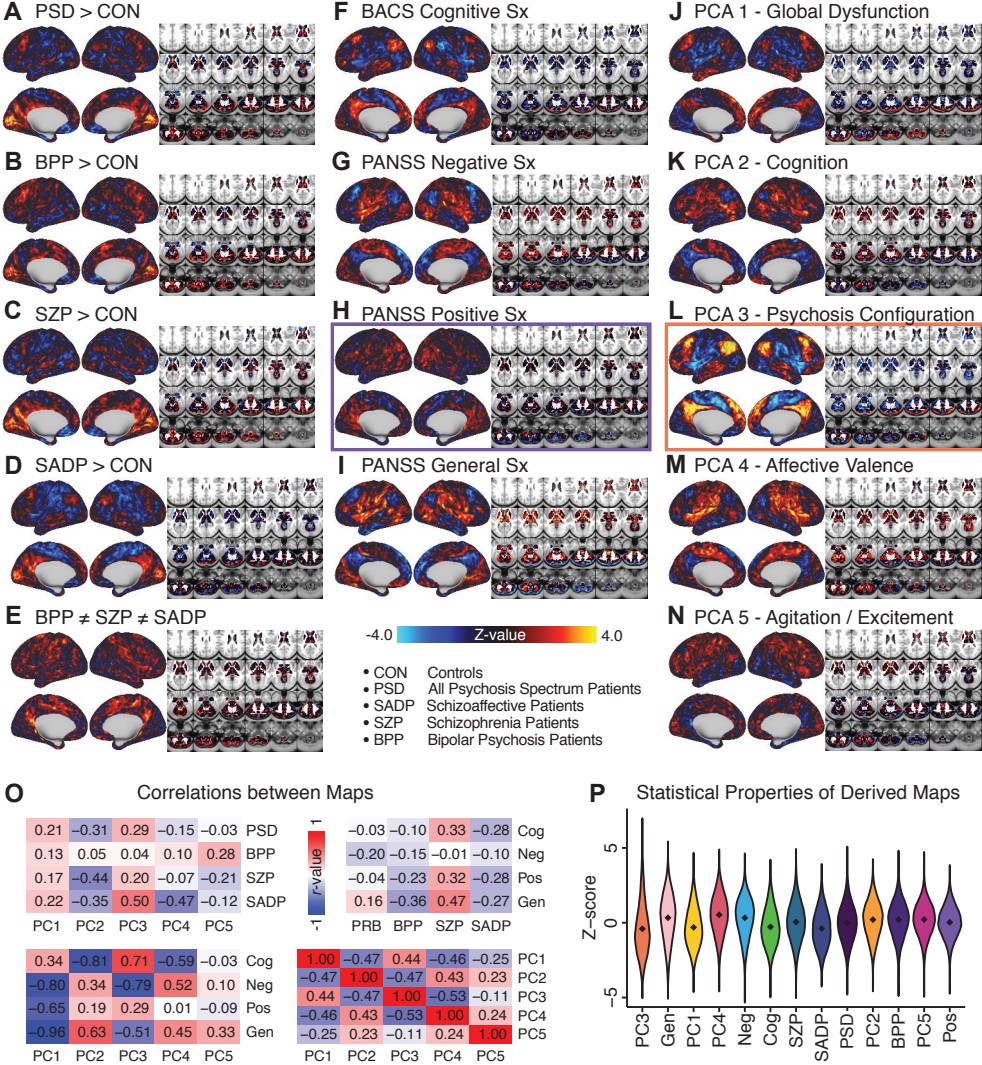

**Appendix 1—figure 8.** Similarity across a priori, categorical and PCA-derived brain-behavioral GBC maps. Z-scored maps of t-test for the difference in group mean GBC between traditional diagnostic groups: (**A**) all patients (PSD) versus all healthy controls (CON); (**B**) patients with bipolar disorder (BPP) versus CON; (**C**) patients with schizophrenia (SZP) versus CON; (**D**) patients with schizoaffective disorder (SADP) versus CON. (**E**) Z-scored map of the F-test for the difference in group mean GBC between patients in all three diagnostic groups (BPP, SZP, SADP). Z-scored map of the regression against GBC, across all patients (PSD), of traditional symptom/behavioral scales: (**F**) BACS cognitive composite performance score; (**G**) PANSS total negative symptom score; (**H**)

*Appendix 1—figure 8 continued*

PANSS total positive symptom score; (**I**) PANSS total general symptom score. Z-scored map of the regression against GBC, across all patients, of data-derived behavioral dimension scores: (**J**) *PC1* score; (**K**) PC2 score; (**L**) *PC3* score; (**M**) *PC4* score; (**N**) *PC5* score. (**O**) Correlation matrices showing the similarity between brain-behavioral maps in A-N. (**P**) Violin plots of the distribution of Z-values in all phenotype maps. Note that although there are strong correlations between the PC maps and the a priori symptom maps, the statistical properties of some of the PC maps are improved (i.e. the range of Z-scores is greater, with more extreme values), suggesting a stronger mapping between neural and behavioral variation.

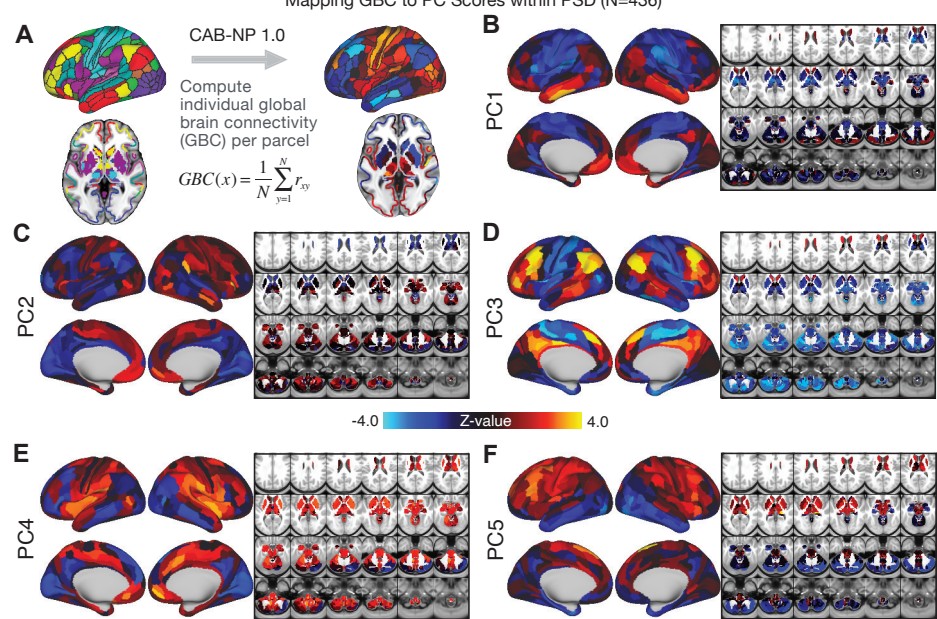

**Appendix 1—figure 9.** Parcellated symptom-neural GBC maps across all PSD patients derived from PCA dimensionality reduction of symptom measures. (**A**) All of the maps shown here were parcellated at the single patient level using the Cole-Anticevic Brain Network Parcellation (CAB-NP) parcellation (*Ji et al., 2019b*), which defines functional networks and regions across cortex and subcortex that leveraged the Human Connectome Project's Multi-Modal Parcellation (MMP1.0) (*Glasser et al., 2016*; *Ji et al., 2019b*). The final published CAB-NP 1.0 parcellation solution can be visualized via the Brain Analysis Library of Spatial maps and Atlases (BALSA) resource (https://balsa. wustl.edu/rrg5v) and downloaded from the public repository (https://github.com/ColeLab/ ColeAnticevicNetPartition). (**B–F**) Relationships across all patients (N = 436) at each parcel location between global brain connectivity (GBC) and PC score for each of the significant PCs. Values shown in each brain parcel is the Z-scored regression coefficient of PC score on to parcel GBC, across all 436 subjects.

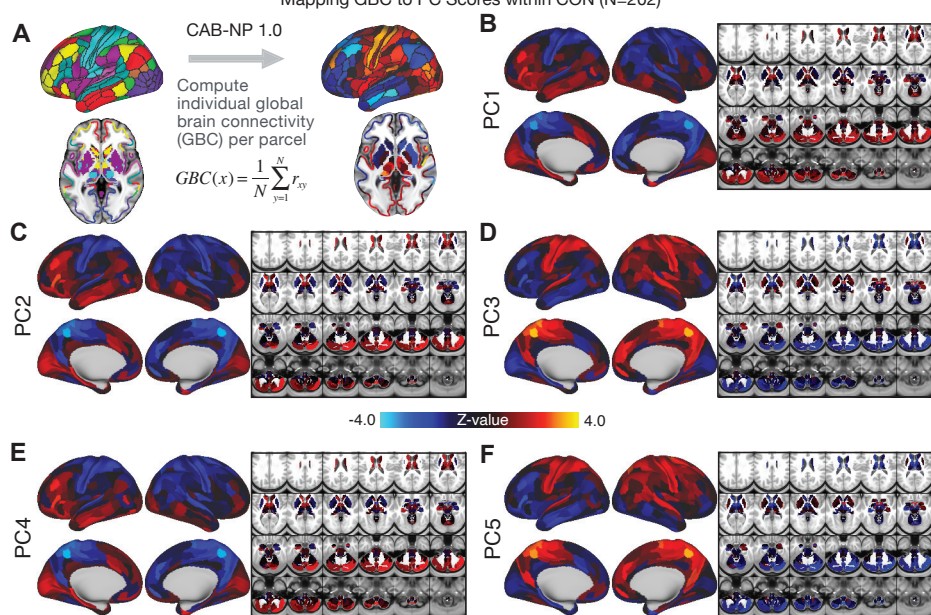

**Appendix 1—figure 10.** Parcellated symptom-neural GBC maps across all CON subjects derived from PCA dimensionality reduction of PSD symptom measures. (**A**) As with PSD maps, all of the maps shown here were parcellated at the single patient level using the Cole-Anticevic Brain Network Parcellation (CAB-NP) parcellation (*Ji et al., 2019b*), which defines functional networks and regions across cortex and subcortex that leveraged the Human Connectome Project's Multi-Modal Parcellation (MMP1.0) (*Glasser et al., 2016*; *Ji et al., 2019b*). The final published CAB-NP 1.0 parcellation solution can be visualized via the Brain Analysis Library of Spatial maps and Atlases (BALSA) resource (https://balsa.wustl.edu/rrg5v) and downloaded from the public repository (https://github.com/ColeLab/ColeAnticevicNetPartition). (**B–F**) Relationships across all controls (N = 202) at each parcel location between global brain connectivity (GBC) and PC score for each of the significant PCs. Values shown in each brain parcel is the Z-scored regression coefficient of PC score on to parcel GBC, across all 202 control subjects.

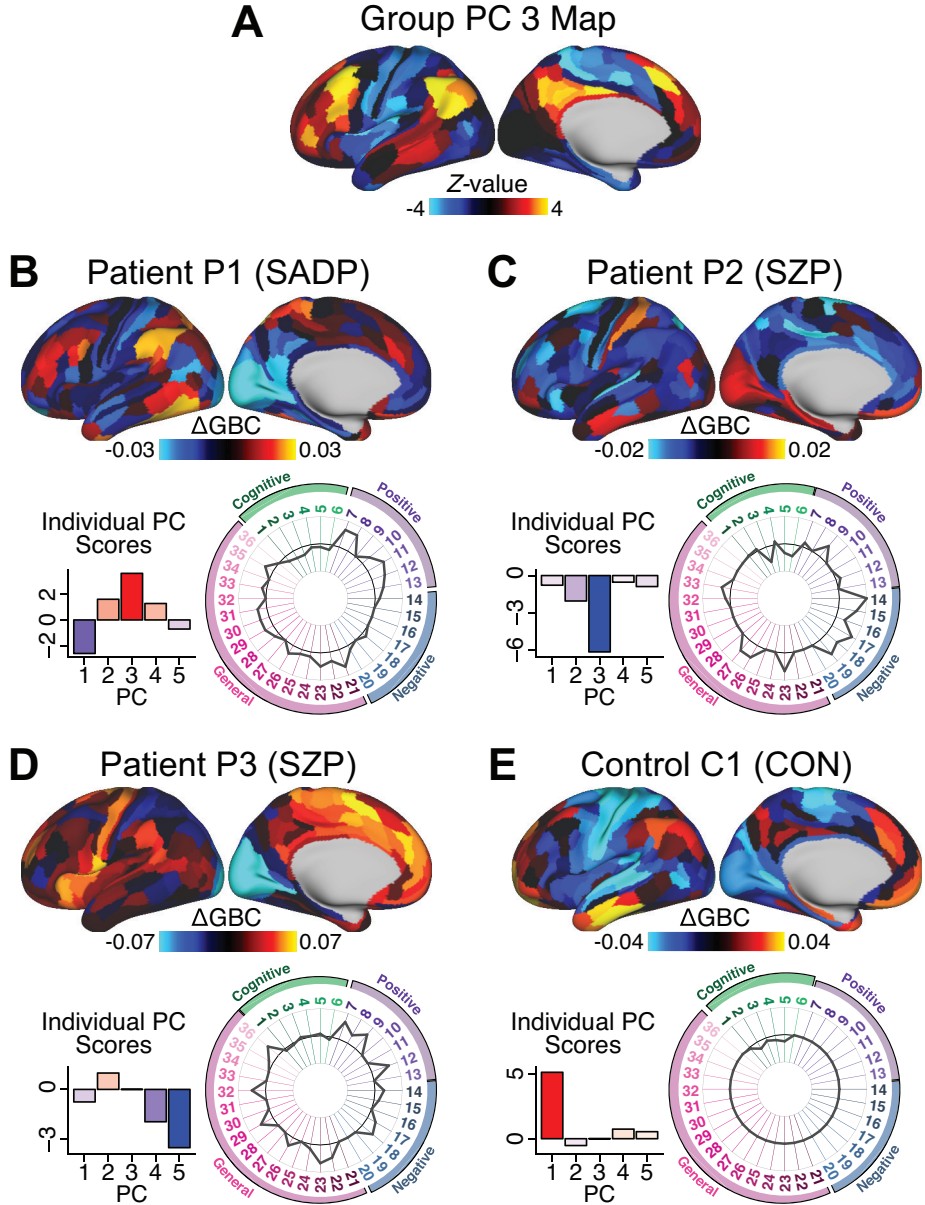

**Appendix 1—figure 11.** Individual patients exhibit complex projections into the PCA-derived brain-behavioral space (BBS) geometry (**A**) The *PC3* group-level cortical map is shown here for comparison purposes. (**B**) Neural and behavioral profile for one PSD patients, 'Patient P1'. Neural data shows the ΔGBC for Patient P1 (i.e. GBC demeaned relative to group mean of the entire PSD patient sample) to reflect the pattern of changes in their GBC relative to the 'average' of the entire sample. The bar plot shows five behavioral PC scores for Patient P1. The radarplot shows original scores on the 36 individual symptom measures for Patient P1 (black circle indicates zero; positive values are indicated by outward deviation of the gray line). (C–D) Data is shown for two other individual Patients P2 and P3. (**E**). Data is shown for one Control C1 participant. Note the difference between the symptom-neural profiles of Patient P1, a positive *PC3* 'loading' individual, and Patient P2, a negative *PC3* 'loading' individual, both of whom were diagnosed with SZP under a conventional categorical DSM approach. These data further illustrate the need to consider the complex relationships in the symptom-neural mapping and highlight that individual-level precision in BBS mapping can be obtained.

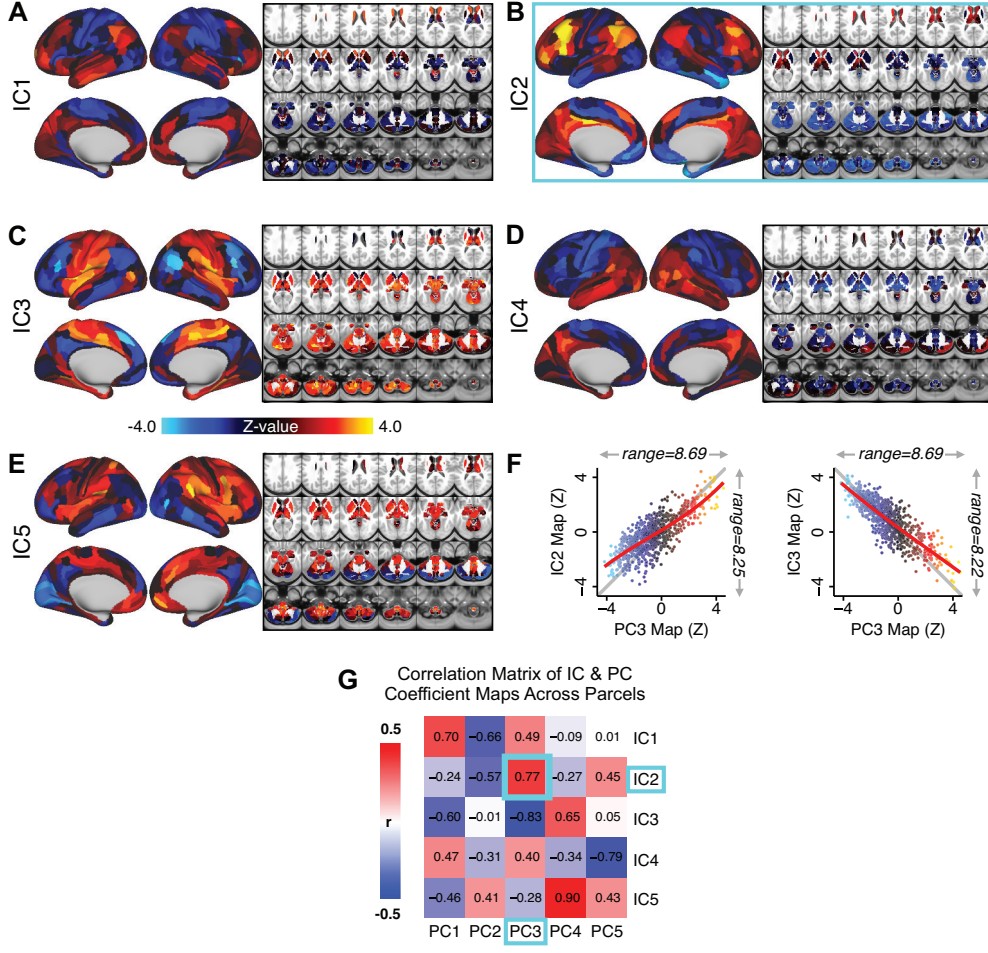

**Appendix 1—figure 12.** Independent component analysis (ICA) as an alternative method of dimensionality-reduction for symptom-neural mapping. (**A–E**) Relationships across all patients (N = 436) at each brain location between global brain connectivity (GBC) and IC score, for ICs 1–5. Values shown in each brain parcel is the Z-scored regression coefficient of IC score on to parcel GBC, across all 436 subjects. (**F**) Scatterplots showing the relationships across parcels between the PC3 map and the IC2 and IC3 maps. Sigmoids and a greater range (max - min) indicate an improvement in the Z-statistics of the PC3 map relative to the IC maps. (**G**) Correlation matrix showing correlations of individual parcel regression coefficients for the five significant PCs and the five ICs (shown in A–E), across all 718 neural parcels. Of note, *PC3 'Psychosis Configuration'* discovered via the PCA solution appears to be oblique to both *IC2* and *IC3*, suggesting that these two *ICs* may capture diverging tails of the *PC3* axis; however, the orthogonality of the PCA solution yielded maximally separated symptom dimensions that mapped on to unique neural variance, as shown by the superior statistics of the PC3 map versus the *IC2/IC3* maps.

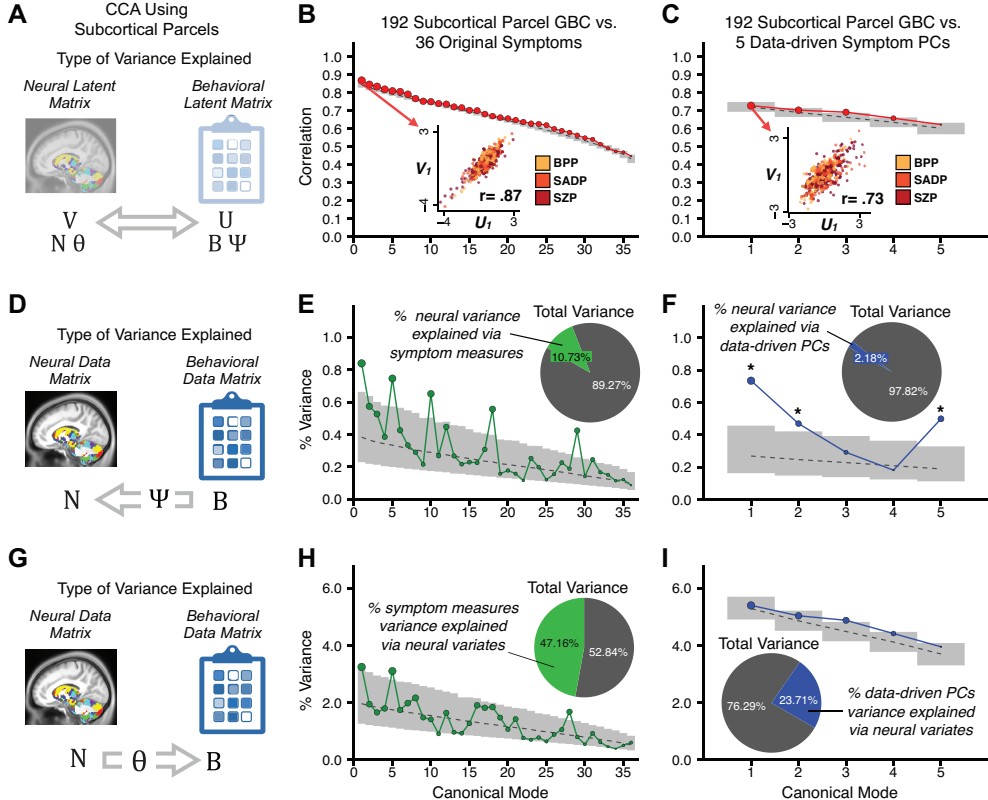

**Appendix 1—figure 13.** Canonical correlation analysis (CCA) of behavioral and subcortical neural features. (**A**) As noted in the main text, CCA maximizes correlations between canonical variates (CVs), that is matrices *U* and *V*. Here we evaluated a version of CCA using subcortical neural parcel features in relation to item-level symptom measures or dimensionality-reduced PC symptom measures. (**B**) Screeplot showing canonical modes for the CVs obtained from 192 subcortical neural features (GBC of parcels from a neurobiologically-derived functional parcellation of the subcortex) and 36 single-item PANSS and BACS symptom measures. Inset illustrates the correlation ($r = 0.85$) between the CV of the first mode, $U_1$ and $V_1$ (note that the correlation was not driven by a separation between diagnoses). (**C**) CCA was obtained from 192 subcortical neural features and five low-dimensional symptom scores derived via the PCA analysis. Here, all modes remained significant after FDR correction. Dashed black line shows the null calculated via a permutation test with 5000 shuffles; gray bars show 95% confidence interval. (**D**) Correlation between the neural data matrix *N* and the behavioral data matrix weighted by the transformation matrix (**B**Ψ) reflects the amount of variance in *N* that can be explained by the final latent neural matrix *U*. Put differently, this transformation calculates how much of the neural variation can be explained by the latent behavioral features. (**E**) Proportion of symptom variance explained by each of the neural CVs in a CCA performed between 192 subcortical neural features and all 36 behavioral measures. Inset shows the total proportion of behavioral variance explained by the neural variates. (**F**) Proportion of total behavioral variance explained by each of the neural CVs in a CCA performed between 192 subcortical neural features and the five low-dimensional symptom scores derived via the PCA analysis. Dashed black line shows the null calculated via a permutation test with 5000 shuffles; gray bars show 95% confidence interval. (**G**) Correlation between the behavioral data matrix *B* and the neural data matrix weighted by the transformation matrix (**N**Θ) reflects the amount of variance in *B* that can be explained by the final latent neural matrix *V*. Put differently, this transformation calculates how much of the symptom variation can be explained by the latent neural features. (**H**) Proportion of symptom variance explained by each of the neural CVs in a CCA performed between 192 subcortical neural features and all 36 behavioral measures. Inset shows the total proportion of

*Appendix 1—figure 13 continued on next page*

*Appendix 1—figure 13 continued*

behavioral variance explained by the neural variates. (**I**) Proportion of total behavioral variance explained by each of the neural CVs in a CCA performed between 192 subcortical neural features and the five low-dimensional symptom scores derived via the PCA analysis. While CCA using symptom PCs has fewer dimensions and thus lower total variance explained (see inset), each neural variate explains a higher amount of symptom variance than seen in F, suggesting that CCA could be further optimized by first obtained a principled low-rank symptom solution. Dashed black line shows the null calculated via a permutation test with 5000 shuffles; gray bars show 95% confidence interval.

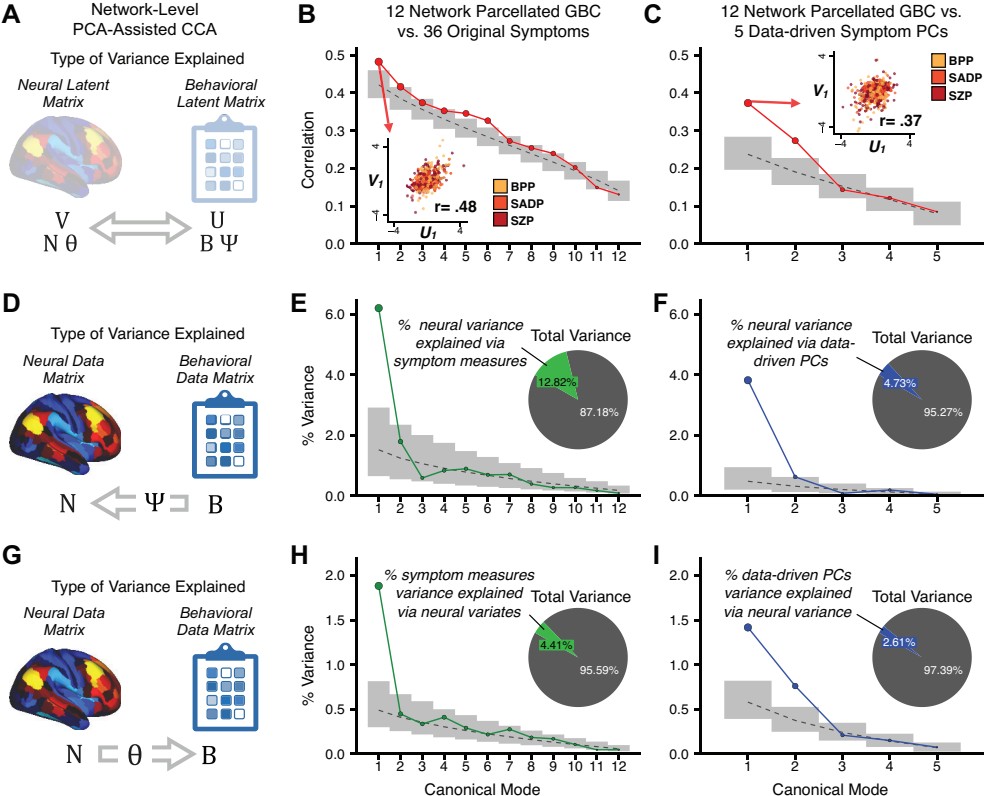

**Appendix 1—figure 14.** Canonical correlation analysis (CCA) of behavioral and network-level neural features. (**A**) As noted in the main text, CCA maximizes correlations between canonical variates (CVs), that is matrices *U* and *V*. Here we evaluated a version of CCA using network-level neural parcel features in relation to item-level symptom measures or dimensionality-reduced PC symptom measures. (**B**) Screenplot showing canonical modes for the CVs obtained from 192 subcortical neural features (GBC of parcels from a neurobiologically derived functional parcellation of the subcortex) and 36 single-item PANSS and BACS symptom measures. Inset illustrates the correlation (*r* = 0.85) between the CV of the first mode, $U_1$ and $V_1$ (note that the correlation was not driven by a separation between diagnoses). (**C**) CCA was obtained from 192 subcortical neural features and five low-dimensional symptom scores derived via the PCA analysis. Here, all modes remained significant after FDR correction. Dashed black line shows the null calculated via a permutation test with 5000 shuffles; gray bars show 95% confidence interval. (**B**) Screeplot showing the correlations between the canonical variates of a CCA performed between 12 neural features (mean GBC of 12 whole-brain networks) and all 36 behavioral measures. Dashed black line shows the null calculated via a permutation test with 5000 shuffles; gray bars show 95% confidence interval. Inset illustrates the correlation (*r* = 0.48) between the canonical variates of the first mode, $U_1$ and $V_1$. Note that the correlation is not driven by a separation between categorical diagnoses. (**C**) Screeplot showing the

*Appendix 1—figure 14 continued*

correlations between the canonical variates of a CCA performed between 12 network neural features and the five behavioral principal components. Dashed black line shows the null calculated via a permutation test with 5000 shuffles; gray bars show 95% confidence interval. Inset illustrates the correlation ($r = 0.37$) between the canonical variates of the first mode, $V_1$ and $V_1$. Note that the strength of the correlations is greatly reduced compared to the parcel-level CCA shown in *Figure 5*. (**D**) The correlation between the neural data N and the transformed behavioral data matrix Bψ reflects the amount of variance in N that can be explained by behavioral canonical variates V. (**E**) Screeplot showing the proportion of neural variance explained by each of the behavioral canonical variates in a CCA performed between 12 network neural features and all 36 behavioral measures. Dashed black line shows the null calculated via a permutation test with 5000 shuffles; gray bars show 95% confidence interval. Inset shows the total proportion of neural variance explained by the behavioral variates. (**F**) Screeplot of the proportion of neural variance explained by each of the behavioral canonical variates in a CCA performed between 12 network neural features and the five PCs of behavior. (**G**) The correlation between the behavioral data B and the transformed neural data matrix NΘ reflects the amount of variance in B that can be explained by neural canonical variates U. (**H**) Screeplot showing the proportion of behavioral variance explained by each of the neural canonical variates in a CCA performed between 12 network neural features and all 36 behavioral measures. Dashed black line shows the null calculated via a permutation test with 5000 shuffles; gray bars show 95% confidence interval. Inset shows the total proportion of behavioral variance explained by the neural variates.t (**I**) Screeplot of the proportion of total behavioral variance explained by each of the neural canonical variates in a CCA performed between 12 network neural features and the 5 PCs of behavior. Note that although the CCA using behavioral PCs has far fewer dimensions, each neural variate explains a higher amount of total behavioral variance than neural variates in H, suggesting that the identified PCs of behavior capture variance with far fewer features.

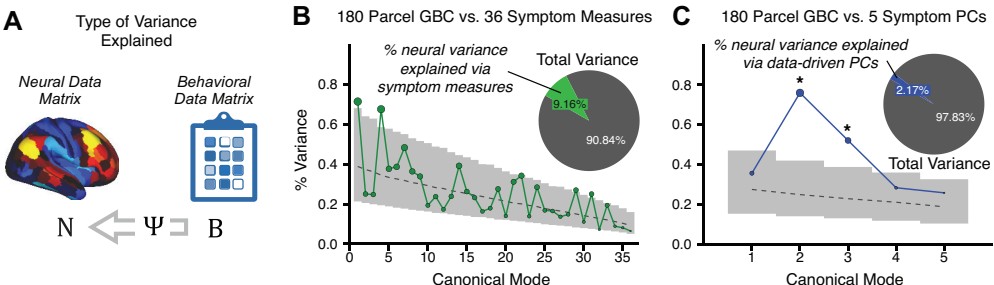

**Appendix 1—figure 15.** Canonical correlation analysis (CCA) results showing the amount of variance explained in the neural features via behavioral canonical variates. (**A**) The correlation between the neural data N and the transformed behavioral data matrix *B*ψ reflects the amount of variance in *N* that can be explained by behavioral canonical variates. (**B**) Screeplot showing the proportion of neural variance explained by each of the behavioral canonical variates in a CCA performed between 180 neural features (symmetrized cortical parcel GBC) and all 36 behavioral measures. Dashed black line shows the null calculated via a permutation test with 5000 shuffles; gray bars show 95% confidence interval. Inset shows the total proportion of neural variance explained by the behavioral variates. (**C**) Screeplot of the proportion of neural variance explained by each of the behavioral canonical variates in a CCA performed between 180 neural features (symmetrized cortical parcel GBC) and the five PCs of behavior.

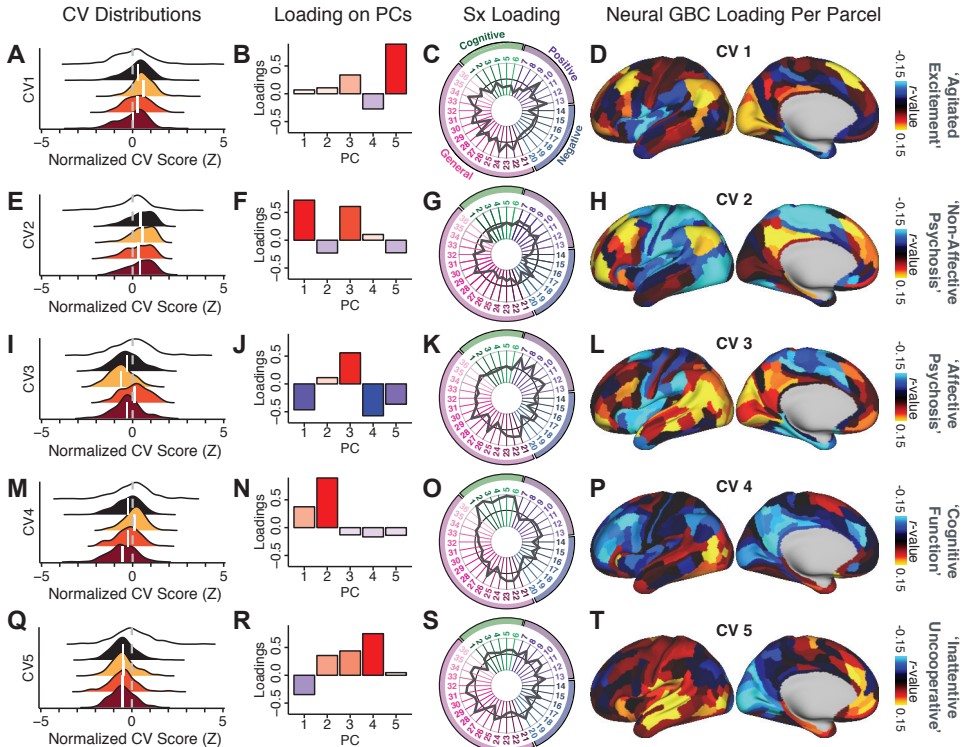

**Appendix 1—figure 16.** Canonical correlation analysis (CCA) symptom and neural configurations. (**A**) Distributions of scores for the first canonical variate (*CV1*) by diagnostic group (white: controls; black: all patients; yellow: bipolar disorder with psychosis; orange: schizoaffective disorder with psychosis; red: schizophrenia). All scores are normalized to controls. (**B**) Loadings of the PC-derived symptom scores for *CV1*. (**C**) Loadings of the original item-level symptom measures for *CV1*. (**D**) Loadings of each neural parcel for *CV1*. (**E–T**) The same analyses are shown for canonical variates 2–5. As noted in the main text, these CCA effects were not stable when tested via out-of-sample cross-validation methods (see *Figure 5*).

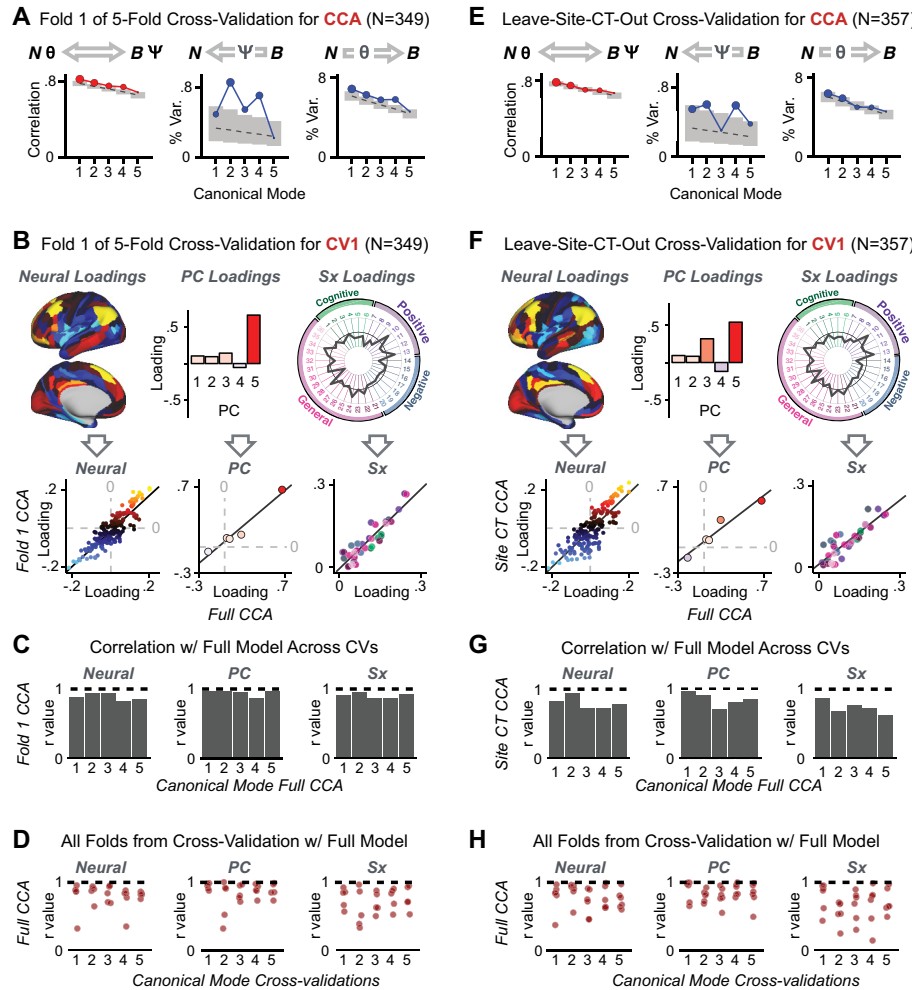

**Appendix 1—figure 17.** K-Fold cross-validation of canonical correlation analysis (CCA). In the main text, we show that CCA effects do not reproduce when tested via out-of-sample cross-validation methods (see *Figure 5*). In contrast, here we show that using within-sample K-fold methods yields effects that appear reproducible but in fact reflect an overfit CCA solution. Panels (**A–E**) illustrate the results from a 5-fold cross-validation analysis to test the reproducibility of the CCA solution. Subjects were first randomly assigned to one of five subsets. Each subset of subjects was then used as an independent 'test sample' in a CCA that was derived from subjects in the other four subsets. (**A**) Results of CCA performed in a subsample of N = 349 subjects. Left: screeplot showing the correlations between the canonical variates of a CCA performed between 180 neural features (symmetrized cortical parcel GBC) and the five behavioral principal components in N = 349 subjects. Middle: Screeplot of the proportion of neural variance explained by each of the behavioral canonical variates in a CCA performed between 180 neural features (symmetrized cortical parcel GBC) and the five PCs of behavior. Right: Screeplot of the proportion of total behavioral variance explained by each of the neural canonical variates in a CCA performed between 180 neural features (symmetrized cortical parcel GBC) and the five PCs of behavior. Dashed black line shows the null calculated via a permutation test with 5000 shuffles; gray bars show 95% confidence interval. Note that the outcome of this CCA is highly similar to the full sample CCA shown in the main text, but it is not similar at all to the split-half cross-validation effects, which reveal how overfit CCA effects seem to be with this number of features *Figure 5*. (**B**) Comparison of the first canonical variate (CV1) in a CCA performed in a subsample of N = 349 subjects with *CV1* from the full model performed with N = 436 subjects. Left: neural factor loadings for *CV1* obtained from the subsample (Fold 1) CCA and from the full

*Appendix 1—figure 17 continued*

sample CCA are highly correlated, at r = 0.85. Middle: Behavioral factor loadings are also highly correlated between the Fold 1 CCA and the full CCA, at r = 0.85. Right: Additionally, absolute values of individual symptom measure loadings associated with *CV1* are highly correlated between the Fold 1 CCA and the full CCA. (**C**) Summary of correlation values between Fold 1 CCA and full CCA results, as in panel B, across all 5 CVs. Note that the neural and behavioral factor loadings as well as individual symptom measure loadings are highly preserved between the sample CCA and the full model. (**D**) Summary of correlation values between all five subsample CCAs and the full sample CCA, for all five CVs. Each of the five CVs is plotted along the X-axis; each point represents the correlation between one of the fivefold subsample CCAs and the full CCA, hence there are five points (in the fivefold cross-validation) for each CV. (**E**) Results of a CCA performed in a subsample of N = 357 subjects, with subjects from one site left out (leave-one-site-out, LOSO). Panels as described in A. (**F**) Comparison of the first canonical variate (CV1) in a LOSO CCA and the *CV1* from the full model performed with N = 436 subjects. Panels as described in B. (**G**) Summary of correlation values between the LOSO CCA and full CCA results, as in panel B, across all five CVs. Panels as described in C. (**H**) Summary of correlation values between all six LOSO CCAs and the full sample CCA, for all five CVs. Panels as described in D; each point represents the correlation between one of the six LOSO CCAs and the full CCA.

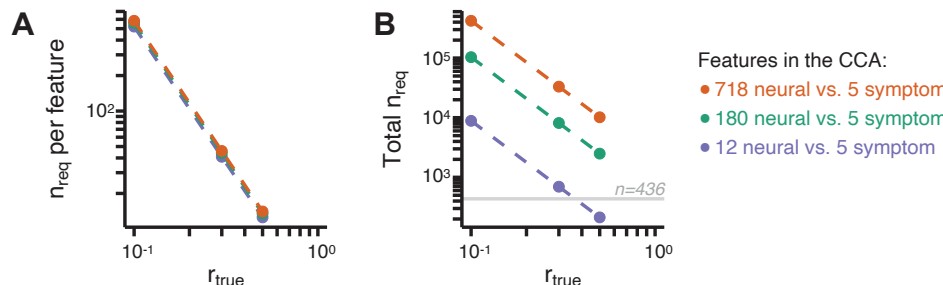

**Appendix 1—figure 18.** Multivariate power analysis for CCA. Sample sizes were calculated according to *Helmer et al., 2020*, see also **Materials and methods** and https://gemmr.readthedocs.io/en/latest/. We computed the multivariate power analyses for three versions of CCA reported in this manuscript: (i) 718 neural vs. five symptom features; (ii) 180 neural vs. five symptom features; (iii) 12 neural vs. five symptom features. (**A**) At different levels of features, the ratio of samples (i.e. subjects) required per feature to derive a stable CCA solution remains approximately the same across all values of $r_{true}$. As discussed in *Helmer et al., 2020*, at $r_{true} = 0.3$ the number of samples required per feature is about 40, which is much greater than the ratio of samples to features available in our dataset. (**B**) The total number of samples required ($n_{req}$) for a stable CCA solution given the total number of neural and symptom features used in our analyses, at different values of $r_{true}$. In general these required sample sizes are much greater than the N = 436 (horizontal gray line) PSD in our present dataset, consistent with the finding that the CCA solutions computed using our data were unstable. Notably, the '12 vs. 5' CCA assuming $r_{true} = 0.3$ requires only ~700 subjects, which is closest to the N = 436 used in the present sample. This may be in line with the observation of the CCA with 12 neural vs five symptom features (*Appendix 1—figure 15C*) that the canonical correlation ($r = 0.38$ for CV1) clearly exceeds the 95% confidence interval, and may be closer to the true effect. However, to to confidently detect effects in such an analysis (particularly if $r_{true}$ is actually less than 0.3), a larger sample would likely still be needed.

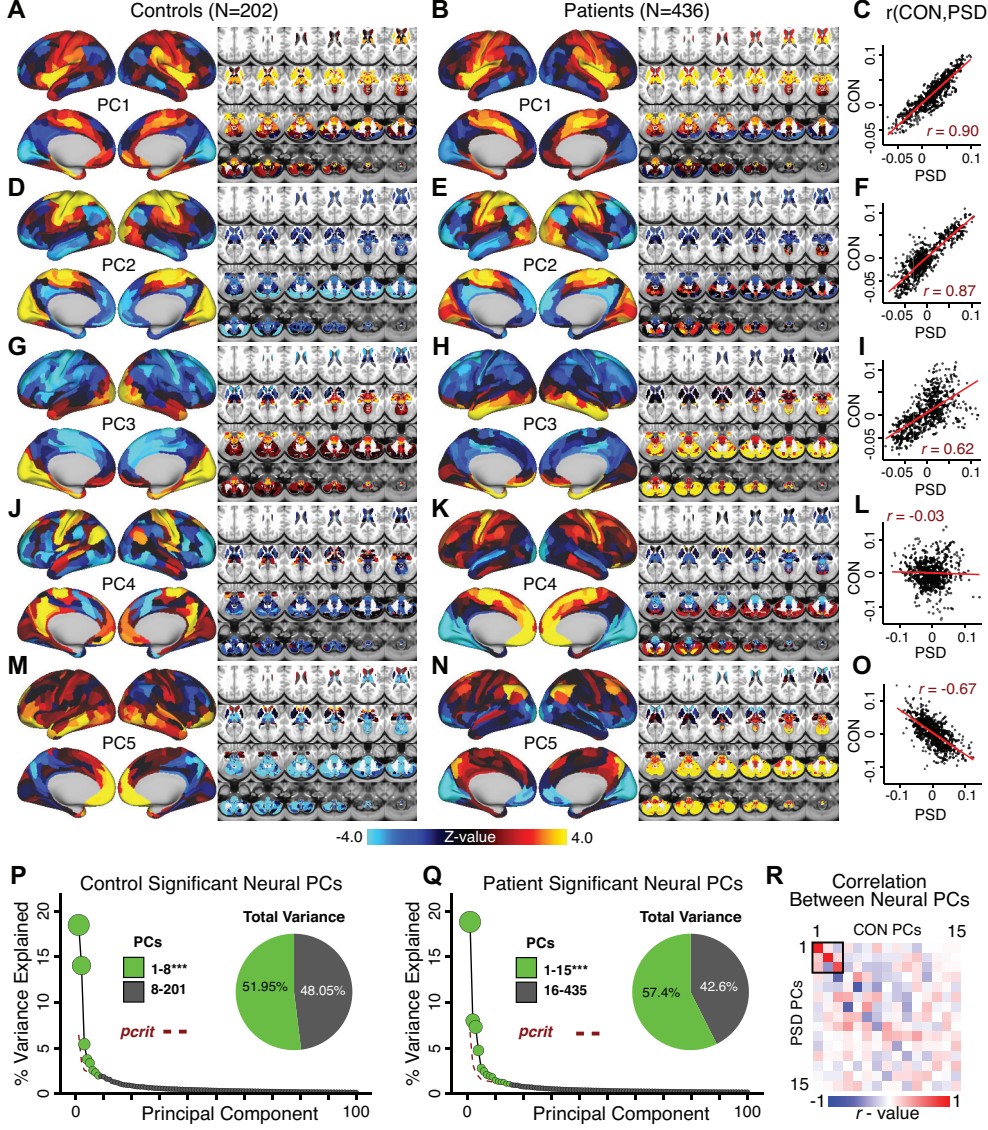

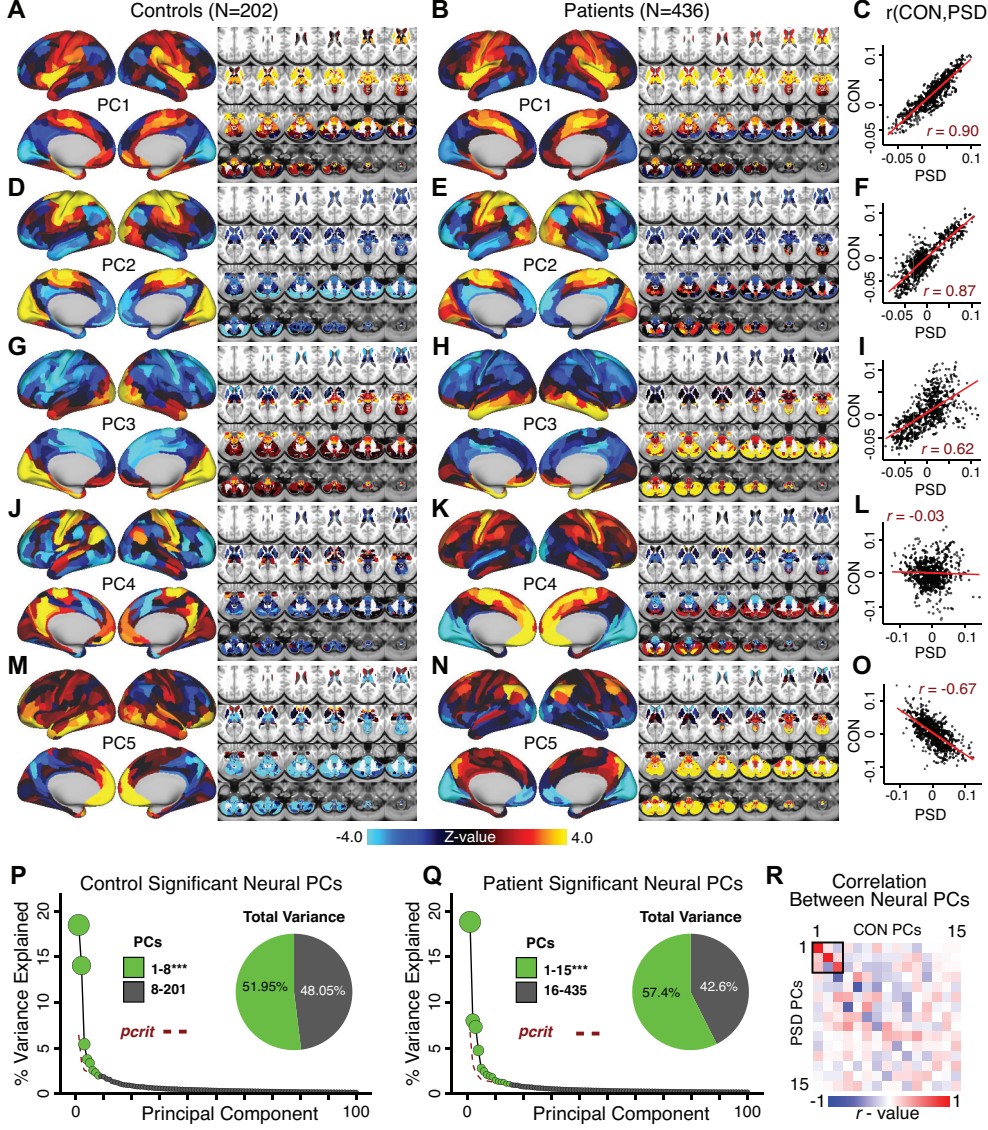

**Appendix 1—figure 19.** Principal component analysis (PCA) of neural features in control and patient subjects. (**A**) Results of PCA performed on neural features (718 whole-brain parcel GBC) across all control subjects (N = 202). (**B**) Results of PCA performed on neural features (718 whole-brain parcel GBC) across all patient subjects (N = 436). (**C**) Note that the first neural PC in control subjects (panel A) and in patient subjects (panel B) are highly similar (r = 0.90 across parcels). This may reflect a component of neural variance common across all humans. (**D–I**) The second and third PCs are also similar between control and patient subjects. (**J–L**) The fourth PC is markedly dissimilar between control and patient subjects, suggesting that this component may reflect neural variance which deviates in individuals with psychiatric symptoms. (**M–O**) The fifth PC is also highly dissimilar between controls and patients, also possibly reflecting diagnosis-relevant differences in neural variance. (**P**) Screeplot showing the total proportion of variance explained by the first 100 PCs from the PCA performed across all 718 neural features in 202 control subjects. The size of each point is proportional to the variance explained by that PC. The first 8 PCs (green) were determined to be significant using a permutation test. Inset shows the proportion of variance both accounted and not accounted for by the eight significant PCs. Together, these 8 PCs capture 51.95% of the total variance in neural GBC in the sample. (**Q**) Screeplot showing the total proportion of variance explained by the first 100 PCs from the PCA performed across all 718 neural features in 436 patient

*Appendix 1—figure 19 continued on next page*

subjects. The size of each point is proportional to the variance explained by that PC. The first 15 PCs (green) were determined to be significant using a permutation test. Inset shows the proportion of variance both accounted and not accounted for by the 15 significant PCs. Together, these five PCs capture 57.4% of the total variance in neural GBC in the sample. (**R**) Correlations between the first 15 neural PCs in controls (CON) and patients (PSD). The first three PCs are common across both control and patient subjects, possibly reflecting common components of human neural variance. After this, the PCs diverge, possibly reflecting differences in neural variance of healthy individuals and those with psychiatric symptoms.

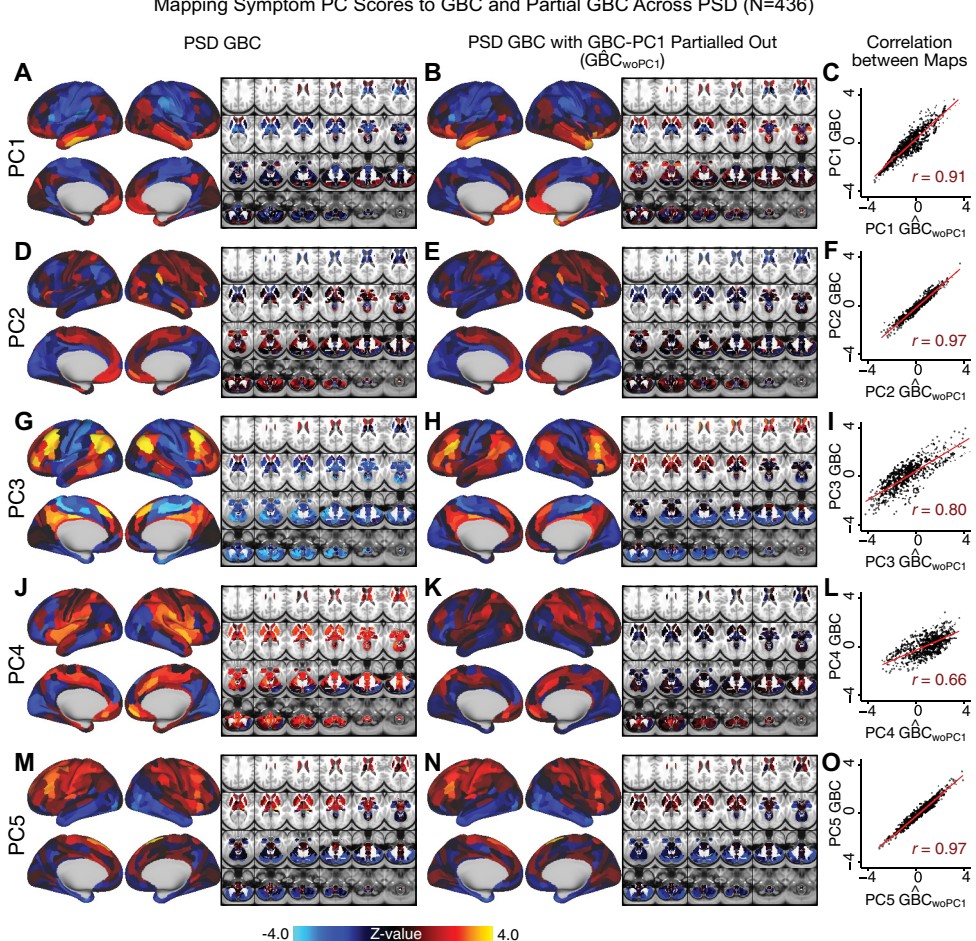

**Appendix 1—figure 20.** Comparison between the PSD $\beta_{PC}GBC$ maps computed using GBC and GBC with the first neural PC parsed out. If a substantial proportion of neural variance is not be clinically relevant then removing the shared neural variance between PSD and CON should not drastically affect the reported symptom-neural univariate mapping solution, because this common variance will not map to clinical features. We therefore performed a PCA on CON and PSD GBC to compute the shared neural variance (see *Materials and methods*), and then parsed out the first GBC-PC from the PSD GBC data ($\hat{GBC}_{woPC1}$). We then reran the univariate regression as described in *Figure 3*, using the same five symptom PC scores across 436 PSD. (**A**) The $\beta_{PC1}GBC$ map, also shown in *Appendix 1—figure 10*. (**B**) The first GBC-PC accounted for about 15.8% of the total GBC variance across CON and PSD. Removing GBC-PC1 from PSD data attenuated the $\beta_{PC1}GBC$ statistics slightly (not unexpected as the variance was by definition reduced) but otherwise did not

strongly affect the univariate mapping solution. (**C**) Correlation across 718 parcels between the two $\beta_{PC1}GBC$ map shown in A and B. (**D–O**) The same results are shown for $\beta_{PC2}GBC$ to $\beta_{PC5}GBC$ maps.

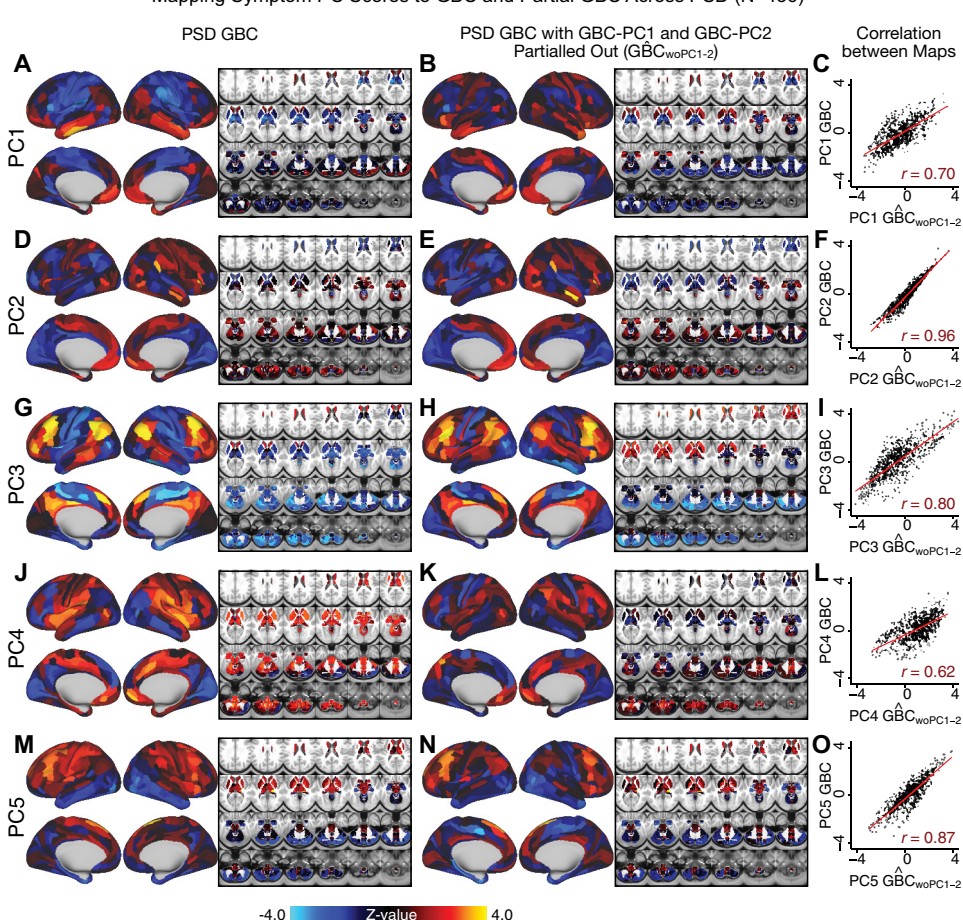

**Appendix 1—figure 21.** Comparison between the PSD $\beta_{PC}GBC$ maps computed using GBC and GBC with the first two neural PCs parsed out. We performed a PCA on CON and PSD GBC and then parsed out the first three GBC-PC from the PSD GBC data ($G\hat{B}C_{woPC1-2}$, see ***Materials and methods***). We then reran the univariate regression as described in ***Figure 3***, using the same five symptom PC scores across 436 PSD. (**A**) The $\beta_{PC1}GBC$ map, also shown in *Appendix 1—figure 10*. (**B**) The second GBC-PC accounted for about 9.5% of the total GBC variance across CON and PSD. (**C**) Correlation across 718 parcels between the two $\beta_{PC1}GBC$ map shown in A and B. (**D–O**) The same results are shown for $\beta_{PC2}GBC$ to $\beta_{PC5}GBC$ maps.

Mapping Symptom PC Scores to GBC and Partial GBC Across PSD (N=436)

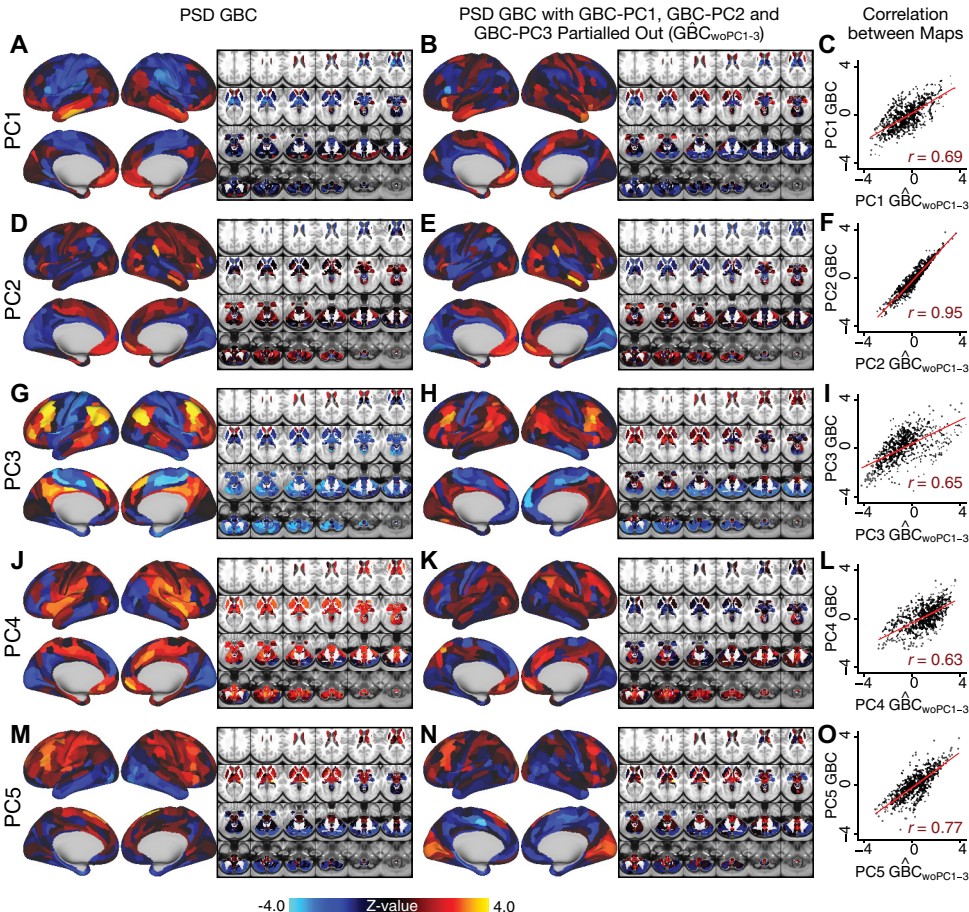

**Appendix 1—figure 22.** Comparison between the PSD $\beta_{PC}GBC$ maps computed using GBC and GBC with the first three neural PCs parsed out. We performed a PCA on CON and PSD GBC and then parsed out the first three GBC-PC from the PSD GBC data ($\hat{GBC}_{woPC1-3}$, see **Materials and methods**). We then reran the univariate regression as described in **Figure 3**, using the same five symptom PC scores across 436 PSD. (**A**) The $\beta_{PC1}GBC$ map, also shown in **Appendix 1—figure 10**. (**B**) The second GBC-PC accounted for about 9.5% of the total GBC variance across CON and PSD. (**C**) Correlation across 718 parcels between the two $\beta_{PC1}GBC$ map shown in A and B. (**D–O**) The same results are shown for $\beta_{PC2}GBC$ to $\beta_{PC5}GBC$ maps.

Mapping Symptom PC Scores to GBC and Partial GBC Across PSD (N=436)

**Appendix 1—figure 23.** Comparison between the PSD $\beta_{PC}GBC$ maps computed using GBC and GBC with the first four neural PCs parsed out. We performed a PCA on CON and PSD GBC and then parsed out the first four GBC-PC from the PSD GBC data ($G\hat{B}C_{woPC1-4}$, see **Materials and methods**). We then reran the univariate regression as described in **Figure 3**, using the same five symptom PC scores across 436 PSD. (**A**) The $\beta_{PC1}GBC$ map, also shown in **Appendix 1—figure 10**. (**B**) The second GBC-PC accounted for about 9.5% of the total GBC variance across CON and PSD. (**C**) Correlation across 718 parcels between the two $\beta_{PC1}GBC$ map shown in A and B. (**D–O**) The same results are shown for $\beta_{PC2}GBC$ to $\beta_{PC5}GBC$ maps.

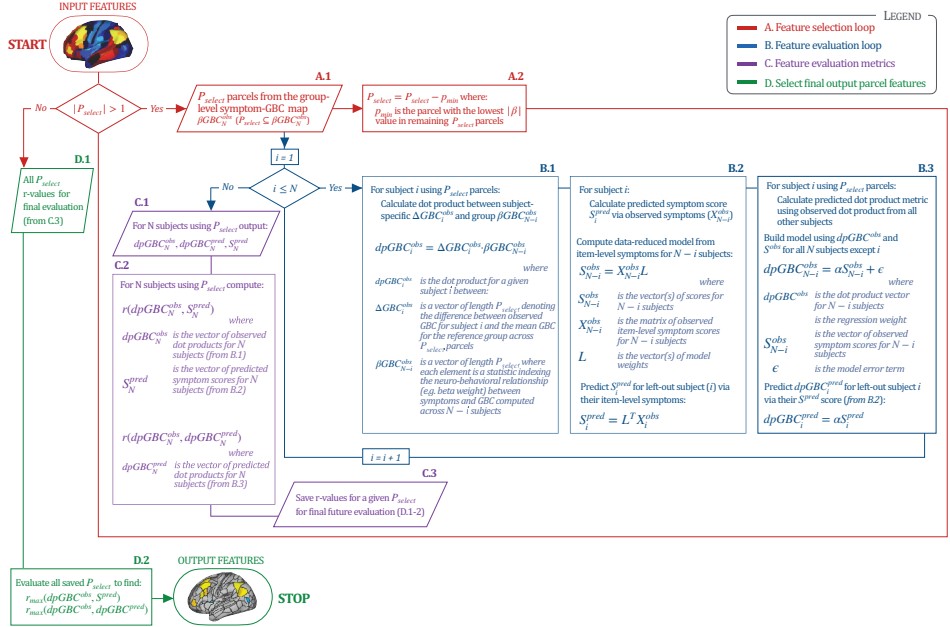

**Appendix 1—figure 24.** Workflow for optimizing the neural feature selection for a predictive symptom-neural subject specific model. We optimized the selection of a subset of neural parcels from the group $\beta GBC_N^{obs}$ map (where $N$ is the total number of subjects) that could be predicted via a single symptom score across subjects (e.g. for the *PC3* beta map and *PC3* symptom scores). (START) We start with $P_{select}$ set as the full parcellated map, $P_{select} = \beta GBC_N^{obs}$. Hence initially there are 718 elements in the vector $P_{select}$, each representing the value of a parcel ($|P_{select}| = 718$). Next, since $|P_{select}| > 1$, the selection process enters the A. Feature Selection Loop (in red). Selects subset of parcels $P_{select}$ (A.1) for which the predictive model will be evaluated. This loop iteratively removes parcels from the $P_{select}$ vector (A.2) and each time passes the updated $P_{select}$ as input in the Feature Evaluation Loop (in blue). B. Feature Evaluation Loop (shown in blue). Using a multi-step algorithm, this loop produces the set of metrics to be evaluated at the final output using a leave-one-subject out approach. B.1: For each subject $i$, this first step is to compute the 'observed' dot product metric $dpGBC_i^{obs}$ for that subject. This is the dot product between $\Delta GBC_i^{obs}$ and $\beta GBC_{N-i}^{obs}$ using the $P_{select}$ features. $\Delta GBC_i^{obs}$ is the vector of the parcel-wise difference between subject $i$'s GBC map and the group mean GBC map; $\beta GBC_{N-i}^{obs}$ denotes group-level relationship between symptom variation and GBC excluding subject $i$. This group $\beta GBC_{N-i}^{obs}$ is used as a reference for the dot-product. B.2: Next for subject $i$ we compute their predicted symptom score $S_i^{pred}$. First, a symptom-prediction model is built using $X_{N-i}^{obs}$ (all item-level symptom measures excluding subject $i$). This yields the 'observed' model scores and coefficients excluding subject $i$ ($S_{N-i}^{obs}$). The symptoms for excluded subject $i$ are then used to obtain the predicted symptom score $S_i^{pred}$, as done in *Figure 6A*. B.3: Lastly, for excluded subject $i$ the loop computes the "predicted' dot product metric $dpGBC_i^{pred}$ to compare it to the "observed' value for model evaluation. Here, a regression model is computed relating $dpGBC_{N-i}^{obs}$ to $S_{N-i}^{obs}$, excluding subject $i$. This yields regression coefficients $\alpha$, which are then used to predict the $dpGBC_i^{pred}$ for the excluded subject $i$ using their predicted symptom scores $S_i^{pred}$. Once $i > N$ (i.e. $dpGBC_i^{pred}$ and $S_i^{pred}$ have been computed for all subjects), the algorithm moves to step "C'. C. Feature Evaluation Metrics (shown in purple). C.1: This step captures feature evaluation metrics across N subjects for the model that was computed using the selected features ($P_{select}$). Specifically, three vectors (outputs from 'B') are computed: $dpGBC_N^{obs}$ (the vector of observed $dpGBC$ values for all $N$ subjects); $dpGBC_N^{pred}$ (the vector of predicted $dpGBC$ values for all $N$ subjects); and $S_N^{pred}$ (the vector of predicted model symptom scores for all $N$ subjects). C.2: Generates two feature evaluation metrics across the entire sample ($r(dpGBC_N^{obs}, S_N^{pred})$ and $r(dpGBC_N^{obs}, dpGBC_N^{pred})$). C.3: The feature

*Appendix 1—figure 24 continued on next page*

*Appendix 1—figure 24 continued*

evaluation metrics for the selected $P_{select}$ are generated, such that they can be evaluated across all $P_{select}$ iterations (red loop). D. Final Feature Selection (in green). This step evaluates the metrics computed for all $P_{select}$ iterations, starting from from $|P_{select}| = 718$ to $|P_{select}| = 1$. D.1: The input is the vector of $r(dpGBC_N^{obs}, S_N^{pred})$ and the vector of $r(dpGBC_N^{obs}, S_N^{pred})$ for all 718 models. D.2: The most predictive feature subset is obtained by finding the model for which $r(dpGBC_N^{obs}, S_N^{pred})$ and $r(dpGBC_N^{obs}, dpGBC_N^{pred})$ were maximized, which leads to the final output features (STOP). *Note:* that the algorithm imposed a threshold of $|P_{select}| > 30$ to ensure that the correlation metrics were not unstable based on a small feature set.

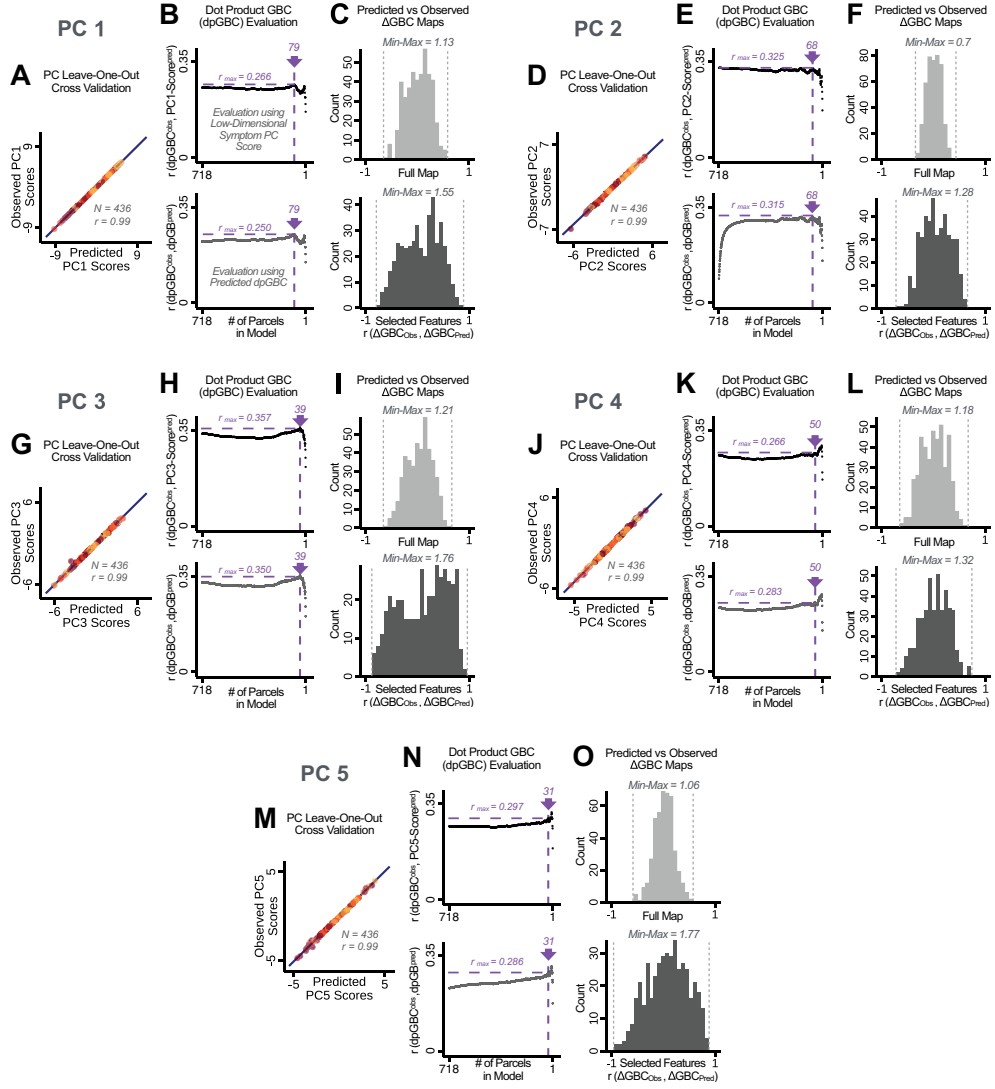

**Appendix 1—figure 25.** Optimizing neural feature selection to inform statistically reliable single-subject prediction via low-dimensional symptom scores only. (**A**) Individual PC-derived symptom scores for *PC1* can be robustly predicted, as shown by leave-one-out cross-validation for the symptom PCA analyses. Here for the leave-one-out cross-validation, a PCA was computed in all except one subject (N = 435) and the PCA loadings were then used to compute the predicted PC scores for the left-out subject. This was repeated such that predicted scores were calculated for

*Appendix 1—figure 25 continued on next page*

*Appendix 1—figure 25 continued*

every subject, using loadings from a PCA computed in the other 435 subjects. Observed scores were obtained from a PCA computed using the full sample of subjects (N = 436). Top panel: Scatterplot at left shows the correlation between each subject's predicted *PC1* score from leave-one-out PCA model and their observed *PC1* score from the full-sample PCA model, *r* = 0.99. Red line shows perfect prediction, *r* = 1. (**B**) Top panel indicates that PC symptom scores can yield robust neural feature selection. ΔGBC denotes an individual subject GBC demeaned parcel-wise relative to the group mean GBC. The process is described in detail in the **Materials and methods** and in *Appendix 1—figure 26*. The correlation between *PC1* behavioral score and [ΔGBC · *PC1* beta] across all 436 subjects was maximal for $P = 79$, at *r* = 0.27 (purple arrow). Bottom panel: For each value of *p*, the regression was performed in 435 subjects and was then used to predict the [ΔGBC · *PC1* beta] of the selected parcels in the left-out subject. The mean correlation between the predicted [ΔGBC·*PC1* beta] and the actual observed [ΔGBC · *PC1* beta] across all 436 subjects was highest for $P = 79$, with *r* = 0.19 across all subjects, highly consistent with the across-subject analysis in the top panel. (**C**) Comparing the correlation between $\Delta GBC^{obs}, \Delta GBC^{pred}$, for the full map ($P = 718$) versus the selected map ($P = 79$) shows that the predictability of $\Delta GBC$ is improved by using the selected parcels in the selected maps. (**D–O**) Similar results for PCs 2–5. Note that for *PC4* and *PC5* a local maximum ($P = 50$ and $P = 31$ respectively) was selected after imposing a threshold of $P>30$ to ensure that the correlation metrics used for evaluation were not made unstable from a small number of parcels in computing the *dpGBC*.

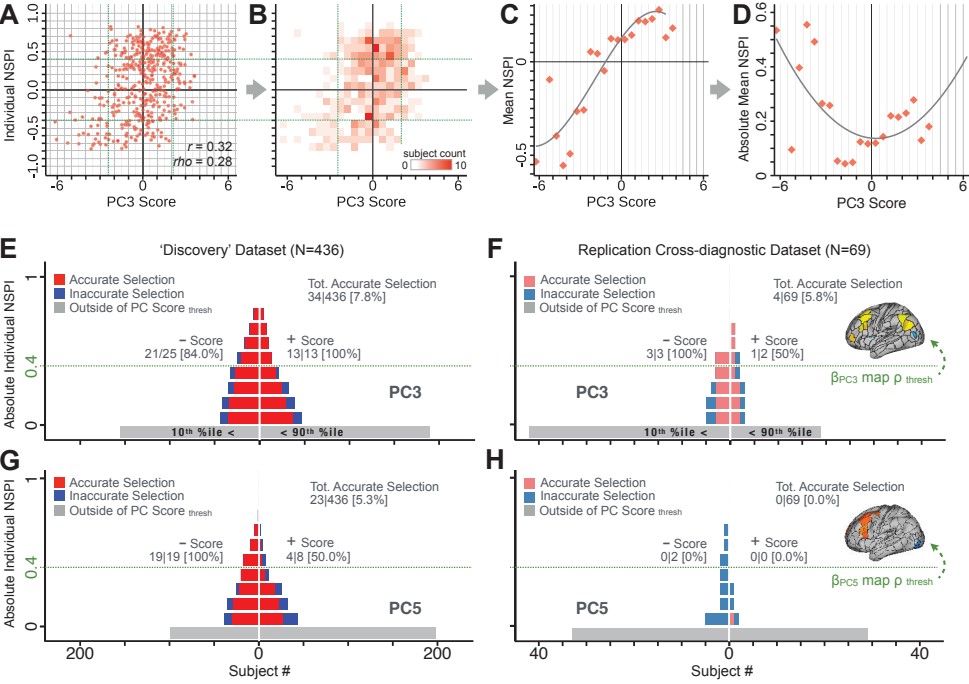

**Appendix 1—figure 26.** Evaluating patient-specific similarity to derived symptom-neural targets via data-reduced symptom scores. (**A**) Scatterplot of *PC3* symptom score (X-axis) versus *PC3* neural similarity prediction index (NSPI, Y-axis) for all 436 PSD subjects. The NSPI is defined as the Spearman's correlation between $\Delta GBC^{obs}$ and the $\beta_{PC3} GBC^{obs}$ map of the maximally-predictable 'selected' $P = 39$ parcels. Alternative metrics are shown in *Appendix 1—figure 28*. (**B**) Bins across axes express subject counts within each cell as a heatmap, indicating a high similarity between symptom PC score and *PC3* NSPI for a number of patients. (**C**) Mean NSPI is computed for a given bin along the the X-axis to visualize patient clustering. Note the sigmoidal shape of the distribution reflecting greater neural similarity at more extreme values of the *PC3* score. (**D**) The absolute value
*Appendix 1—figure 26 continued on next page*

*Appendix 1—figure 26 continued*

of the mean NSPI reflects the magnitude irrespective of neural similarity direction. This highlights a quadratic effect, showing that patients with higher *PC3* symptom scores (either positive/negative) exhibited higher neural correspondence of their maps with the target neural reference map. (**E**) Using the NSPI and PC scores we demonstrate one possible brain-behavioral patient selection strategy. We first imposed a PC score symptom threshold to select patients at the extreme tails (i.e. outside of the $10th - 90th\%tile$ behavioral range [>+2.17 or <−2.41]). Note that this patient selection strategy excludes patients (shown in gray) below the PC symptom score threshold. This yielded n = 38 patients. Next, for each patient we predicted the sign of their individual NSPI based on their individual PC3 score, which served as the basis for the neural selection. Next, at each NSPI threshold we evaluated the proportion of patients correctly selected until there were no inaccurately selected patients in at least one PC3 direction (green line or higher). The number of accurately (**A**) vs. inaccurately (**I**) selected patients within each bin is shown in red and blue respectively. Note that as the neural ρ threshold increases the A/I ratio increases. (**F**) The neural and behavioral thresholds defined in the 'discovery' sample were applied to an independent 'replication' dataset (N = 69, see *Materials and methods*), yielding a similar final proportion of accurately selected patients. (**G**) The same brain-behavioral patient selection strategy was repeated for *PC5* in the discovery sample (thresholds of $10^{th}\%tile = -1.89$ and $90^{th}\%tile = +1.47$; NSPI threshold of $\rho = 0.4$ optimized for PC5). Results yielded similar A/I ratios as found for PC3. (**H**) The neural and behavioral thresholds for PC5 defined in the discovery sample were applied to the replication sample. Here the results failed to generalize due true clinical differences between the discovery and replication samples.

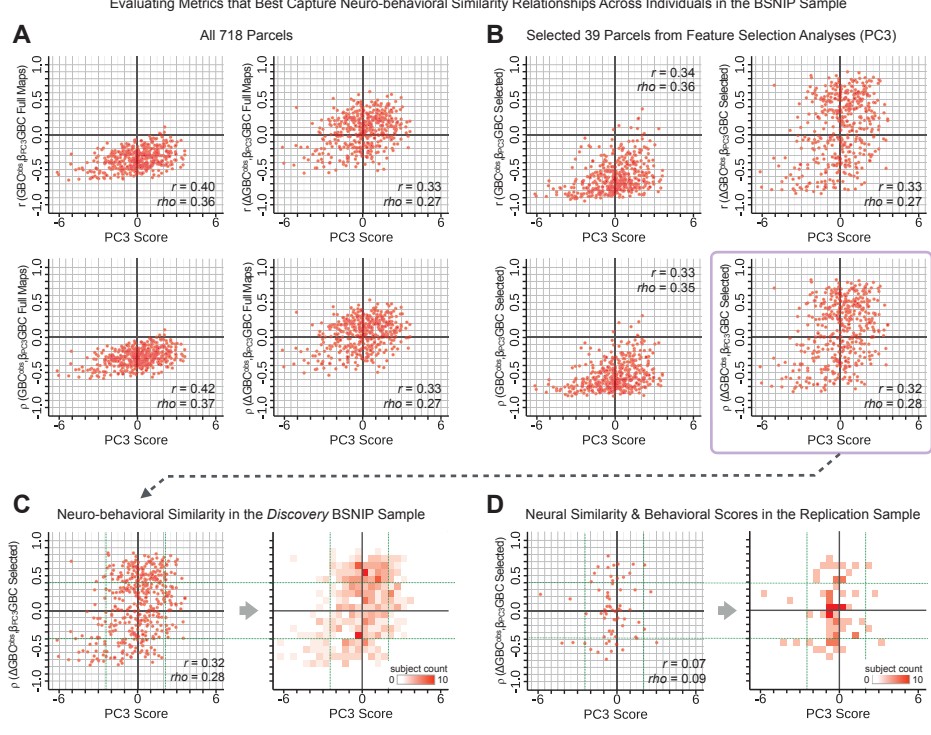

**Appendix 1—figure 27.** Metrics quantifying the relationship between individual-level neural effects relative to the group mean and individual-level PC-derived symptom scores. The relationship between various symptom-neural similarity metrics and the behavioral *PC3* symptom scores across N = 436 PSD patients are shown. The goal was to evaluate metrics that maximize the spread of the Y-axis, while maintaining a centered distribution of scores around zero to yield an interpretable
*Appendix 1—figure 27 continued on next page*

*Appendix 1—figure 27 continued*

symptom-neural variation pattern across subjects. Specifically, along the Y-axis we examined either the Pearson's correlation (top panels in A and B) or Spearman's ρ (bottom panels in A and B) between two neural maps at the individual patient level: (i) the group-level $\beta_{PC3}GBC$ map and (ii) an individual subjects' raw GBC map which can be expressed as either a raw map (left panels in A and B) or as $\Delta GBC^{obs}$ relative to the group mean map (right panels in A and B). (**A**) These four panels show group-level symptom-neural relationships for sample using the entire 718-parcel neural map. As noted, the neural similarity metrics on each Y-axis show the relationship between individual-patient neural GBC (or $\Delta GBC^{obs}$) and the group-level $\beta_{PC3}GBC$ map across 718 parcels. Note that the *r* and *rho* values reported in the corner of each plot reflect the *group-level* relationship between the neural similarity metric and the individual-level PC3 symptom score computed across across 436 patients. (**B**) These four parcels show group-level symptom-neural relationships using only the selected 39 *PC3* parcels derived from the feature selection workflow (*Appendix 1—figure 25*). The metric used in the main text is the ρ within selected parcels (shown with a light purple envelope) because the spread along Y-axis was maximized and the values centered around zero for both the x and the Y-axis. Furthermore, the ρ metric is desirable given violations of normality that the *r* metric assumes. (**C**) The neural-behavioral similarity plot for the Discovery BSNIP sample was binned by $\rho = 0.1$ & $PC3_{score} = 0.5$ to provide a visual intuition for patient segmentation across both the neural and symptom indices. For patients with highly positive or negative scores, the neuro-behavioral relationship was robust. Conversely, patients with a low absolute *PC3* score showed a weak relationship with symptom-relevant neural features. (**D**) The neural-behavioral similarity plot for the independent replication sample (see **Materials and methods** for sample details) was binned by $\rho = 0.1$ & $PC3_{score} = 0.5$, using the same neural similarity index ($\rho[\Delta GBC^{obs}, \beta_{PC3}GBC^{obs}]$ within selected 39 parcels) and projected *PC3* symptom scores. Although the grid is far more sparse (total N = 69 for this sample), a similar pattern between PC3 symptom score and neural similarity index emerges.

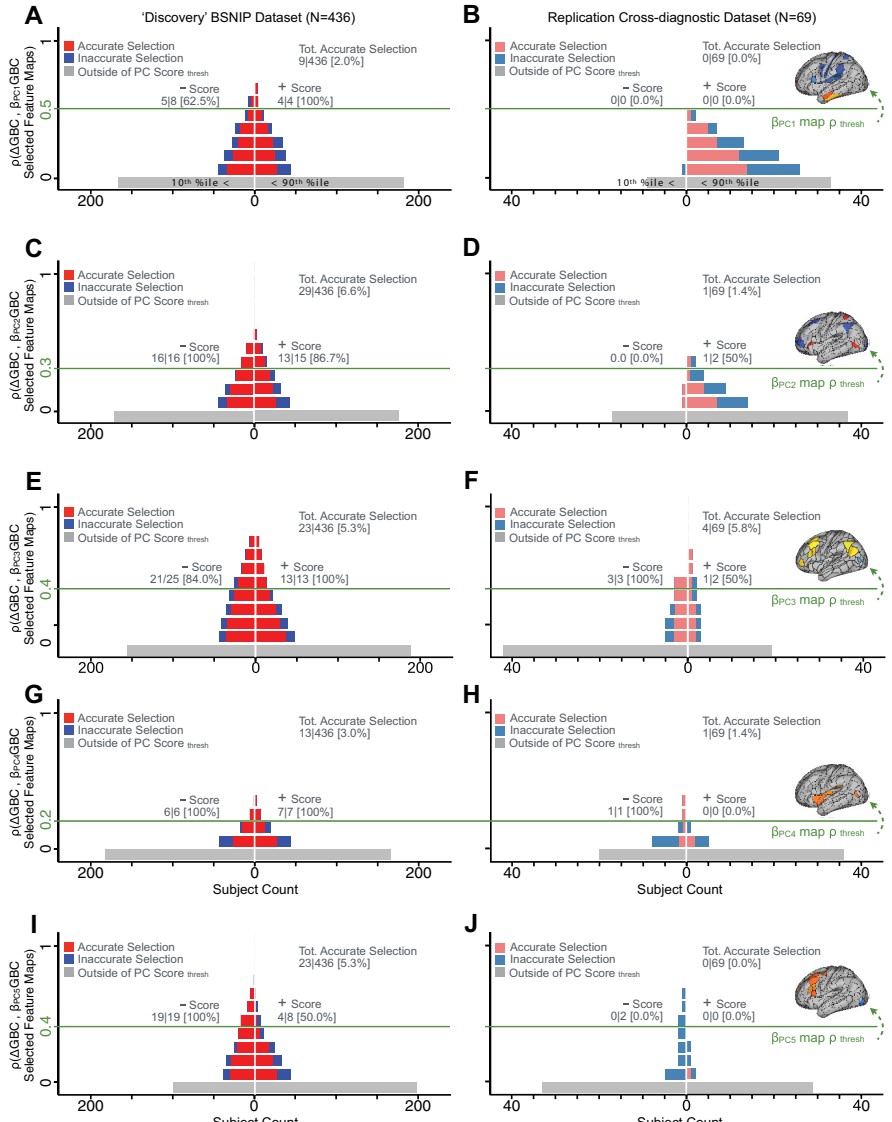

**Appendix 1—figure 28.** Evaluating patient-specific similarity to derived symptom-neural targets via data-reduced symptom scores for all *PCs*. As noted in the main text for *PC3*, here we plot the *PC* symptom score (X-axis) versus the neural similarity prediction index (NSPI, *Appendix 1—figure 26*) for all 436 PSD patients. The NSPI is defined as the Spearman's correlation between $\Delta GBC^{obs}$ and the $\beta_{PC3}GBC^{obs}$ map of the maximally-predictable 'selected' *P* parcels (Y-axis for each plot). Alternative metrics were evaluated in *Appendix 1—figure 27*. (**A**) A parameter sweep was performed for 0<|rho|<1 in bins of 0.1, for all subjects with *PC1* score<−3.97 'negative behavioral threshold' set at the 10th percentile ($P_{90-Disc}$) of the distribution and for all patients with *PC1* score > +3.60 ('positive behavioral threshold' set at the 90th percentile ($P_{90-Disc}$) of the distribution). The number of patients accurately selected within each bin is shown in red; the number of inaccurately selected patients is shown in blue. Note that this selection process excludes a number of patients (shown in gray) who do not meet the PC threshold cutoff. Note that as the neural ρ threshold increases the proportion of accurate to inaccurate selections increases and the number of inaccurately selected subjects decreases. For illustrative purposes, a neural threshold of ρ = 0.5 is applied at the point where no inaccurate selections occur for the positive behavioral threshold. (**B**) The neural and symptom score thresholds defined in discovery patients were applied on an independent external 'replication' dataset (N = 69, see **Materials and methods**). For the symptom

*Appendix 1—figure 28 continued*

score (p_{90-Disc} = −3.97 and p_{90-Disc} = +3.60 ) and neural (ρ = 0.5) thresholds as those defined in the discovery sample, no patients are selected for *PC1*. (**C–D**) Similar data to A–B for *PC2*, using PC score thresholds of p_{90-Disc} = −2.43 and p_{90-Disc} = +2.32 and a PC map threshold of ρ = 0.3 on a subset of $P = 68$ optimized parcels. (**E–F**) Similar data to A–B for *PC3* using PC score thresholds of p_{90-Disc} = −2.41 and p_{90-Disc} = +2.17 and a PC map threshold of ρ = 0.4 on a patients of $P = 39$ optimized parcels. (**G–H**) Similar data to A–B for *PC4*, using PC score thresholds of p_{90-Disc} = −1.69 and p_{90-Disc} = +1.72 and a PC map threshold of ρ = 0.2 on a subset of $P = 50$ optimized parcels. (**I–J**) Similar data to A–B for *PC5*, using PC score thresholds of p_{90-Disc} = −1.89 and p_{90-Disc} = +1.47 and a PC map threshold of ρ = 0.4 on a subset of $P = 31$ optimized parcels.

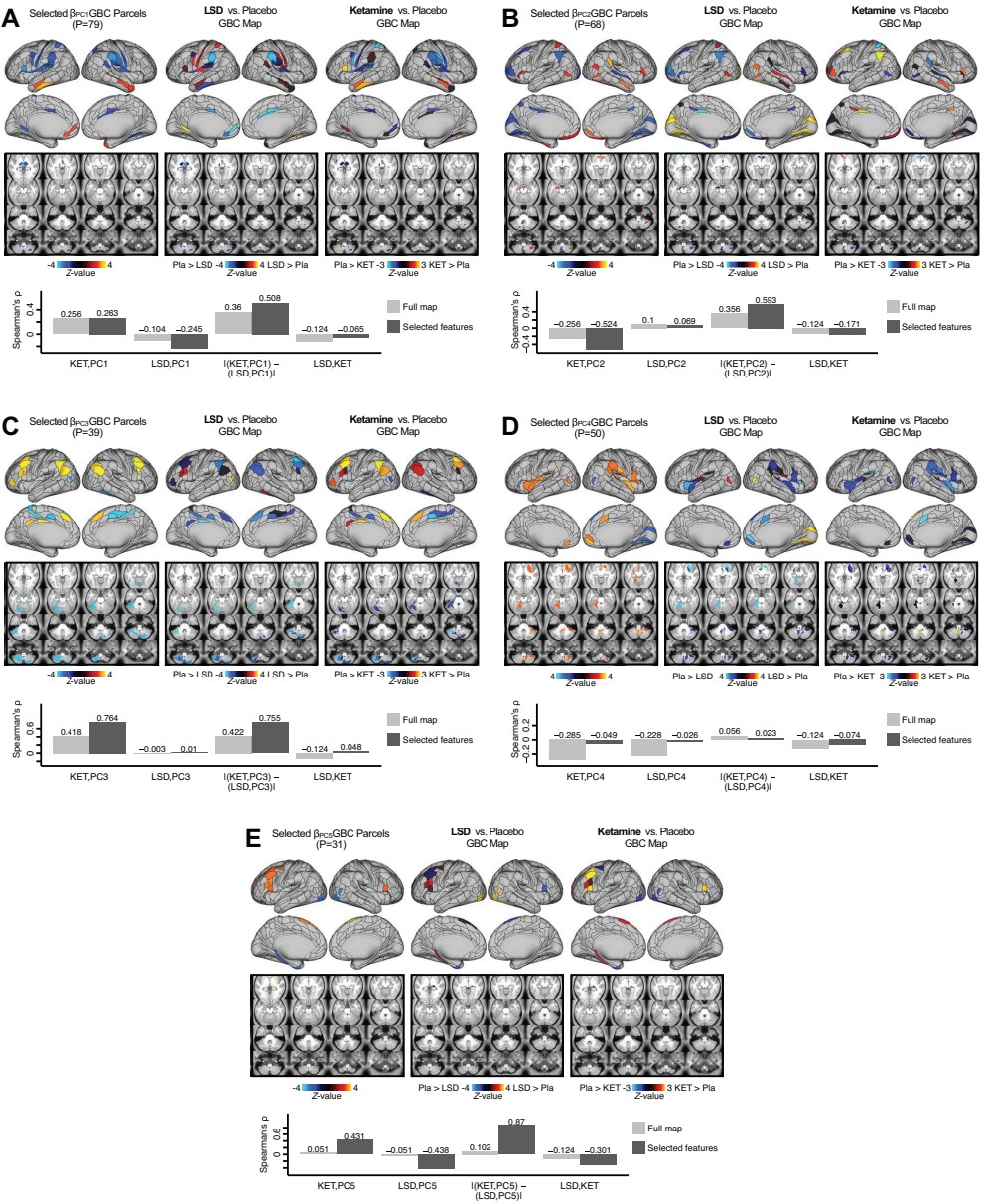

**Appendix 1—figure 29.** Examining most predictive parcels for all *PC*-driven symptom-neural effects and their relationship with pharmacological neuroimaging maps. (**A**) (Left) The selected group-level

$\beta_{PC1}GBC$ map showing the 79 selected parcels derived from the feature selection workflow (*Appendix 1—figure 25*). (Middle) The same parcels are highlighted from the group-level $\Delta GBC$ LSD map reflecting the pharmacological effect relative to placebo. (Right) the group-level ketamine $\Delta GBC$ map reflecting the pharmacological effect relative to placebo. (Bottom bar plot) Here we show four values: (i) Spearman's $\rho$ between the ketamine $\Delta GBC$ and $\beta_{PC1}GBC$ maps (KET,PC1); (ii) Spearman's $\rho$ between the LSD $\Delta GBC$ and $\beta_{PC1}GBC$ maps (LSD,PC1); i(ii) Absolute value of the difference for the (i) and (ii), i.e. differential similarity; |(KET,PC1) - (LSD,PC1)|; (iv) Spearman's $\rho$ between the ketamine $\Delta GBC$ response and LSD ketamine $\Delta GBC$ response maps (KET,LSD). These relationships are quantified within both the selected 79 features (dark gray bars) and the full P=718 neural map (light gray bars). (B–E) These analyses are repeated and shown for *PCs 2–5*. Note that a strong differential similarity for a pair of pharmacological maps may indicate a strategy for patient treatment selection along a particular brain-behavioral axis. Specifically, in this framework patients loading on *PC3* behaviorally (or neurally) would be indicated as highly similar to neural variation induced by ketamine but not LSD. Such metrics may be clinically useful for selecting one pharmacological target over another in the process making individual patient segmentation decisions.

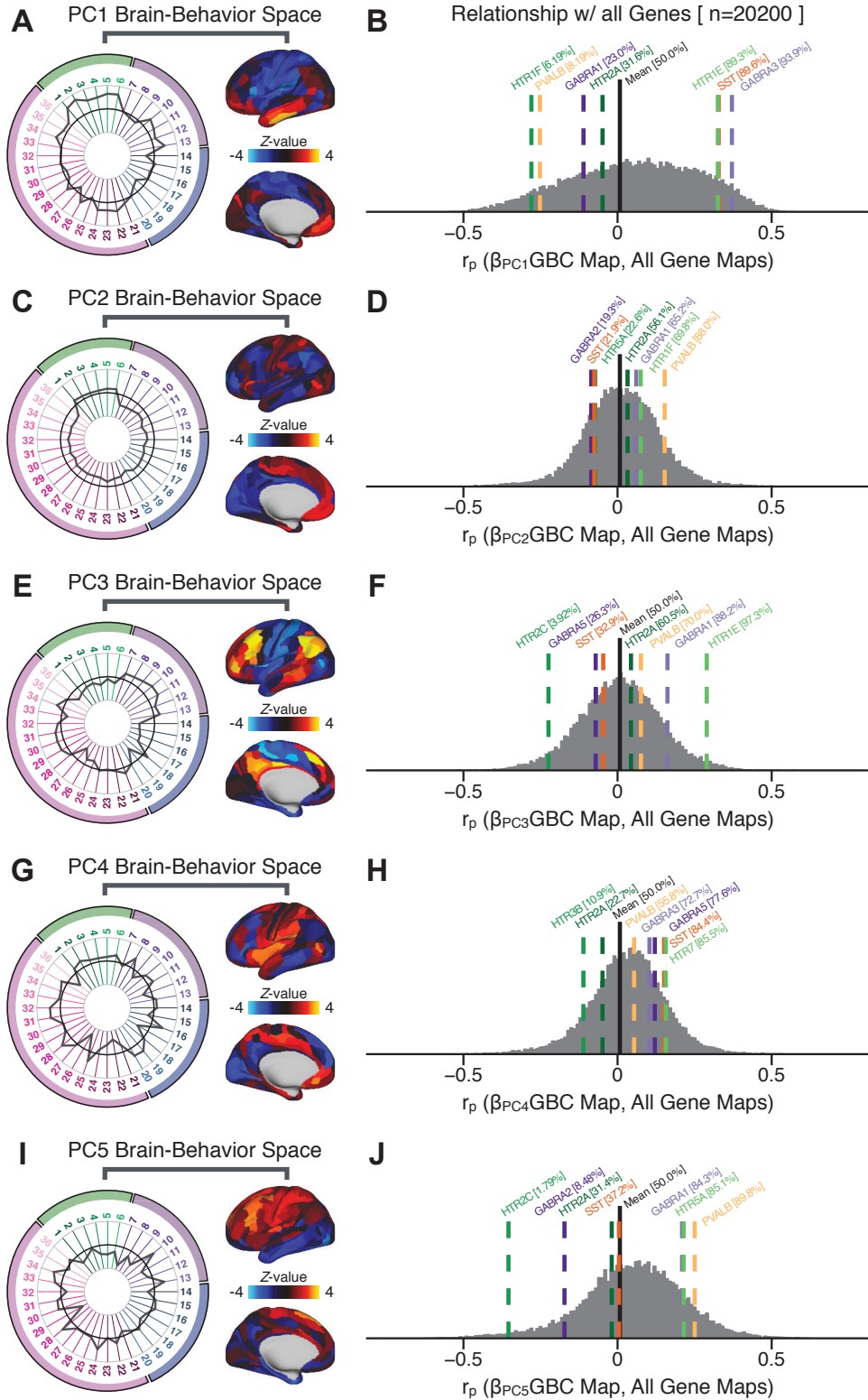

**Appendix 1—figure 30.** Quantifying the relationship between *PC*-derived symptom-neural effects and expression maps for genes implicated in psychosis spectrum disorder (PSD) neurobiology. (**A**) The symptom loading profile (radarplot) and neural loading map derived in the main analyses are

*Appendix 1—figure 30 continued on next page*

*Appendix 1—figure 30 continued*

shown for the *PC1* BBS. (**B**) The distribution shows the Pearson's correlation coefficients between the *PC1* symptom-neural map and all 20,200 gene expression maps derived from the Allen Human Brain Atlas (AHBA, see **Materials and methods** and for details on gene expression analyses see [*Burt et al., 2018*]). In this analysis we focused on a select number of PSD-relevant genes, as highlighted in the main text for PC3 specifically *Figure 8*. Seven genes of interest are shown. Specifically, these include two interneuron marker genes, somatostatin (SST) and parvalbumin (PVALB); two GABA$_A$ receptor subunit genes: GABRA1 and GABRA3; and three serotonin receptor subunit genes: HTR1E, HTR1F, and HTR2A. The percentile for each highlighted gene relative to the entire distribution of 20,200 genes is reported in square brackets. Note that the gene expression maps of HTR1F and PVALB are at the negative tail of the entire distribution, that is, anti-correlated with *PC1* symptom-neural map. Conversely, GABRA3, SST, and HTR1E are on the far positive end, reflecting a highly similar spatial pattern of the gene expression maps with the *PC1* symptom-neural map. (**C–J**) *PC* symptom-neural map profiles and distributions of gene expression map similarities are also shown for PCs 2–5. Note that some of these exemplar genes show a strong spatial similarity (or anti-similarity) with the PC-derived neural maps relative to the entire distribution of all gene-PC map pairs.

