## [Decision Letter]

**Acceptance summary:**

The authors assessed multivariate relations between a dimensionality-reduced symptom space and brain-imaging features, using a large database of individuals with psychosis-spectrum disorders (PSD). Demonstrating both high stability and reproducibility of their approaches, this work showed a promise that diagnosis or treatment of PSD can benefit from a proposed data-driven brain-symptom mapping framework.

**Decision letter after peer review:**

Thank you for submitting your article "Mapping Brain-Behavioral Relationships Along the Psychosis Spectrum" for consideration by *eLife*. Your article has been reviewed by 2 peer reviewers, and the evaluation has been overseen by Drs. Shackman (Reviewing Editor) and Gold (Senior Editor).

– The reviewers expressed some enthusiasm for the report:

– Unique dataset!

– I really appreciate the authors' efforts to demonstrate the robustness of their findings. Overall, this is really great work.

– This paper has the potential to provide helpful guidance to neuroimaging scientists who aim to discover generalizable associations between multimodal imaging and behavior.

– The strength of this study is that (i) in every analysis, the authors provide high-level evidence of reproducibility, (ii) the study included several control analyses to test alternatives or independent techniques (e.g., ICA, univariate vs. multivariate), and (iii) the report demonstrates relations with independently acquired pharmacological neuroimaging and gene expression maps.

– Overall the study has originality and several important tips and guidance for behavior-brain mapping.

– The report details an interesting and rather comprehensive analysis of the trend of developing data-driven methods for developing brain-symptom dimension biomarkers that bring a biological basis to the symptoms (across PANSS and cognitive features) that relate to psychotic disorders.

– Combined analysis of canonical psychosis symptoms and cognitive deficits across multiple traditional psychosis-related diagnoses offers one of the most comprehensive mappings of impairments experienced within PSD to brain features to date.

– Cross-validation analyses and use of various datasets (diagnostic replication, pharmacological neuroimaging) is extremely impressive, well-motivated, and thorough. In addition, the authors use a large dataset and provide "out of sample" validity.

– Medication status and dosage also accounted for.

– Similarly, the extensive examination of both univariate and multivariate neuro-behavioral solutions from a methodological viewpoint, including the testing of multiple configurations of CCA (i.e. with different parcellation granularities), offers very strong support for the selected symptom-to-neural mapping.

– The plots of the obtained PC axes compared to those of standard clinical symptom aggregate scales provide a really elegant illustration of the differences and demonstrate clearly the value of data-driven symptom reduction over conventional categories.

– The comparison of the obtained neuro-behavioral map for the "Psychosis configuration" symptom dimension to both pharmacological neuroimaging and neural gene expression maps highlights direct possible links with both underlying disorder mechanisms and possible avenues for treatment development and application.

– The authors' explicit investigation of whether PSD and healthy controls share a major portion of neural variance (possibly present across all people) has strong implications for future brain-behavior mapping studies, and provides a starting point for narrowing the neural feature space to just the subset of features showing symptom-relevant variance in PSD.

– Overall, the work could advance our knowledge in the development of biomarkers or subject level identifiers for psychiatric disorders and potentially be elevated to the level of an individual "subject screener". While this is a noble goal, this will require more data and information in the future as a means to do this. This is certainly an important step forward in this regard.

Essential revisions:

Nevertheless, both reviewers (and a third, who was unable to provide a full review in a timely manner) expressed some significant, conceptually overlapping concerns.

– Significance/Innovation – As written, it's unclear that the report represents a significant advance.

– As it stands, while the results are very interesting and presented in a quite compelling way, it is unclear whether this is an advance in modeling beyond what has been done before.

– Significance/Aims – Lack of clarity about the aims and goals, which weakens significance and makes the report more challenging to understand.

- A major concern is the emphasis. Is the paper about the methods, or about understanding the neural underpinnings of PSD?

– I was not clear about the target domain of this paper. If this was to show the reproducibility of datamining findings, I guess one can give some other examples, not fully testing on the PSD. But then if I read in that angle, the authors also hold strong domain opinion and knowledge on the PSD. So it would be much straightforward to read if they clarify their stance a bit more between clinical neuroscience and datamining/reproducibility.

Clarity

– Overall I found the paper very hard to read. There are abbreviations everywhere. The paper is methods heavy (which I am not opposed to and quite like). It is clear that the authors took a lot of care in thinking about the methods that were chosen. That said, I think that the organization would benefit from a more traditional Intro, Methods, Results, and Discussion formatting so that it would be easier to parse the Results. The figures are extremely dense and there are often terms that are coined or used that are not or poorly defined.

– The paper contains heavy descriptions about dimensionality reduction (e.g., PCA, ICA, and CCA) and prediction techniques which, at times, made it challenging to read.

Inadequate motivation for targeting the Xia paper as a point of reference, weakening significance.

– One thing I found conceptually difficult is the explicit comparison to the work in the Xia paper from the Satterthwaite group. Is this a fair comparison? The sample demographics are extremely different (e.g. non-clinical, general population, younger). Can it be suggested that the groups that are clinically defined here are comparable? Is this an appropriate comparison and standard to make? To suggest that the work in that paper is not reproducible seems flawed, when viewed in this light.

– The authors' argument seems to hinge on their methods being better insofar as it increases overall reproducibility (a good thing). However, they use the Xia paper as a strawman – this paper is done in a non-clinical, younger sample. Perhaps, it's not that surprising that they cannot replicate.

Key methodological details and motivations for specific choices are lacking.

– Many researcher degrees of freedom in what is reported (e.g., why is PC3 selected? Because it looks like default network? this is not explicitly stated, and selection criteria are not quantified).

– Why was PCA selected for the analysis rather than ICA? Authors mention that PCA enables the discovery of orthogonal symptom dimensions, but don't elaborate on why this is expected to better capture behavioral variation within PSD compared to non-orthogonal dimensions. Given that symptom and/or cognitive items in conventional assessments are likely to be correlated in one way or another, allowing correlations to be present in the low-rank behavioral solution may better represent the original clinical profiles and drive more accurate brain-behavior mapping. Moreover, as alluded to in the Discussion, employing an oblique rotation in the identification of dimensionality-reduced symptom axes may have actually resulted in a brain-behavior space that is more generalizable to other psychiatric spectra. Why not use something more relevant to symptom/behavior data like a factor analysis?

– The gene expression mapping section lacks some justification for why the 7 genes of interest were specifically chosen from among the numerous serotonin and GABA receptors and interneuron markers (relevant for PSD) available in the AHBA. Brief reference to the believed significance of the chosen genes in psychosis pathology would have helped to contextualize the observed relationship with the neuro-behavioral map.

Need to clearly distinguish 'stability' from 'reproducibility'

– To enhance the clarity of the results, the authors need to be more explicit about what they demonstrated is stability or reproducibility in each result section. As discussed on page 18, during the k-fold or leave-one-site out validation, at each iteration the training data for PCA is largely overlapped between the full sample model and the one excluding the held-out data, which makes the result of dimensionality reduction quite similar. In this case, an extremely high prediction-observation correlation is not totally unexpected, and those tests may not be ideal for generalizability but more fit to the stability test (i.e., whether the result is robust against potential outliers). I know that the authors already used the word 'stable' in many sentences, so my comment is only minor, but some sentences "… symptom-derived PCA solution remained highly robust across all sites.…", ".… This similarity was high for all sites (Figure S6).…" (p2) and others give a nuance as if the authors found a replicated effect from each site independently (i.e., PCA on each side), so generalizable. Please note that I am not questioning the fact that the authors demonstrated reproducibility. They did it based on the 'split-half' independent test. The only thing I point out here is that it would be clearer if the study deals with the stability and reproducibility separately across the result section.

Some key aspects of the results need a much more thorough discussion

– Weakness – A lack of overarching interpretation for PC's other than #3. Although the authors provided all supplementary materials for the relationship between behaviors and other principal components than the 3rd one, the current presentation of the results (only the single component throughout the study) makes the study partially complete. Unless the whole point of the study is a thorough data mining test on neuroimaging, I think that the authors could provide an integrative interpretation or discussion on other behavioral PCs that are relevant to explain the PSD behaviors and brain.

– What the identified univariate neuro-behavioral mapping for PC3 ("psychosis configuration") actually means from an empirical or brain network perspective is not really ever discussed in detail. E.g., in Results, "a high positive PC3 score was associated with both reduced GBC across insular and superior dorsal cingulate cortices, thalamus, and anterior cerebellum and elevated GBC across precuneus, medial prefrontal, inferior parietal, superior temporal cortices and posterior lateral cerebellum." While the meaning and calculation of GBC can be gleaned from the Methods, a direct interpretation of the neuro-behavioral results in terms of the types of symptoms contributing to PC3 and relative hyper-/hypo-connectivity of the DMN compared to e.g. healthy controls could facilitate easier comparisons with the findings of past studies (since GBC does not seem to be a very commonly-used measure in the psychosis fMRI literature). Also important since GBC is a summary measure of the average connectivity of a region, and doesn't provide any specificity in terms of which regions in particular are more or less connected within a functional network (an inherent limitation of this measure which warrants further attention).

– While the inclusion of cognitive measures for PSD individuals is a main (self-)selling point of the paper, there's very limited focus on the "Cognitive functioning" component (PC2) of the PCA solution. Examining Figure S8K, the GBC map for this cognitive component seems almost to be the inverse for that of the "Psychosis configuration" component (PC3) focused on in the rest of the paper. Since PC3 does not seem to have high loadings from any of the cognitive items, but it is known that psychosis spectrum individuals tend to exhibit cognitive deficits which also have strong predictive power for illness trajectory, some discussion of how multiple univariate neuro-behavioral features could feasibly be used in conjunction with one another could have been really interesting.

*Reviewer 1:*

The paper assessed the relationship between a dimensionality-reduced symptom space and functional brain imaging features based on the large multicentric data of individuals with psychosis-spectrum disorders (PSD).

The strength of this study is that (i) in every analysis, the authors provided high-level evidence of reproducibility in their findings, (ii) the study included several control analyses to test other comparable alternatives or independent techniques (e.g., ICA, univariate vs. multivariate), and (iii) correlating to independently acquired pharmacological neuroimaging and gene expression maps, the study highlighted neurobiological validity of their results.

Overall the study has originality and several important tips and guidance for behavior-brain mapping, although the paper contains heavy descriptions about data mining techniques such as several dimensionality reduction algorithms (e.g., PCA, ICA, and CCA) and prediction models.

Although relatively minors, I also have few points on the weaknesses, including (i) an incomplete description about how to tell the PSD effects from the normal spectrum, (ii) a lack of overarching interpretation for other principal components rather than only the third one, and (iii) somewhat expected results in the stability of PC and relevant indices.

*Reviewer 2:*

The work by Ji et al., is an interesting and rather comprehensive analysis of the trend of developing data-driven methods for developing brain-symptom dimension biomarkers that bring a biological basis to the symptoms (across PANSS and cognitive features) that relate to psychotic disorders. To this end, the authors performed several interesting multivariate analyses to decompose the symptom/behavioural dimensions and functional connectivity data. To this end, the authors use data from individuals from a transdiagnostic group of individuals recruited by the BSNIP cohort and combine high-level methods in order to integrate both types of modalities. Conceptually there are several strengths to this paper that should be applauded. However, I do think that there are important aspects of this paper that need revision to improve readability and to better compare the methods to what is in the field and provide a balanced view relative to previous work with the same basic concepts that they are building their work around. Overall, I feel as though the work could advance our knowledge in the development of biomarkers or subject level identifiers for psychiatric disorders and potentially be elevated to the level of an individual "subject screener". While this is a noble goal, this will require more data and information in the future as a means to do this. This is certainly an important step forward in this regard.

Strengths:

– Combined analysis of canonical psychosis symptoms and cognitive deficits across multiple traditional psychosis-related diagnoses offers one of the most comprehensive mappings of impairments experienced within PSD to brain features to date.

– Cross-validation analyses and use of various datasets (diagnostic replication, pharmacological neuroimaging) is extremely impressive, well motivated, and thorough. In addition the authors use a large dataset and provide "out of sample" validity.

– Medication status and dosage also accounted for.

– Similarly, the extensive examination of both univariate and multivariate neuro-behavioural solutions from a methodological viewpoint, including the testing of multiple configurations of CCA (i.e. with different parcellation granularities), offers very strong support for the selected symptom-to-neural mapping.

– The plots of the obtained PC axes compared to those of standard clinical symptom aggregate scales provide a really elegant illustration of the differences and demonstrate clearly the value of data-driven symptom reduction over conventional categories

- The comparison of the obtained neuro-behavioural map for the "Psychosis configuration" symptom dimension to both pharmacological neuroimaging and neural gene expression maps highlights direct possible links with both underlying disorder mechanisms and possible avenues for treatment development and application.

- The authors' explicit investigation of whether PSD and healthy controls share a major portion of neural variance (possibly present across all people) has strong implications for future brain-behaviour mapping studies, and provides a starting point for narrowing the neural feature space to just the subset of features showing symptom-relevant variance in PSD

Critiques:

– Overall I found the paper very hard to read. There are abbreviation everywhere for every concept that is introduced. The paper is methods heavy (which I am not opposed to and quite like). It is clear that the authors took a lot of care in thinking about the methods that were chosen. That said, I think that the organization would benefit from a more traditional Intro, Methods, Results, and Discussion formatting so that it would be easier to parse the Results. The figures are extremely dense and there are often terms that are coined or used that are not or poorly defined.

– One thing I found conceptually difficult is the explicit comparison to the work in the Xia paper from the Satterthwaite group. Is this a fair comparison? The sample is extremely different as it is non clinical and comes from the general population. Can it be suggested that the groups that are clinically defined here are comparable? Is this an appropriate comparison and standard to make. To suggest that the work in that paper is not reproducible is flawed in this light.

– Why was PCA selected for the analysis rather than ICA? Authors mention that PCA enables the discovery of orthogonal symptom dimensions, but don't elaborate on why this is expected to better capture behavioural variation within PSD compared to non-orthogonal dimensions. Given that symptom and/or cognitive items in conventional assessments are likely to be correlated in one way or another, allowing correlations to be present in the low-rank behavioural solution may better represent the original clinical profiles and drive more accurate brain-behaviour mapping. Moreover, as alluded to in the Discussion, employing an oblique rotation in the identification of dimensionality-reduced symptom axes may have actually resulted in a brain-behaviour space that is more generalizable to other psychiatric spectra. Why not use something more relevant to symptom/behaviour data like a factor analysis?

– The gene expression mapping section lacks some justification for why the 7 genes of interest were specifically chosen from among the numerous serotonin and GABA receptors and interneuron markers (relevant for PSD) available in the AHBA. Brief reference to the believed significance of the chosen genes in psychosis pathology would have helped to contextualize the observed relationship with the neuro-behavioural map.

– What the identified univariate neuro-behavioural mapping for PC3 ("psychosis configuration") actually means from an empirical or brain network perspective is not really ever discussed in detail. E.g., in Results, "a high positive PC3 score was associated with both reduced GBC across insular and superior dorsal cingulate cortices, thalamus, and anterior cerebellum and elevated GBC across precuneus, medial prefrontal, inferior parietal, superior temporal cortices and posterior lateral cerebellum." While the meaning and calculation of GBC can be gleaned from the Methods, a direct interpretation of the neuro-behavioural results in terms of the types of symptoms contributing to PC3 and relative hyper-/hypo-connectivity of the DMN compared to e.g. healthy controls could facilitate easier comparisons with the findings of past studies (since GBC does not seem to be a very commonly-used measure in the psychosis fMRI literature). Also important since GBC is a summary measure of the average connectivity of a region, and doesn't provide any specificity in terms of which regions in particular are more or less connected within a functional network (an inherent limitation of this measure which warrants further attention).

– Possibly a nitpick, but while the inclusion of cognitive measures for PSD individuals is a main (self-)selling point of the paper, there's very limited focus on the "Cognitive functioning" component (PC2) of the PCA solution. Examining Figure S8K, the GBC map for this cognitive component seems almost to be the inverse for that of the "Psychosis configuration" component (PC3) focused on in the rest of the paper. Since PC3 does not seem to have high loadings from any of the cognitive items, but it is known that psychosis spectrum individuals tend to exhibit cognitive deficits which also have strong predictive power for illness trajectory, some discussion of how multiple univariate neuro-behavioural features could feasibly be used in conjunction with one another could have been really interesting.

Another nitpick, but the Y axes of Figure 8C-E are not consistent, which causes some of the lines of best fit to be a bit misleading (e.g. GABRA1 appears to have a more strongly positive gene-PC relationship than HTR1E, when in reality the opposite is true.)

– The authors explain the apparent low reproducibility of their multivariate PSD neuro-behavioural solution using the argument that many psychiatric neuroimaging datasets are too small for multivariate analyses to be sufficiently powered. Applying an existing multivariate power analysis to their own data as empirical support for this idea would have made it even more compelling. The following paper suggests guidelines for sample sizes required for CCA/PLS as well as a multivariate calculator: Helmer et al., (2020). On stability of Canonical Correlation Analysis and Partial Least Squares with application to brain-behavior associations (p. 2020.08.25.265546). https://doi.org/10.1101/2020.08.25.265546

– Given the relatively even distribution of males and females in the dataset, some examination of sex effects on symptom dimension loadings or neuro-behavioural maps would have been interesting (other demographic characteristics like age and SES are summarized for subjects but also not investigated). I think this is a missed opportunity.

---

## [Author Response]

The reviewers expressed some enthusiasm for the report.[…]– Overall, the work could advance our knowledge in the development of biomarkers or subject level identifiers for psychiatric disorders and potentially be elevated to the level of an individual "subject screener". While this is a noble goal, this will require more data and information in the future as a means to do this. This is certainly an important step forward in this regard.

We are very grateful for the positive feedback and the careful read of our paper. We would especially like to thank the Reviewers for taking the time to read this admittedly lengthy and complex manuscript and for providing their helpful and highly constructive feedback. Overall, we hope the Editor and the Reviewers will find that our responses address all the comments and that the requested changes and edits improved the paper.

Essential revisions:Nevertheless, both reviewers (and a third, who was unable to provide a full review in a timely manner) expressed some significant, conceptually overlapping concerns.– Significance/Innovation – As written, it's unclear that the report represents a significant advance.– As it stands, while the results are very interesting and presented in a quite compelling way, it is unclear whether this is an advance in modeling beyond what has been done before.

We appreciate the Reviewer’s perspective that there is a need to add more clarity around what precise element of this study represents a significant extension versus a key innovation. We sincerely hope that the revised Introduction highlights the innovation with more clarity such that our framework and results can be seen as fundamentally innovative with important implications for precision medicine in psychiatry. Briefly, the key innovation we now highlight is the ’precision’ element of the paper, with the focus on obtaining reproducible neuro-behavioral models that can yield single-subject predictive potential. We have rewritten parts of the Introduction and Discussion to highlight these points.

Specifically, we highlight the following innovations of the current study:

– The low-dimensional symptom solution across a broad PSD sample is stable, reproducible, encompassing of major diagnostic categories, and covers a broad range of PSD symptoms, including canonical psychosis symptoms as well as cognitive deficits (the latter being an innovation that one of the Reviewers actually singled out – see p. 1 of the response letter).

– This data-reduced symptom geometry can be mapped robustly on neural features in a way that is reproducible at the group level (i.e. the neural map replicates across split-half cross validations).

– The identified neuro-behavioral solution underwent a subsequent feature selection to achieve individual level precision for a subset of parcels. This type of feature reduction, going from a reproducible full neuro-behavioral map, to map a subset of individually predictive features, has not been attempted to our knowledge in any study relevant to psychopathology.

– While other studies have evaluated relationships between brain maps and complementary molecular datasets (e.g. transcriptomics), no study to our knowledge has benchmarked single-subject selected neural maps using reproducible neuro-behavioral features against both pharmacological neuroimaging maps and gene expression maps that may inform single subject selection for targeted treatment.

Introduction:

“To our knowledge, no study to date has mapped a data-reduced symptom geometry encompassing a transdiagnostic PSD cohort across combined cognitive and psychopathology symptom domains, with demonstrable statistical stability and reproducibility at the single subject level. Additionally, this symptom geometry can be mapped robustly onto neural data to achieve reproducible group-level effects as well as individual-level precision following feature selection. Furthermore, while other other studies have evaluated relationships between neural maps and complementary molecular datasets (e.g. transcriptomics), no study to our knowledge has benchmarked single-subject selected neural maps using reproducible neuro-behavioral features against both pharmacological neuroimaging maps and gene expression maps that may inform single subject selection for targeted treatment. Collectively, this study used data-driven dimensionality-reduction methods to derive stable orthogonal symptom dimensions across 436 PSD patients and mapped them onto novel and stable neuroimaging features. These effects were then benchmarked against molecular imaging targets, outlining an actionable path towards making personalized recommendations for clinical endpoints.”

Discussion:

“We demonstrate, for the first time in the literature, a transdiagnostic data-reduced PSD symptom geometry across hallmark psychopathology symptoms and cognitive measures, that shows stable and robust relationships with neural connectivity with singles subject-level precision. Critically, we anchor these symptom-neural maps to mechanistically-informed molecular pharmacological and gene expression benchmarks that can be linked to putative pathophysiology treatment mechanism to inform targeted patient selection decisions.”

– Significance/Aims – Lack of clarity about the aims and goals, which weakens significance and makes the report more challenging to understand.- A major concern is the emphasis. Is the paper about the methods, or about understanding the neural underpinnings of PSD?– I was not clear about the target domain of this paper. If this was to show the reproducibility of datamining findings, I guess one can give some other examples, not fully testing on the PSD. But then if I read in that angle, the authors also hold strong domain opinion and knowledge on the PSD. So it would be much straightforward to read if they clarify their stance a bit more between clinical neuroscience and datamining/reproducibility.

We thank the Reviewers for bringing up this point. We now more explicitly specify the core goals of the paper, in line with the innovations articulated above. The primary goal of this study is to derive a comprehensive, reproducible symptom-neural geometry across cognitive and psychopathology symptom domains in chronic PSD patients, that can be resolved at the single-subject level and mechanistically linked to molecular benchmarks. However, achieving this goal necessitates the development of novel methods and rigorous testing, which we have also presented in the paper for transparency and reproducibility. These methods can be iteratively improved upon to further elucidate the neural underpinnings of PSD and further expanded to generalize to other psychiatric spectra, symptom domains, and neural features. We have clarified this throughout the text and in particular the Introduction:

“The primary purpose of this paper is to derive a comprehensive, stable, and reproducible symptom-neural geometry across symptom domains in chronic PSD, that can be resolved at the single-subject level and mechanistically linked to molecular benchmarks. This necessitated the development of novel methods and rigorous testing (as we found that existing multivariate methods are drastically underpowered (1)) which we have also presented in the paper for transparency and reproducibility. This approach can be iteratively improved upon to further elucidate the neural underpinnings of PSD variation, and expanded to apply to broader research questions and psychiatric spectra, symptom domains, and neural features.”

Clarity– Overall I found the paper very hard to read. There are abbreviations everywhere. The paper is methods heavy (which I am not opposed to and quite like). It is clear that the authors took a lot of care in thinking about the methods that were chosen. That said, I think that the organization would benefit from a more traditional Intro, Methods, Results, and Discussion formatting so that it would be easier to parse the Results. The figures are extremely dense and there are often terms that are coined or used that are not or poorly defined.

We appreciate the constructive feedback around how to remove the dense content and to pay more attention to the frequency of abbreviations, which impact readability. We implemented the strategies suggested by the Reviewer and have moved the Methods section after the Introduction to make the subsequent Results section easier to understand and contextualize. For clarity and length, we have moved methodological details previously in the Results and figure captions to the Methods (e.g. descriptions of dimensionality reduction and prediction techniques). This way, the Methods are now expanded for clarity without detracting from the readability of the core results of the paper. Also, we have also simplified the text in places where there was room for more clarity. For convenience and ease of use of the numerous abbreviations, we have also added a table to the Supplement (Supplementary Table S1):

– The paper contains heavy descriptions about dimensionality reduction (e.g., PCA, ICA, and CCA) and prediction techniques which, at times, made it challenging to read.

We have removed these descriptions out of the main Results text and figure captions. Detailed descriptions are still provided in the Methods, so that they do not detract from the core message of the paper but can still be referenced if a reader wishes to look up the details of these methods within the context of our analyses.

– Inadequate motivation for targeting the Xia paper as a point of reference, weakening significance.– One thing I found conceptually difficult is the explicit comparison to the work in the Xia paper from the Satterthwaite group. Is this a fair comparison? The sample demographics are extremely different (e.g. non-clinical, general population, younger). Can it be suggested that the groups that are clinically defined here are comparable? Is this an appropriate comparison and standard to make? To suggest that the work in that paper is not reproducible seems flawed, when viewed in this light.

This is an extremely important point to clarify and we apologize that we did not make it sufficiently clear in the initial submission. Here we are not attempting to replicate the results of Xia et al., which we understand were derived in a fundamentally different sample than ours both demographically and clinically, with testing very different questions. Rather, this paper is just one example out of a number of recent papers which employed multivariate methods (CCA) to tackle the mapping between neural and behavioral features. The key point here is that this approach does not produce reproducible results due to over-fitting, as demonstrated robustly in the present paper. It is very important to highlight that in fact we did not single out any one paper when making this point. In fact, we do not mention the Xia paper explicitly anywhere and we were very careful to cite multiple papers in support of the multivariate over-fitting argument, which is now a well-known issue (2). Nevertheless, the Reviewers make an excellent point here and we acknowledge that while CCA was not reproducible in the present dataset, this does not explicitly imply that the results in the Xia et al., paper (or any other paper for that matter) are not reproducible by definition (i.e. until someone formally attempts to falsify them). We have made this point explicit in the revised paper, as shown in the next response below. Furthermore, in line with the provided feedback, we also applied the multivariate power calculator derived by Helmer et al., (1), which quantitatively illustrates the statistical point around CCA instability.

– The authors' argument seems to hinge on their methods being better insofar as it increases overall reproducibility (a good thing). However, they use the Xia paper as a strawman – this paper is done in a non-clinical, younger sample. Perhaps, it's not that surprising that they cannot replicate.

As noted above, this is an extremely important point to clarify and we apologize that we did not make it sufficiently clear in the initial submission. As we articulated in the response above, we are not attempting a replication of Xia et al., 2018 and we did not single out any particular paper when making our case regarding over-fitting in CCA. We have clarified in the revised manuscript:

Results:

“Several recent studies have reported “latent” neuro-behavioral relationships using multivariate statistics (3–5), which would be preferable because they simultaneously solve for maximal covariation across neural and behavioral features. Given the possibility of deriving a stable multivariate effect, here we tested if results improve with canonical correlation analysis (CCA) (6) (Figure 5A)."

Discussion:

“Here we attempted to use multivariate solutions (i.e. CCA) to quantify symptom and neural feature co-variation. In principle, CCA is theoretically well-suited to address the brain-behavioral mapping problem. However, symptom-neural mapping using CCA in our sample was not reproducible even when using a low-dimensional symptom solution and parcellated neural data as a starting point. Therefore, while CCA (and related multivariate methods such as partial least squares) are theoretically appropriate and may be helped by regularization methods such as sparse CCA, in practice many available psychiatric neuroimaging datasets may not provide sufficient power to resolve stable multivariate symptom-neural solutions (1). A key pressing need for forthcoming studies will be to use multivariate power calculators to inform sample sizes needed for resolving stable symptom-neural geometries at the single subject level. Of note, though we were unable to derive a stable CCA in the present sample, this does not imply that the multivariate neuro-behavioral effect may not be reproducible with larger effect sizes and/or sample sizes. Critically, this does highlight the importance of power calculations prior to computing multivariate brain-behavioral solutions (1).”

Key methodological details and motivations for specific choices are lacking.– Many researcher degrees of freedom in what is reported (e.g., why is PC3 selected? Because it looks like default network? this is not explicitly stated, and selection criteria are not quantified).

We appreciate the Reviewer’s point and the importance of justifying the focus on PC3 more explicitly. To be clear, we do present the entire set of PC neuro-behavioral maps in the Supplement. We decided to focus our Results section on PC3 in the original submission because of the obtained pattern of behavioral loadings mapped onto hallmark symptoms of PSD, including strong loadings across both Positive and Negative symptom items in the PANSS. In contrast PC1 and PC2 were associated with a more general impairment index and cognitive deficits respectively. We found that the focus on PC3 was innovative because it provided an opportunity to quantify a fully data-driven dimension of symptom variation that is highly characteristic of the PSD patient population. Additionally, this bi-directional symptom axis captured shared variance from measures in other traditional symptoms factors, such the PANSS General factor and cognition. We found that the PC3 result provides a powerful empirical demonstration of how using a data-driven low-dimensional solution (via PCA) can reveal novel patterns intrinsic to the structure of PSD psychopathology, which in turn does improve the mapping of symptom-to-neural relationships. Therefore, given space considerations, we opted to not fully characterize the reproducibility of all five PCs in detail because it would further extend an already lengthy, complex, and dense study (which the Reviewers’ themselves acknowledge). Instead, we show the data for all PCs across all analyses in the Supplement. We hope that this argument provides sufficient rationale for focusing the methodological and empirical analyses of the main text of the paper on the most highly innovative finding. We have now stated this more explicitly in the relevant section of the Results:

“Because *PC3* loads most strongly on to hallmark symptoms of PSD (including strong positive loadings across PANSS Positive symptom measures in the PANSS and strong negative loadings onto most Negative measures), we focus on this PC as an opportunity to quantify an innovative, fully data-driven dimension of symptom variation that is highly characteristic of the PSD patient population. Additionally, this bi-directional symptom axis captured shared variance from measures in other traditional symptoms factors, such the PANSS General factor and cognition. We found that the *PC3* result provided a powerful empirical demonstration of how using a data-driven dimensionality-reduced solution (via PCA) can reveal novel patterns intrinsic to the structure of PSD psychopathology.”

– Why was PCA selected for the analysis rather than ICA? Authors mention that PCA enables the discovery of orthogonal symptom dimensions, but don't elaborate on why this is expected to better capture behavioral variation within PSD compared to non-orthogonal dimensions. Given that symptom and/or cognitive items in conventional assessments are likely to be correlated in one way or another, allowing correlations to be present in the low-rank behavioral solution may better represent the original clinical profiles and drive more accurate brain-behavior mapping. Moreover, as alluded to in the Discussion, employing an oblique rotation in the identification of dimensionality-reduced symptom axes may have actually resulted in a brain-behavior space that is more generalizable to other psychiatric spectra. Why not use something more relevant to symptom/behavior data like a factor analysis?

This is a very important point! We agree with the Reviewer that an oblique solution may better fit the data. For this reason, we performed an ICA as shown in the Supplement. We chose to show PCA for the main analyses here because it is a deterministic solution and the number of significant components could be computed via permutation testing. Importantly, certain components from the ICA solution in this sample were highly similar to the PCs shown in the main solution (Supplementary Note 1), as measured by comparing the subject behavioral scores (Figure S4), and neural maps (Figure S11). However, notably, certain components in the ICA and PCA solutions did not appear to have a one-to-one mapping (e.g. PCs 1-3 and ICs 1-3). The orthogonality of the PCA solution forces the resulting components to capture maximally separated, unique symptom variance, which in turn map robustly on to unique neural circuits. We observed that the data may be distributed in such a way that in the ICA highly correlated independent components emerge, which do not maximally separate the symptom variance associate with neural variance. We demonstrate this by plotting the relationship between parcel β coefficients for the *β_PC_*_3_*GBC* map versus the *β_IC_*_2_*GBC* and *β_IC_*_3_*GBC* maps. The sigmoidal shape of the distribution indicates an improvement in the Z-statistics for the *β_PC_*_3_*GBC* map relative to the *β_IC_*_2_*GBC* and *β_IC_*_3_*GBC* maps. We have added this language to the main text Results:

“Notably, independent component analysis (ICA), an alternative dimensionality reduction procedure which does not enforce component orthogonality, produced similar effects for this PSD sample (see Supplementary Note 1 & Figure S4A). Certain pairs of components between the PCA and ICA solutions appear to be highly similar and exclusively mapped (IC5 and PC4; IC4 and PC5) (Figure S4B). On the other hand, PCs 1-3 and ICs 1-3 do not exhibit a one-to-one mapping. For example, *PC3* appears to correlate positively with *IC2* and equally strongly negatively with *IC3*, suggesting that these two ICs are oblique to the PC and perhaps reflect symptom variation that is explained by a single PC. The orthogonality of the PCA solution forces the resulting components to capture maximally separated, unique symptom variance, which in turn map robustly on to unique neural circuits. We observed that the data may be distributed in such a way that in the ICA highly correlated independent components emerge, which do not maximally separate the symptom variance associate with neural variance. We demonstrate this by plotting the relationship between parcel β coefficients for the *β_PC_*_3_*GBC* map versus the *β_IC_*_2_*GBC* and *β_IC_*_3_*GBC* maps (Figure ?G). The sigmoidal shape of the distribution indicates an improvement in the Z-statistics for the *β_PC_*_3_*GBC* map relative to the *β_IC_*_2_*GBC* and *β_IC_*_3_*GBC* maps.”

Additionally, the Reviewer raises an important point, and we agree that orthogonal versus oblique solutions warrant further investigation especially with regards to other psychiatric spectra and/or other stages in disease progression. For example, oblique components may better capture dimensions of behavioral variation in prodromal individuals, as these individuals are in the early stages of exhibiting psychosis-relevant symptoms and may show early diverging of dimensions of behavioral variation. We elaborate on this further in the Discussion:

“Another important aspect that will require further characterization is the possibility of oblique axes in the symptom-neural geometry. While orthogonal axes derived via PCA were appropriate here and similar to the ICA-derived axes in this solution, it is possible that oblique dimensions more clearly reflect the geometry of other psychiatric spectra and/or other stages in disease progression. For example, oblique components may better capture dimensions of neuro-behavioral variation in a sample of prodromal individuals, as these patients are exhibiting early-stage psychosis-like symptoms and may show signs of diverging along different trajectories.

Critically, these factors should constitute key extensions of an iteratively more robust model for individualized symptom-neural mapping across the PSD and other psychiatric spectra. Relatedly, it will be important to identify the ‘limits’ of a given BBS solution – namely a PSD-derived effect may not generalize into the mood spectrum (i.e. both the symptom space and the resulting symptom-neural mapping is orthogonal). It will be important to evaluate if this framework can be used to initialize symptom-neural mapping across other mental health symptom spectra, such as mood/anxiety disorders.”

– The gene expression mapping section lacks some justification for why the 7 genes of interest were specifically chosen from among the numerous serotonin and GABA receptors and interneuron markers (relevant for PSD) available in the AHBA. Brief reference to the believed significance of the chosen genes in psychosis pathology would have helped to contextualize the observed relationship with the neuro-behavioral map.

We thank the Reviewer for providing this suggestion and agree that it will strengthen the section on gene expression analysis. Of note, we did justify the choice for these genes, but we appreciate the opportunity to expand on the neurobiology of selected genes and their relevance to PSD. We have made these edits to the text:

“We focus here on serotonin receptor subunits (HTR1E, HTR2C, HTR2A), GABA receptor subunits (GABRA1, GABRA5), and the interneuron markers somatostatin (SST) and parvalbumin (PVALB). Serotonin agonists such as LSD have been shown to induce PSD-like symptoms in healthy adults (7) and the serotonin antagonism of “second-generation” antipsychotics are thought to contribute to their efficacy in targeting broad PSD symptoms (8–10). Abnormalities in GABAergic interneurons, which provide inhibitory control in neural circuits, may contribute to cognitive deficits in PSD (11–13) and additionally lead to downstream excitatory dysfunction that underlies other PSD symptoms (14, 15). In particular, a loss of prefrontal parvalbumin-expression fast-spiking interneurons has been implicated in PSD (16–19).”

Need to clearly distinguish 'stability' from 'reproducibility'– To enhance the clarity of the results, the authors need to be more explicit about what they demonstrated is stability or reproducibility in each result section. As discussed on page 18, during the k-fold or leave-one-site out validation, at each iteration the training data for PCA is largely overlapped between the full sample model and the one excluding the held-out data, which makes the result of dimensionality reduction quite similar. In this case, an extremely high prediction-observation correlation is not totally unexpected, and those tests may not be ideal for generalizability but more fit to the stability test (i.e., whether the result is robust against potential outliers). I know that the authors already used the word 'stable' in many sentences, so my comment is only minor, but some sentences "… symptom-derived PCA solution remained highly robust across all sites.…", ".… This similarity was high for all sites (Figure S6).…" (p2) and others give a nuance as if the authors found a replicated effect from each site independently (i.e., PCA on each side), so generalizable. Please note that I am not questioning the fact that the authors demonstrated reproducibility. They did it based on the 'split-half' independent test. The only thing I point out here is that it would be clearer if the study deals with the stability and reproducibility separately across the result section.

This is excellent feedback and we appreciate the need for more precise use of these terms. We now use the term “stable” to describe results using overlapping samples in the cross-validation tests and the term “reproducible” to describe results from independent non-overlapping split-half samples. We have now adjusted this wording throughout and also edited the pertinent Results and Methods sections to distinguish clearly between within-sample stability assessments and independent replication assessments:

Results:

“Next, we show that the PCA solution was highly *stable* when tested across sites, and k-fold crossvalidations, and was *reproducible* in independent split-half samples. First, we show that the symptom-derived PCA solution remained highly robust across all sites (Figure 2A) and 5-fold crossvalidation iterations (see Methods). Notably, the total proportion of variance explained as well as the total number of significant PCs remained stable (Figure 2A). Second, PCA loadings accurately and reliably computed the scores of subjects in a hold-out sample (Figure 2B). Specifically, for each left-out site, we tested if single-subject predicted scores were similar to the originally observed scores of these same subjects in the full PCA (i.e. performed with all 436 subjects). This similarity was high for all sites (Figure S6). Third, we verified that the predicted-versus-observed PCA score similarities also hold via a *k*-fold cross-validation (for *k*=2 to 10, Figure 2C & Figure S5). Finally, across all 5 PCs, PCA loadings were stable when testing via leave-site-out analysis and 5-fold cross-validation. We furthermore demonstrated the reproducibility of the solution using 1,000 independent split-half replications (Figure 2D).”

Methods:

“Stability and Reproducibility of the Symptom PCA. Stability of the symptom PCA solution was assessed with within-sample cross-validation. To evaluate if there were site differences that uniquely drove the PCA solution, we performed a leave-site-out cross-validation analysis. Specifically, we re-ran the PCA using all patients except those from one site, which was held out. This process was repeated for each of the 6 sites. To further evaluate the stability of the derived PCA solutions, we performed 1,000 runs of *k*-fold cross-validation for values of *k* between 2 and 10. For each *k*-fold run, the full sample of patients was randomly divided into equally-sized *k* sets and a PCA was re-computed using subjects in *k*-1 sets (as the left-out set was used as the held-out sample). We also assessed the reproducibility of the PCA solution using independent split-half samples. The leave-site-out and *k*-fold cross-validation PCA analyses by definition use overlapping samples of patients for each iteration. Therefore, we additionally conducted a full split-half replication using entirely non-overlapping sets of patients in each iteration.”

Some key aspects of the results need a much more thorough discussion– Weakness – A lack of overarching interpretation for PC's other than #3. Although the authors provided all supplementary materials for the relationship between behaviors and other principal components than the 3rd one, the current presentation of the results (only the single component throughout the study) makes the study partially complete. Unless the whole point of the study is a thorough data mining test on neuroimaging, I think that the authors could provide an integrative interpretation or discussion on other behavioral PCs that are relevant to explain the PSD behaviors and brain.

We appreciate the feedback. As we mention above we spotlighted PC3 because we felt it was the most critical to fully explore for the clinical interest of the study, given that it most strongly loads on hallmark symptoms of PSD. We also use PC5 in the main text specifically to demonstrate relating PC symptom maps to neuropharmacological maps for informing single-subject treatment response selection and Figure 7. However,

we agree that it is important to provide an integrative interpretation of the full PCA solution. We had originally included a brief interpretation of these PCs behaviorally in Supplementary Note 2: Interpretation and Naming of the Symptom PCA Solution and have now moved this to the main text Results.

“Figure 1E highlights loading configurations of symptom measures forming each PC. To aid interpretation, we assigned a name for each PC based on its most strongly weighted symptom measures. This naming is qualitative but informed by the pattern of loadings of the original 36 symptom measures (Figure 1). For example, *PC1* was highly consistent with a general impairment dimension (i.e. “Global Functioning”); *PC2* reflected more exclusively variation in cognition (i.e. “Cognitive Functioning”); *PC3* indexed a complex configuration of psychosis-spectrum relevant items (i.e. “Psychosis Configuration”); *PC4* generally captured variation mood and anxiety related items (i.e. “Affective Valence”); finally, *PC5* reflected variation in arousal and level of excitement (i.e. “Agitation/Excitation”). For instance, a generally impaired patient would have a highly negative *PC1* score, which would reflect low performance on cognition and elevated scores on most other symptomatic items. Conversely, an individual with a high positive *PC3* score would exhibit delusional, grandiose, and/or hallucinatory behavior, whereas a person with a negative *PC3* score would exhibit motor retardation, social avoidance, possibly a withdrawn affective state with blunted affect (20). Comprehensive loadings for all 5 PCs are shown in Figure 3G. Figure 1F highlights the mean of each of the 3 diagnostic groups (colored spheres) and healthy controls (black sphere) projected into a 3-dimensional orthogonal coordinate system for *PCs 1,2 & 3* (x,y,z axes respectively; alternative views of the 3-dimensional coordinate system with all patients projected are shown in Figure 3). Critically, PC axes were not parallel with traditional aggregate symptom scales. For instance, *PC3* is angled at ∼45 to the dominant direction of PANSS Positive and Negative symptom variation (purple and blue arrows respectively in Figure 1F).

[…] Because *PC3* loads most strongly on to hallmark symptoms of PSD (including strong positive loadings across PANSS Positive symptom measures in the PANSS and strong negative loadings onto most Negative measures), we focus on this PC as an opportunity to quantify an innovative, fully data-driven dimension of symptom variation that is highly characteristic of the PSD patient population. Additionally, this bi-directional symptom axis captured shared variance from measures in other traditional symptoms factors, such the PANSS General factor and cognition. We found that the *PC3* result provided a powerful empirical demonstration of how using a data-driven dimensionality-reduced solution (via PCA) can reveal novel patterns intrinsic to the structure of PSD psychopathology.”

We also have added a section in the Discussion to include discussion of the PC neural maps:

“The symptom-neural mapping of dimensionality-reduced symptom scores also produced notable neural maps along other *PC* axes (Figure S10). Higher *PC1* scores, indicating higher general functioning, may be associated with lower global connectivity in sensory/cingulate cortices and thalamus, but higher global connectivity with the temporal lobe. […] Interestingly, however, the nucleus accumbens (as well as hippocampus, amygdala, cerebellum, and temporal lobe) appear to be negatively associated with *PC5* score (i.e. more severe grandiosity, unusual thought, delusional and disorganized thinking), illustrating another example of a bi-directional axis.”

– What the identified univariate neuro-behavioral mapping for PC3 ("psychosis configuration") actually means from an empirical or brain network perspective is not really ever discussed in detail. E.g., in Results, "a high positive PC3 score was associated with both reduced GBC across insular and superior dorsal cingulate cortices, thalamus, and anterior cerebellum and elevated GBC across precuneus, medial prefrontal, inferior parietal, superior temporal cortices and posterior lateral cerebellum." While the meaning and calculation of GBC can be gleaned from the Methods, a direct interpretation of the neuro-behavioral results in terms of the types of symptoms contributing to PC3 and relative hyper-/hypo-connectivity of the DMN compared to e.g. healthy controls could facilitate easier comparisons with the findings of past studies (since GBC does not seem to be a very commonly-used measure in the psychosis fMRI literature). Also important since GBC is a summary measure of the average connectivity of a region, and doesn't provide any specificity in terms of which regions in particular are more or less connected within a functional network (an inherent limitation of this measure which warrants further attention).

This is important feedback and we have accordingly provided additional discussion around the possible interpretation of the PC symptom-neural maps in the Discussion. We also added Figure S11, which shows the five *β_PC_GBC* maps computed with PC scores and GBC within CON subjects:

“For example, we named the *PC3* axis “Psychosis Configuration” because of its strong loadings onto conventional “positive” and “negative” PSD symptoms. This PC3 “Psychosis Configuration” showed strong positive variation along neural regions that map onto the canonical default mode network (DMN), which has frequently been implicated in PSD (26–31) suggesting that individuals with severe “positive” symptoms exhibit broadly elevated connectivity with regions of the DMN. On the contrary, this bidirectional “Psychosis Configuration” axis also showed strong negative variation along neural regions that map onto the sensory-motor and associative control regions, also strongly implicated in PSD (32, 33). The “bi-directionality” property of the *PC* symptom-neural maps may thus be desirable for identifying neural features that support individual patient selection. For instance, it may be possible that *PC3* reflects residual untreated psychosis symptoms in this chronic PSD sample, which may reveal key treatment neural targets. In support of this circuit being symptom-relevant, it is notable that we observed a mild association between GBC and *PC* scores in the CON sample (Figure S11).”

We acknowledge that GBC is a linear combination measure that by definition does not provide information on connectivity between any one specific pair of neural regions. However, as shown by highly robust and reproducible neurobehavioral maps, GBC seems to be suitable as a first-pass metric in the absence of a priori assumptions of how specific regional connectivity may map to the PC symptom dimensions, and it has been shown to be sensitive to altered patterns of overall neural connectivity in PSD cohorts (34–37) as well as in models of psychosis (7, 38). Moreover, it is an assumption free method for dimensionality reduction of the neural connectivity matrix (which is a massive feature space). Furthermore, GBC provides neural maps (where each region can be represented by a value, in contrast to full functional connectivity matrices), which were necessary for quantifying the relationship with independent molecular benchmark maps (i.e. pharmacological maps and gene expression maps). We do acknowledge that there are limitations to the method which we now discuss in the paper. Furthermore we agree with the Reviewer that the specific regions implicated in these symptom-neural relationships warrants a more detailed investigation and we plan to develop this further in future studies, such as with seed-based functional connectivity using regions implicated in PSD (e.g. thalamus (29, 33)) or restricted GBC (34) which can summarize connectivity information for a specific network or subset of neural regions. We have provided elaboration and clarification regarding this point in the Discussion:

“Another improvement would be to optimize neural data reduction sensitivity for specific symptom variation (39). We chose to use GBC for our initial geometry characterizations as it is a principled and assumption-free data-reduction metric that captures (dys)connectivity across the whole brain and generates neural maps (where each region can be represented by a value, in contrast to full functional connectivity matrices) that are necessary for benchmarking against molecular imaging maps. However, GBC is a summary measure that by definition does not provide information regarding connectivity between specific pairs of neural regions, which may prove to be highly symptom-relevant and informative. Thus symptom-neural relationships should be further explored with higher-resolution metrics, such as restricted GBC (34) which can summarize connectivity information for a specific network or subset of neural regions, or seed-based FC using regions implicated in PSD (e.g. thalamus (29, 33)).”

– While the inclusion of cognitive measures for PSD individuals is a main (self-)selling point of the paper, there's very limited focus on the "Cognitive functioning" component (PC2) of the PCA solution. Examining Figure S8K, the GBC map for this cognitive component seems almost to be the inverse for that of the "Psychosis configuration" component (PC3) focused on in the rest of the paper. Since PC3 does not seem to have high loadings from any of the cognitive items, but it is known that psychosis spectrum individuals tend to exhibit cognitive deficits which also have strong predictive power for illness trajectory, some discussion of how multiple univariate neuro-behavioral features could feasibly be used in conjunction with one another could have been really interesting.

This is an important piece of feedback concerning the cognitive measure aspect of the study. As the Reviewer recognizes, cognition is a core element of PSD symptoms and the key reason for including this symptom into the model. Notably, the finding that one dimension captures a substantial proportion of cognitive performance-related variance, independent of other residual symptom axes, has not previously been reported and we fully agree that expanding on this effect is important and warrants further discussion. We would like to take two of the key points from the Reviewers’ feedback and expand further. First, we recognize that upon qualitative inspection PC2 and PC3 neural maps appear strongly anti-correlated. However, as demonstrated in Figure S9O, PC2 and PC3 maps were anti-correlated at r=-0.47. For comparison, the PC2 map was highly anti-correlated with the BACS composite cognitive map (r=-0.81). This implies that the PC2 map in fact reflects unique neural circuit variance that is relevant for cognition, but not necessarily an inverse of the PC3. In other words, these data suggest that there are PSD patients with more (or less) severe cognitive deficits independent of any other symptom axis, which would be in line with the observation that these symptoms are not treatable with antipsychotic medication (and therefore should not correlate with symptoms that are treatable by such medications; i.e. *PC3*). We have now added these points into the revised paper:

Results:

“Notably, the PC2 – Cognitive Functioning dimension, which captured a substantial proportion of cognitive performance-related symptom variance independent of other symptom axes, revealed a circuit that was moderately (anti-)correlated with other PC maps but strongly anti-correlated with the BACS composite cognitive deficit map (*r*=-0.81, Figure S9O). This implies that the *PC2* map reflects unique neural circuit variance that is relevant for cognition, independent of the other *PC* symptom dimensions.”

Discussion:

“The symptom-neural mapping of dimensionality-reduced symptom scores also produced notable neural maps along other *PC* axes (Figure S10). Higher *PC1* scores, indicating higher general functioning, may be associated with lower global connectivity in sensory/cingulate cortices and thalamus, but higher global connectivity with the temporal lobe. […] Given the key role of cognitive deficits in PSD (21), this will be a particularly important direction to pursue, especially in prodromal or early-stage individuals where cognitive dysfunction has been shown to predict illness trajectories (22–25).”

Second, we appreciate the comment that there may be a way to create a ‘conjunction’ approach that combines both PC2 and PC3 information for the purposes of patient selection, we have now also detailed in the Discussion:

We also expanded the section on using scores along several symptom dimensions for informing treatment selection:

“Importantly, on each data-reduced symptom axis, some subset of PSD patients received a score near zero. […] Such studies rarely consider how much variance remains unexplained in the final data-reduced model and, relatedly, if the proportion of explained variance is reproducible across samples. This is a key property for reliable symptom-to-neural mapping.”

Reviewer 1:The paper assessed the relationship between a dimensionality-reduced symptom space and functional brain imaging features based on the large multicentric data of individuals with psychosis-spectrum disorders (PSD).The strength of this study is that (i) in every analysis, the authors provided high-level evidence of reproducibility in their findings, (ii) the study included several control analyses to test other comparable alternatives or independent techniques (e.g., ICA, univariate vs. multivariate), and (iii) correlating to independently acquired pharmacological neuroimaging and gene expression maps, the study highlighted neurobiological validity of their results.Overall the study has originality and several important tips and guidance for behavior-brain mapping, although the paper contains heavy descriptions about data mining techniques such as several dimensionality reduction algorithms (e.g., PCA, ICA, and CCA) and prediction models.

We thank the Reviewer for their insightful comments and we appreciate the positive feedback. Regarding the descriptions of methods and analytical techniques, we have removed these descriptions out of the main Results text and figure captions. Detailed descriptions are still provided in the Methods, so that they do not detract from the core message of the paper but can still be referenced if a reader wishes to look up the details of these methods within the context of our analyses.

Although relatively minors, I also have few points on the weaknesses, including (i) an incomplete description about how to tell the PSD effects from the normal spectrum, (ii) a lack of overarching interpretation for other principal components rather than only the 3rd one, and (iii) somewhat expected results in the stability of PC and relevant indices.

We are very appreciative of the constructive feedback and feel that these revisions have strengthened our paper. We have addressed these points in the revision as following:

i) We are grateful to the Reviewer for bringing up this point as it has allowed us to further explore the interesting observation we made regarding shared versus distinct neural variance in our data. It is important to not confuse the neural PCA (i.e. the independent neural features that can be detected in the PSD and healthy control samples) versus the neuro-behavioral mapping. In other words, both PSD patients and healthy controls are human and therefore there are a number of neural functions that both cohorts exhibit that may have nothing to do with the symptom mapping in PSD patients. For instance, basic regulatory functions such as control of cardiac and respiratory cycles, motor functions, vision, etc. We hypothesized therefore that there are more common than distinct neural features that are on average shared across humans irrespective of their psychopathology status. Consequently, there may only be a ‘residual’ symptom-relevant neural variance. Therefore, in the manuscript we bring up the possibility that a substantial proportion of neural variance may not be clinically relevant. If this is in fact true then removing the shared neural variance between PSD and CON should not drastically affect the reported symptom-neural univariate mapping solution, because this common variance *does not map to clinical features* and therefore is orthogonal statistically. We have now verified this hypothesis quantitatively and have added extensive analyses to highlight this important observation made the Reviewer. We first conducted a PCA using the parcellated GBC data from all 436 PSD and 202 CON (a matrix with dimensions 638 subjects x 718 parcels). We will refer to this as the GBC-PCA to avoid confusion with the symptom/behavioral PCA described elsewhere in the manuscript. This GBC-PCA resulted in 637 independent GBC-PCs. Since PCs are orthogonal to each other, we then partialled out the variance attributable to GBC-PC1 from the PSD data by reconstructing the PSD GBC matrix using only scores and coefficients from the remaining 636 GBC-PCs (*GBC*^ˆ^_*woPC*1_). We then reran the univariate regression as described in Figure 3, using the same five symptom PC scores across 436 PSD. The results are shown in Figure S21 and reproduced below. Removing the first PC of shared neural variance (which accounted for about 15.8% of the total GBC variance across CON and PSD) from PSD data attenuated the statistics slightly (not unexpected as the variance was by definition reduced) but otherwise did not strongly affect the univariate mapping solution.

We repeated the symptom-neural regression next with the first 2 GBC-PCs partialled out of the PSD data Figure S22, with the first 3 PCs parsed out Figure S23, and with the first 4 neural PCs parsed out Figure S24. The symptom-neural maps remain fairly robust, although the similarity with the original *β_PC_GBC* maps does drop as more common neural variance is parsed out. These figures are also shown below:

For comparison, we also computed the *β_PC_GBC* maps for control subjects, shown in Figure S11. In support of the *β_PC_GBC* in PSD being circuit-relevant, we observed only mild associations between GBC and PC scores in healthy controls:

Results:

“All 5 PCs captured unique patterns of GBC variation across the PSD (Figure S10), which were not observed in CON (Figure S11).”

Discussion:

“On the contrary, this bi-directional *“Psychosis Configuration”* axis also showed strong negative variation along neural regions that map onto the sensory-motor and associative control regions, also strongly implicated in PSD (1, 2). The “bi-directionality” property of the *PC* symptom-neural maps may thus be desirable for identifying neural features that support individual patient selection. For instance, it may be possible that *PC3* reflects residual untreated psychosis symptoms in this chronic PSD sample, which may reveal key treatment neural targets. In support of this circuit being symptom-relevant, it is notable that we observed a mild association between GBC and *PC* scores in the CON sample (Figure S11).”

ii) In our original submission we spotlighted PC3 because of its pattern of loadings on to hallmark symptoms of PSD, including strong positive loadings across Positive symptom items in the PANSS and conversely strong negative loadings on to most Negative items. It was necessary to fully examine this dimension in particular because these are key characteristics of the target psychiatric population, and we found that the focus on PC3 was innovative because it provided an opportunity to quantify a fully data-driven dimension of symptom variation that is highly characteristic of the PSD patient population. Additionally, this bi-directional axis captured shared variance from measures in other traditional symptoms factors, such the PANSS General factor and cognition. This is a powerful demonstration of how data-driven techniques such as PCA can reveal properties intrinsic to the structure of PSD-relevant symptom data which may in turn improve the mapping of symptom-neural relationships. We refrained from explaining each of the five PCs in detail in the main text as we felt that it would further complicate an already dense manuscript. Instead, we opted to provide the interpretation and data from all analyses for all five PCs in the Supplement. However, in response to the Reviewers’ thoughtful feedback that more focus should be placed on other components, we have expanded the presentation and discussion of all five components (both regarding the symptom profiles and neural maps) in the main text:

Results:

“Because *PC3* loads most strongly on to hallmark symptoms of PSD (including strong positive loadings across PANSS Positive symptom measures in the PANSS and strong negative loadings onto most Negative measures), we focus on this PC as an opportunity to quantify an innovative, fully data-driven dimension of symptom variation that is highly characteristic of the PSD patient population. Additionally, this bi-directional symptom axis captured shared variance from measures in other traditional symptoms factors, such the PANSS General factor and cognition. We found that the *PC3* result provided a powerful empirical demonstration of how using a data-driven dimensionality-reduced solution (via PCA) can reveal novel patterns intrinsic to the structure of PSD psychopathology.”

We felt that demonstrating the stability of the PCA solution was extremely important, given that this degree of rigor has not previously been tested using broad behavioral measures across psychosis symptoms and cognition in a cross-diagnostic PSD sample. Additionally, we demonstrated reproducibility of the PCA solution using independent split-half samples. Furthermore, we derived stable neural maps using the PCA solution. In our original submission we show that the CCA solution was not reproducible in our dataset. Following the Reviewers’ feedback, we computed the estimated sample sizes needed to sufficiently power our multivariate analyses for stable/reproducible solutions. using the methods in (3). These results are

Reviewer 2:The work by Ji et al., is an interesting and rather comprehensive analysis of the trend of developing data-driven methods for developing brain-symptom dimension biomarkers that bring a biological basis to the symptoms (across PANSS and cognitive features) that relate to psychotic disorders. To this end, the authors performed several interesting multivariate analyses to decompose the symptom/behavioural dimensions and functional connectivity data. To this end, the authors use data from individuals from a transdiagnostic group of individuals recruited by the BSNIP cohort and combine high-level methods in order to integrate both types of modalities. Conceptually there are several strengths to this paper that should be applauded. However, I do think that there are important aspects of this paper that need revision to improve readability and to better compare the methods to what is in the field and provide a balanced view relative to previous work with the same basic concepts that they are building their work around. Overall, I feel as though the work could advance our knowledge in the development of biomarkers or subject level identifiers for psychiatric disorders and potentially be elevated to the level of an individual "subject screener". While this is a noble goal, this will require more data and information in the future as a means to do this. This is certainly an important step forward in this regard.

We thank the Reviewer for their insightful and constructive comments about our manuscript. We have revised the text to make it easier to read and to clarify our results in the context of prior works in the field. We fully agree that a great deal more work needs to be completed before achieving single-subject level treatment selection, but we hope that our manuscript provides a helpful step towards this goal.

Strengths:– Combined analysis of canonical psychosis symptoms and cognitive deficits across multiple traditional psychosis-related diagnoses offers one of the most comprehensive mappings of impairments experienced within PSD to brain features to date.– Cross-validation analyses and use of various datasets (diagnostic replication, pharmacological neuroimaging) is extremely impressive, well motivated, and thorough. In addition the authors use a large dataset and provide "out of sample" validity.– Medication status and dosage also accounted for.– Similarly, the extensive examination of both univariate and multivariate neuro-behavioural solutions from a methodological viewpoint, including the testing of multiple configurations of CCA (i.e. with different parcellation granularities), offers very strong support for the selected symptom-to-neural mapping.– The plots of the obtained PC axes compared to those of standard clinical symptom aggregate scales provide a really elegant illustration of the differences and demonstrate clearly the value of data-driven symptom reduction over conventional categories- The comparison of the obtained neuro-behavioural map for the "Psychosis configuration" symptom dimension to both pharmacological neuroimaging and neural gene expression maps highlights direct possible links with both underlying disorder mechanisms and possible avenues for treatment development and application.- The authors' explicit investigation of whether PSD and healthy controls share a major portion of neural variance (possibly present across all people) has strong implications for future brain-behaviour mapping studies, and provides a starting point for narrowing the neural feature space to just the subset of features showing symptom-relevant variance in PSD

We are very grateful for the positive feedback. We would like to thank the Reviewers for taking the time to read this admittedly dense manuscript and for providing their helpful critique.

Critiques:– Overall I found the paper very hard to read. There are abbreviation everywhere for every concept that is introduced. The paper is methods heavy (which I am not opposed to and quite like). It is clear that the authors took a lot of care in thinking about the methods that were chosen. That said, I think that the organization would benefit from a more traditional Intro, Methods, Results, and Discussion formatting so that it would be easier to parse the Results. The figures are extremely dense and there are often terms that are coined or used that are not or poorly defined.

We appreciate the constructive feedback around how to remove the dense content and to pay more attention to the frequency of abbreviations, which impact readability. We implemented the strategies suggested by the Reviewer and have moved the Methods section after the Introduction to make the subsequent Results section easier to understand and contextualize. For clarity and length, we have moved methodological details previously in the Results and figure captions to the Methods (e.g. descriptions of dimensionality reduction and prediction techniques). This way, the Methods are now expanded for clarity without detracting from the readability of the core results of the paper. Also, we have also simplified the text in places where there was room for more clarity. For convenience and ease of use of the numerous abbreviations, we have also added a table to the Supplement (Supplementary Table S1):

– One thing I found conceptually difficult is the explicit comparison to the work in the Xia paper from the Satterthwaite group. Is this a fair comparison? The sample is extremely different as it is non clinical and comes from the general population. Can it be suggested that the groups that are clinically defined here are comparable? Is this an appropriate comparison and standard to make. To suggest that the work in that paper is not reproducible is flawed in this light.

This is an extremely important point to clarify and we apologize that we did not make it sufficiently clear in the initial submission. Here we are not attempting to replicate the results of Xia et al., which we understand were derived in a fundamentally different sample than ours both demographically and clinically, with testing very different questions. Rather, this paper is just one example out of a number of recent papers which employed multivariate methods (CCA) to tackle the mapping between neural and behavioral features. The key point here is that this approach does not produce reproducible results due to over-fitting, as demonstrated robustly in the present paper. It is very important to highlight that in fact we did not single out any one paper when making this point. In fact, we do not mention the Xia paper explicitly anywhere and we were very careful to cite multiple papers in support of the multivariate over-fitting argument, which is now a well-know issue (4). Nevertheless, the Reviewers make an excellent point here and we acknowledge that while CCA was not reproducible in the present dataset, this does not explicitly imply that the results in the Xia et al. paper (or any other paper for that matter) are not reproducible by definition (i.e. until someone formally attempts to falsify them). We have made this point explicit in the revised paper, as shown below. Furthermore, in line with the provided feedback, we also applied the multivariate power calculator derived by Helmer et al. (3), which quantitatively illustrates the statistical point around CCA instability.

Results:

“Several recent studies have reported “latent” neuro-behavioral relationships using multivariate statistics (5–7), which would be preferable because they simultaneously solve for maximal covariation across neural and behavioral features. Given the possibility of deriving a stable multivariate effect, here we tested if results improve with canonical correlation analysis (CCA) (8) (Figure 5A).”

Discussion:

“Here we attempted to use multivariate solutions (i.e. CCA) to quantify symptom and neural feature covariation. In principle, CCA is well-suited to address the brain-behavioral mapping problem. However, symptom-neural mapping using CCA in our sample was not reproducible even when using a low-dimensional symptom solution and parcellated neural data as a starting point. Therefore, while CCA (and related multivariate methods such as partial least squares) are theoretically appropriate and may be helped by regularization methods such as sparse CCA, in practice many available psychiatric neuroimaging datasets may not provide sufficient power to resolve stable multivariate symptom-neural solutions (3). A key pressing need for forthcoming studies will be to use multivariate power calculators to inform sample sizes needed for resolving stable symptom-neural geometries at the single subject level. Of note, though we were unable to derive a stable CCA in the present sample, this does not imply that the multivariate neuro-behavioral effect may not be reproducible with larger effect sizes and/or sample sizes. Critically, this does highlight the importance of power calculations prior to computing multivariate brain-behavioral solutions (3).”

– Why was PCA selected for the analysis rather than ICA? Authors mention that PCA enables the discovery of orthogonal symptom dimensions, but don't elaborate on why this is expected to better capture behavioural variation within PSD compared to non-orthogonal dimensions. Given that symptom and/or cognitive items in conventional assessments are likely to be correlated in one way or another, allowing correlations to be present in the low-rank behavioural solution may better represent the original clinical profiles and drive more accurate brain-behaviour mapping. Moreover, as alluded to in the Discussion, employing an oblique rotation in the identification of dimensionality-reduced symptom axes may have actually resulted in a brain-behaviour space that is more generalizable to other psychiatric spectra. Why not use something more relevant to symptom/behaviour data like a factor analysis?

This is a very important point! We agree with the Reviewer that an oblique solution may better fit the data. For this reason, we performed an ICA as shown in the Supplement. We chose to show PCA for the main analyses here because it is a deterministic solution and the number of significant components could be computed via permutation testing. Importantly, certain components from the ICA solution in this sample were highly similar to the PCs shown in the main solution (Supplementary Note 1), as measured by comparing the subject behavioral scores (Figure S4), and neural maps (Figure S13). However, notably, certain components in the ICA and PCA solutions did not appear to have a one-to-one mapping (e.g. PCs 1-3 and ICs 1-3). The orthogonality of the PCA solution forces the resulting components to capture maximally separated, unique symptom variance, which in turn map robustly on to unique neural circuits. We observed that the data may be distributed in such a way that in the ICA highly correlated independent components emerge, which do not maximally separate the symptom variance associate with neural variance. We demonstrate this by plotting the relationship between parcel β coefficients for the *β_PC_*_3_*GBC* map versus the *β_IC_*_2_*GBC* and *β_IC_*_3_*GBC* maps. The sigmoidal shape of the distribution indicates an improvement in the Z-statistics for the *β_PC_*_3_*GBC* map relative to the *β_IC_*_2_*GBC* and *β_IC_*_3_*GBC* maps. We have added this language to the main text Results:

Notably, independent component analysis (ICA), an alternative dimensionality reduction procedure which does not enforce component orthogonality, produced similar effects for this PSD sample (see Supplementary Note 1 & Figure S4A). Certain pairs of components between the PCA and ICA solutions appear to be highly similar and exclusively mapped (IC5 and PC4; IC4 and PC5) (Figure S4B). On the other hand, PCs 1-3 and ICs 1-3 do not exhibit a one-to-one mapping. For example, *PC3* appears to correlate positively with *IC2* and equally strongly negatively with *IC3*, suggesting that these two ICs are oblique to the PC and perhaps reflect symptom variation that is explained by a single PC. The orthogonality of the PCA solution forces the resulting components to capture maximally separated, unique symptom variance, which in turn map robustly on to unique neural circuits. We observed that the data may be distributed in such a way that in the ICA highly correlated independent components emerge, which do not maximally separate the symptom variance associate with neural variance. We demonstrate this by plotting the relationship between parcel β coefficients for the *β_PC_*_3_*GBC* map versus the *β_IC_*_2_*GBC* and *β_IC_*_3_*GBC* maps (Figure ?G). The sigmoidal shape of the distribution indicates an improvement in the Z-statistics for the *β_PC_*_3_*GBC* map relative to the *β_IC_*_2_*GBC* and *β_IC_*_3_*GBC* maps.”

Additionally, the Reviewer raises an important point, and we agree that orthogonal versus oblique solutions warrant further investigation especially with regards to other psychiatric spectra and/or other stages in disease progression. For example, oblique components may better capture dimensions of behavioral variation in prodromal individuals, as these individuals are in the early stages of exhibiting psychosis-relevant symptoms and may show early diverging of dimensions of behavioral variation. We elaborate on this further in the Discussion:

“Another important aspect that will require further characterization is the possibility of oblique axes in the symptom-neural geometry. While orthogonal axes derived via PCA were appropriate here and similar to the ICA-derived axes in this solution, it is possible that oblique dimensions more clearly reflect the geometry of other psychiatric spectra and/or other stages in disease progression. For example, oblique components may better capture dimensions of neuro-behavioral variation in a sample of prodromal individuals, as these patients are exhibiting early-stage psychosis-like symptoms and may show signs of diverging along different trajectories.

Critically, these factors should constitute key extensions of an iteratively more robust model for individualized symptom-neural mapping across the PSD and other psychiatric spectra. Relatedly, it will be important to identify the ‘limits’ of a given BBS solution – namely a PSD-derived effect may not generalize into the mood spectrum (i.e. both the symptom space and the resulting symptom-neural mapping is orthogonal). It will be important to evaluate if this framework can be used to initialize symptom-neural mapping across other mental health symptom spectra, such as mood/anxiety disorders.”

– The gene expression mapping section lacks some justification for why the 7 genes of interest were specifically chosen from among the numerous serotonin and GABA receptors and interneuron markers (relevant for PSD) available in the AHBA. Brief reference to the believed significance of the chosen genes in psychosis pathology would have helped to contextualize the observed relationship with the neuro-behavioural map.

We thank the Reviewer for providing this suggestion and agree that it will strengthen the section on gene expression analysis. Of note, we did justify the choice for these genes, but we appreciate the opportunity to expand on the neurobiology of selected genes and their relevance to PSD. We have made these edits to the text:

“We focus here on serotonin receptor subunits (HTR1E, HTR2C, HTR2A), GABA receptor subunits (GABRA1, GABRA5), and the interneuron markers somatostatin (SST) and parvalbumin (PVALB). Serotonin agonists such as LSD have been shown to induce PSD-like symptoms in healthy adults (9) and the serotonin antagonism of “second-generation” antipsychotics are thought to contribute to their efficacy in targeting broad PSD symptoms (10–12). Abnormalities in GABAergic interneurons, which provide inhibitory control in neural circuits, may contribute to cognitive deficits in PSD (13–15) and additionally lead to downstream excitatory dysfunction that underlies other PSD symptoms (16, 17). In particular, a loss of prefrontal parvalbumin-expression fast-spiking interneurons has been implicated in PSD (18–21).”

– What the identified univariate neuro-behavioural mapping for PC3 ("psychosis configuration") actually means from an empirical or brain network perspective is not really ever discussed in detail. E.g., in Results, "a high positive PC3 score was associated with both reduced GBC across insular and superior dorsal cingulate cortices, thalamus, and anterior cerebellum and elevated GBC across precuneus, medial prefrontal, inferior parietal, superior temporal cortices and posterior lateral cerebellum." While the meaning and calculation of GBC can be gleaned from the Methods, a direct interpretation of the neuro-behavioural results in terms of the types of symptoms contributing to PC3 and relative hyper-/hypo-connectivity of the DMN compared to e.g. healthy controls could facilitate easier comparisons with the findings of past studies (since GBC does not seem to be a very commonly-used measure in the psychosis fMRI literature). Also important since GBC is a summary measure of the average connectivity of a region, and doesn't provide any specificity in terms of which regions in particular are more or less connected within a functional network (an inherent limitation of this measure which warrants further attention).

We acknowledge that GBC is a linear combination measure that by definition does not provide information on connectivity between any one specific pair of neural regions. However, as shown by highly robust and reproducible neurobehavioral maps, GBC seems to be suitable as a first-pass metric in the absence of a priori assumptions of how specific regional connectivity may map to the PC symptom dimensions, and it has been shown to be sensitive to altered patterns of overall neural connectivity in PSD cohorts (22–25) as well as in models of psychosis (9, 26). Moreover, it is an assumption free method for dimensionality reduction of the neural connectivity matrix (which is a massive feature space). Furthermore, GBC provides neural maps (where each region can be represented by a value, in contrast to full functional connectivity matrices), which were necessary for quantifying the relationship with independent molecular benchmark maps (i.e. pharmacological maps and gene expression maps). We do acknowledge that there are limitations to the method which we now discuss in the paper. Furthermore we agree with the Reviewer that the specific regions implicated in these symptom-neural relationships warrants a more detailed investigation and we plan to develop this further in future studies, such as with seed-based functional connectivity using regions implicated in PSD (e.g. thalamus (2, 27)) or restricted GBC (22) which can summarize connectivity information for a specific network or subset of neural regions. We have provided elaboration and clarification regarding this point in the Discussion:

“Another improvement would be to optimize neural data reduction sensitivity for specific symptom variation (28). We chose to use GBC for our initial geometry characterizations as it is a principled and assumption-free data-reduction metric that captures (dys)connectivity across the whole brain and generates neural maps (where each region can be represented by a value, in contrast to full functional connectivity matrices) that are necessary for benchmarking against molecular imaging maps. However, GBC is a summary measure that by definition does not provide information regarding connectivity between specific pairs of neural regions, which may prove to be highly symptom-relevant and informative. Thus symptom-neural relationships should be further explored with higher-resolution metrics, such as restricted GBC (22) which can summarize connectivity information for a specific network or subset of neural regions, or seed-based FC using regions implicated in PSD (e.g. thalamus (2, 27)).”

– Possibly a nitpick, but while the inclusion of cognitive measures for PSD individuals is a main (self-)selling point of the paper, there's very limited focus on the "Cognitive functioning" component (PC2) of the PCA solution. Examining Figure S8K, the GBC map for this cognitive component seems almost to be the inverse for that of the "Psychosis configuration" component (PC3) focused on in the rest of the paper. Since PC3 does not seem to have high loadings from any of the cognitive items, but it is known that psychosis spectrum individuals tend to exhibit cognitive deficits which also have strong predictive power for illness trajectory, some discussion of how multiple univariate neuro-behavioural features could feasibly be used in conjunction with one another could have been really interesting.

This is an important piece of feedback concerning the cognitive measure aspect of the study. As the Reviewer recognizes, cognition is a core element of PSD symptoms and the key reason for including this symptom into the model. Notably, the finding that one dimension captures a substantial proportion of cognitive performance-related variance, independent of other residual symptom axes, has not previously been reported and we fully agree that expanding on this effect is important and warrants further discussion. We would like to take two of the key points from the Reviewers’ feedback and expand further. First, we recognize that upon qualitative inspection PC2 and PC3 neural maps appear strongly anti-correlated. However, as demonstrated in Figure S9O, PC2 and PC3 maps were anti-correlated at r=-0.47. For comparison, the PC2 map was highly anti-correlated with the BACS composite cognitive map (r=-0.81). This implies that the PC2 map in fact reflects unique neural circuit variance that is relevant for cognition, but not necessarily an inverse of the PC3.

In other words, these data suggest that there are PSD patients with more (or less) severe cognitive deficits independent of any other symptom axis, which would be in line with the observation that these symptoms are not treatable with antipsychotic medication (and therefore should not correlate with symptoms that are treatable by such medications; i.e. *PC3*). We have now added these points into the revised paper:

Results

“Figure 1E highlights loading configurations of symptom measures forming each PC. To aid interpretation, we assigned a name for each PC based on its most strongly weighted symptom measures. This naming is qualitative but informed by the pattern of loadings of the original 36 symptom measures (Figure 1). For example, *PC1* was highly consistent with a general impairment dimension (i.e. “Global Functioning”); *PC2* reflected more exclusively variation in cognition (i.e. “Cognitive Functioning”); *PC3* indexed a complex configuration of psychosis-spectrum relevant items (i.e. “Psychosis Configuration”); *PC4* generally captured variation mood and anxiety related items (i.e. “Affective Valence”); finally, *PC5* reflected variation in arousal and level of excitement (i.e. “Agitation/Excitation”). For instance, a generally impaired patient would have a highly negative *PC1* score, which would reflect low performance on cognition and elevated scores on most other symptomatic items. Conversely, an individual with a high positive *PC3* score would exhibit delusional, grandiose, and/or hallucinatory behavior, whereas a person with a negative *PC3* score would exhibit motor retardation, social avoidance, possibly a withdrawn affective state with blunted affect (29). Comprehensive loadings for all 5 PCs are shown in Figure 3G. Figure 1F highlights the mean of each of the 3 diagnostic groups (colored spheres) and healthy controls (black sphere) projected into a 3-dimensional orthogonal coordinate system for *PCs 1,2 & 3* (x,y,z axes respectively; alternative views of the 3-dimensional coordinate system with all patients projected are shown in Figure 3). Critically, PC axes were not parallel with traditional aggregate symptom scales. For instance, *PC3* is angled at ∼45° to the dominant direction of PANSS Positive and Negative symptom variation (purple and blue arrows respectively in Figure 1F).

[…]

Because *PC3* loads most strongly on to hallmark symptoms of PSD (including strong positive loadings across PANSS Positive symptom measures in the PANSS and strong negative loadings onto most Negative measures), we focus on this PC as an opportunity to quantify an innovative, fully data-driven dimension of symptom variation that is highly characteristic of the PSD patient population. Additionally, this bi-directional symptom axis captured shared variance from measures in other traditional symptoms factors, such the PANSS General factor and cognition. We found that the *PC3* result provided a powerful empirical demonstration of how using a data-driven dimensionality-reduced solution (via PCA) can reveal novel patterns intrinsic to the structure of PSD psychopathology.”

Another nitpick, but the Y axes of Figure 8C-E are not consistent, which causes some of the lines of best fit to be a bit misleading (e.g. GABRA1 appears to have a more strongly positive gene-PC relationship than HTR1E, when in reality the opposite is true.)

We have scaled each axis to best show the data in each plot but see how this is confusing and recognise the need to correct this. We have remade the plots with consistent axes labelling.

– The authors explain the apparent low reproducibility of their multivariate PSD neuro-behavioural solution using the argument that many psychiatric neuroimaging datasets are too small for multivariate analyses to be sufficiently powered. Applying an existing multivariate power analysis to their own data as empirical support for this idea would have made it even more compelling. The following paper suggests guidelines for sample sizes required for CCA/PLS as well as a multivariate calculator: Helmer et al., (2020). On stability of Canonical Correlation Analysis and Partial Least Squares with application to brain-behavior associations (p. 2020.08.25.265546). https://doi.org/10.1101/2020.08.25.265546

We deeply appreciate the Reviewer’s suggestion and the opportunity to incorporate the methods from the Helmer et al., paper. We now highlight the importance of having sufficiently powered samples for multivariate analyses in our other manuscript first-authored by our colleague Dr. Markus Helmer (3). Using the method described in the above paper (GEMMR version 0.1.2), we computed the estimated sample sizes required to power multivariate CCA analyses with 718 neural features and 5 behavioral (PC) features (i.e. the feature set used throughout the rest of the paper):

if *rtrue* = 0.1: *nrequired* = 421,608

if *rtrue* = 0.3: *nrequired* = 33,166

if *rtrue* = 0.5: *nrequired* = 10,168

As argued in Helmer et al., *r_true_
*is likely below 0.3 in many cases, thus the estimated sample size of ∼33k is likely a lower bound for the required sample size for sufficiently-powered CCA analyses using the 718+5 features leveraged throughout the univariate analyses in the present manuscript. This number is two orders of magnitude greater than our available sample (and at least one order of magnitude greater than any single existing clinical dataset). Even if *r_true_
*is 0.5, a sample size of ∼10k would likely be required.

if *rtrue* = 0.1: *nrequired* = 421,608

if *rtrue* = 0.3: *nrequired* = 33,166

if *rtrue* = 0.5: *nrequired* = 10,168

As argued in Helmer et al., *r_true_
*is likely below 0.3 in many cases, thus the estimated sample size of ∼33k is likely a lower bound for the required sample size for sufficiently-powered CCA analyses using the 718+5 features leveraged throughout the univariate analyses in the present manuscript. This number is two orders of magnitude greater than our available sample (and at least one order of magnitude greater than any single existing clinical dataset). Even if *r_true_
*is 0.5, a sample size of ∼10k would likely be required.

We also computed the estimated sample sizes required for 180 neural features (symmetrized neural cortical parcels) and 5 symptom PC features, consistent with the CCA reported in our main text:

if *rtrue* = 0.1: *nrequired* = 103,545

if *rtrue* = 0.3: *nrequired* = 8,145

if *rtrue* = 0.5: *nrequired* = 2,497

Assuming that *r_true_
*is likely below 0.3, this minimal required sample size remains at least an order of magnitude greater than the size of our present sample, consistent with the finding that the CCA solution computed using these data was unstable.

As a lower limit for the required sample size plausible using the feature sets reported in our paper, we additionally computed for comparison the estimated N needed with the smallest number of features explored in our analyses, i.e. 12 neural functional network features and 5 symptom PC features:

if *rtrue* = 0.1: *nrequired* = 8,855

if *rtrue* = 0.3: *nrequired* = 696

if *rtrue* = 0.5: *nrequired* = 213

These required sample sizes are closer to the N=436 used in the present sample and samples reported in the clinical neuroimaging literature. This is consistent with the observation that when using 12 neural and 5 symptom features (Figure S15C) the detected canonical correlation *r* = 0.38 for CV1 is much lower (and likely not inflated due to overfitting) and may be closer to the true effect because with the n=436 this effect is resolvable. This is in contrast to the 180 neural features and 5 symptom feature CCA solution where we observed a null CCA effect around *r >* 0.6 across all 5 CVs. This clearly highlights the inflation of the effect in the situation where the feature space grows. There is no a priori plausible reason to believe that the effect for 180 vs. 5 feature mapping is literally double the effect when using 12 vs. 5 feature mapping – especially as the 12 features are networks derived from the 180 parcels (i.e. the effect should be comparable rather than 2x smaller). Consequently, if the true CCA effect with 180 vs. 5 features was actually in the more comparable *r* = 0.38, we would need >5,000 subjects to resolve a reproducible neuro-behavioral CCA map (an order of magnitude more than in the BSNIP sample). Moreover, to confidently detect effects if *r_true_
*is actually less than 0.3, we would require a sample size >8,145 subjects. We have added this to the Results section on our CCA results:

“Next, we tested if the 180-parcel CCA solution is stable and reproducible, as done with PC-to-GBC univariate results. The CCA solution was robust when tested with *k*-fold and leave-site-out crossvalidation (Figure S16) likely because these methods use CCA loadings derived from the full sample. However, the CCA loadings did not replicate in non-overlapping split-half samples (Figure 5L, see see Supplementary Note 4). Moreover, a leave-one-subject-out cross-validation revealed that removing a single subject from the sample affected the CCA solution such that it did not generalize to the left-out subject (Figure 5M). This is in contrast to the PCA-to-GBC univariate mapping, which was substantially more reproducible for all attempted cross-validations relative to the CCA approach. This is likely because substantially more power is needed to resolve a stable multivariate neuro-behavioral effect with this many features. Indeed, a multivariate power analysis using 180 neural features and 5 symptom features, and assuming a true canonical correlation of *r* = 0.3, suggests that a minimal sample size of *N* = 8145 is needed to sufficiently detect the effect (3), an order of magnitude greater than the available sample size. Therefore, we leverage the univariate neuro-behavioral result for subsequent subject-specific model optimization and comparisons to molecular neuroimaging maps.”

Additionally, we added the following to Supplementary Note 4: Establishing the Reproducibility of the CCA Solution:

“Here we outline the details of the split-half replication for the CCA solution. Specifically, the full patient sample was randomly split (referred to as “H1” and “H2” respectively), while preserving the proportion of patients in each diagnostic group. Then, CCA was performed independently for H1 and H2. While the loadings for behavioral PCs and original behavioral items are somewhat similar (mean *r* ≈ 0.5) between the two CCAs in each run, the neural loadings were not stable across H1 and H2 CCA solutions. Critically, CCA results did not perform well for leave-one-subject-out cross-validation (Figure 5M). Here, one patient was held out while CCA was performed using all data from the remaining 435 patients. The loadings matrices *Ψ* and *Θ* from the CCA were then used to calculate the “predicted” neural and behavioral latent scores for all 5 CVs for the patient that was held out of the CCA solution. This process was repeated for every patient and the final result was evaluated for reproducibility. As described in the main text, this did not yield reproducible CCA effects (Figure 5M).

Of note, CCA may yield higher reproducibility if the neural feature space were to be further reduced. As noted, our approach was to first parcellate the BOLD signal and then use GBC as a data-driven method to yield a neuro-biologically and quantitatively interpretable neural data reduction, and we additionally symmetrized the result across hemispheres. Nevertheless, in sharp contrast to the PCA univariate feature selection approach, the CCA solutions were still not stable in the present sample size of *N* = 436. Indeed, a multivariate power analysis (3) estimates that the following sample sizes will be required to sufficiently power a CCA between 180 neural features and 5 symptom features, at different levels of true canonical correlation (*r_t_rue*):

if *rtrue* = 0.1: *nrequired* = 103,545

if *rtrue* = 0.3: *nrequired* = 8,145

if *rtrue* = 0.5: *nrequired* = 2,497

To test if further neural feature space reduction may be improve reproducibility, we also evaluated CCA solutions with neural GBC parcellated according to 12 brain-wide functional networks derived from the recent HCP driven network parcellation (30). Again, we computed the CCA for all 36 item-level symptom as well as 5 PCs (Figure S15). As with the parcel-level effects, the network-level CCA analysis produced significant results (for CV1 when using 36 item-level scores and for all 5 CVs when using the 5 PC-derived scores). Here the result produced much lower canonical correlations (∼0.3-0.5); however, these effects (for CV1) clearly exceeded the 95% confidence interval generated via random permutations, suggesting that they may reflect the true canonical correlation. We observed a similar result when we evaluated CCAs computed with neural GBC from 192 symmetrized subcortical parcels and 36 symptoms or 5 PCs (Figure S14). In other words, data-reducing the neural signal to 12 functional networks likely averaged out parcel-level information that may carry symptom-relevant variance, but may be closer to capturing the true effect. Indeed, the power analysis suggests that the current sample size is closer to that needed to detect an effect with 12 + 5 features:

if *rtrue* = 0.1: *nrequired* = 8,855

if *rtrue* = 0.3: *nrequired* = 696

if *rtrue* = 0.5: *nrequired* = 213

Note that we do not present a CCA conducted with parcels across the whole brain, as the number of variables would exceed the number of observations. However, the multivariate power analysis using 718 neural features and 5 symptom features estimates that the following sample sizes would be required to detect the following effects:

if *rtrue* = 0.1: *nrequired* = 421,608

if *rtrue* = 0.3: *nrequired* = 33,166

if *rtrue* = 0.5: *nrequired* = 10,168

This analysis suggests that even the lowest bound of ∼10k samples exceeds the present available sample size by two orders of magnitude.”

We have also added Figure S19, illustrating these power analyses results:

We also added the corresponding methods in the Methods section:

“Multivariate CCA Power Analysis. Multivariate power analyses to estimate the minimum sample size needed to sufficiently power a CCA were computed using methods described in (3), using the Generative Modeling of Multivariate Relationships tool (*gemmr*, https://github.com/murraylab/gemmr (v0.1.2)). Briefly, a model was built by: 1) Generating synthetic datasets for the two input data matrices, by sampling from a multivariate normal distribution with a joint covariance matrix that was structured to encode CCA solutions with specified properties; 2) Performing CCAs on these synthetic datasets. Because the joint covariance matrix is known, the true values of estimated association strength, weights, scores, and loadings of the CCA, as well as the errors for these four metrics, can also be computed. In addition, statistical power that the estimated association strength is different from 0 is determined through permutation testing; 3) Varying parameters of the generative model (number of features, assumed true between-set correlation, within-set variance structure for both datasets) the required sample size *N*_req_ is determined in each case such that statistical power reaches 90% and all of the above described error metrics fall to a target level of 10%; and 4) Fitting and validating a linear model to predict the required sample size *N*_req_ from parameters of the generative model. This linear model was then used to calculate *N*_req_ for CCA in three data scenarios: i) 718 neural vs. 5 symptom features; ii) 180 neural vs. 5 symptom features; iii) 12 neural vs. 5 symptom features.”

– Given the relatively even distribution of males and females in the dataset, some examination of sex effects on symptom dimension loadings or neuro-behavioural maps would have been interesting (other demographic characteristics like age and SES are summarized for subjects but also not investigated). I think this is a missed opportunity.

We have now provided additional analyses for the core PCA and univariate GBC mapping results, testing for effects of age, sex, and SES in Figure S8. Briefly, we observed a significant positive relationship between age and PC3 scores, which may be because older patients (whom presumably have been ill for a longer time) exhibit more severe symptoms along the positive PC3 – Psychosis Configuration dimension. We also observed a significant negative relationship between Hollingshead index of SES and PC1 and PC2 scores. Lower PC1 and PC2 scores indicate poorer general functioning and cognitive performance respectively, which is consistent with higher Hollingshead indices (i.e. lower-skilled jobs or unemployment and fewer years of education). We also found significant sex differences in PC2 – Cognitive Functioning, PC4 – Affective Valence, and PC5 – Agitation/Excitement scores.